# Spherical Motion Dynamics:
# Learning Dynamics of Normalized Neural Network using SGD and Weight Decay

**Ruosi Wan**
Megvii Technology
`wanruosi@megvii.com`

**Zhanxing Zhu**
University of Edinburgh
`zhanxing.zhu@ed.ac.uk`
`zhanxing.zhu@pku.edu.cn`

**Xiangyu Zhang**
Megvii Technology
`zhangXiangyu@megvii.com`

**Jian Sun**
Megvii Technology
`sunjian@megvii.com`

## Abstract

In this paper, we comprehensively reveal the learning dynamics of normalized neural network using Stochastic Gradient Descent (with momentum) and Weight Decay (WD), named as Spherical Motion Dynamics (SMD). Most related works on this topic focus on studying "effective learning rate" using "equilibrium" assumption, i.e. assuming weight norm has converge to a fixed value. However, their discussion on why equilibrium can be reached is either absent or unjustified. To clarify the mechanism behind, our work directly explores the cause of equilibrium, which should be regarded as a special state of SMD. Specifically, 1) we introduce the assumptions that can lead to equilibrium state in SMD, and prove equilibrium can be reached in a linear rate regime; 2) we propose "angular update" as a substitute for effective learning rate to depict the state of SMD, and derive the theoretical value of angular update in equilibrium state; 3) we verify our assumptions and theoretical results on various large-scale computer vision tasks including ImageNet and MSCOCO with standard settings. Experiment results show our theoretical findings agree well with empirical observations. Furthermore, we provide intuitive interpretations, showing how the behavior of angular update in SMD affects the optimization of neural network, and yields unexpected phenomenon in practice. We believe our findings and theoretical results can deepen our understanding on current training techniques for deep neural network.

## 1 Introduction

Normalization techniques (e.g. Batch Normalization (Ioffe & Szegedy, 2015) or its variants) are one of the most commonly adopted techniques for training deep neural networks (DNN). A typical normalization can be formulated as following: consider a single unit in a neural network, the input is $\boldsymbol{X}$, the weight of linear layer is $\boldsymbol{w}$ (bias is included in $\boldsymbol{w}$), then its output is

$$y(\boldsymbol{X}; \boldsymbol{w}; \gamma; \beta) = g\left(\frac{\boldsymbol{X}\boldsymbol{w} - \mu(\boldsymbol{X}\boldsymbol{w})}{\sigma(\boldsymbol{w}\boldsymbol{X})}\gamma + \beta\right), \tag{1}$$

where $g$ is a nonlinear activation function like ReLU or sigmoid, $\mu$, $\sigma$ are mean and standard deviation computed across specific dimension of $\boldsymbol{X}\boldsymbol{w}$ (like Batch Normalization (Ioffe & Szegedy, 2015), Layer Normalization Ba et al. (2016), Group Normalization (Wu & He, 2018), etc.). $\beta$, $\gamma$ are learnable

35th Conference on Neural Information Processing Systems (NeurIPS 2021).

parameters to remedy for the limited range of normalized feature map. Aside from normalizing feature map, Salimans & Kingma (2016) normalizes weight by $l^2$ norm instead:

$$y(\boldsymbol{X}; \boldsymbol{w}; \gamma; \beta) = g(\boldsymbol{X} \frac{\boldsymbol{w}}{||\boldsymbol{w}||_2} \gamma + \beta), \tag{2}$$

where $|| \cdot ||_2$ denotes $l_2$ norm of a vector. Though formulated in different manners, all normalization techniques mentioned above share an interesting property: *scale-invariance*

**Definition 1** (Scale-invariance). *Given loss function $\mathcal{L}(\boldsymbol{w})$, $\boldsymbol{w}$ is scale-invariant w.r.t. $\mathcal{L}$ if and only if $\forall k \in \mathbb{R}^+$, we have $\mathcal{L}(\boldsymbol{w}) = \mathcal{L}(k\boldsymbol{w})$.*

By definition of scale-invariance, we can directly derive the following properties of scale-invariant weights in Lemma 1

**Lemma 1.** *If $\boldsymbol{w}$ is scale-invariant with respect to $\mathcal{L}(\boldsymbol{w})$ , then for all $k > 0$, we have:*

$$\langle \boldsymbol{w}_t, \frac{\partial \mathcal{L}}{\partial \boldsymbol{w}} \Big|_{\boldsymbol{w}=\boldsymbol{w}_t} \rangle = 0 \tag{3}$$

$$\frac{\partial \mathcal{L}}{\partial \boldsymbol{w}} \Big|_{\boldsymbol{w}=k\boldsymbol{w}_t} = \frac{1}{k} \cdot \frac{\partial \mathcal{L}}{\partial \boldsymbol{w}} \Big|_{\boldsymbol{w}=\boldsymbol{w}_t}. \tag{4}$$

Proof is in appendix. Lemma 1 is also discussed in Hoffer et al. (2018); van Laarhoven (2017); Li & Arora (2020); Li et al. (2020), it makes the learning dynamics of normalized neural network exhibit an interesting phenomenon when using Stochastic Gradient Descent (SGD) with Weight Decay (WD): a typical SGD update rule with WD is

$$\boldsymbol{w}_{t+1} = \boldsymbol{w}_t - \eta(\frac{\partial \mathcal{L}}{\partial \boldsymbol{w}} \Big|_{\boldsymbol{w}=\boldsymbol{w}_t} + \lambda \boldsymbol{w}_t) = (1 - \eta\lambda)\boldsymbol{w}_t - \eta \frac{\partial \mathcal{L}}{\partial \boldsymbol{w}} \Big|_{\boldsymbol{w}=\boldsymbol{w}_t}, \tag{5}$$

where $\eta$ denotes learning rate, $\lambda$ denotes WD factor. Then dynamics of $\boldsymbol{w}_t$ is like a physical process – a satellite's motion around the earth (see illustration in Fig.1): according to Eq.(3), $-\eta\partial\mathcal{L}/\partial\boldsymbol{w}\big|_{\boldsymbol{w}=\boldsymbol{w}_t}$ (green line in Fig.1) is always perpendicular to $\boldsymbol{w}_t$, providing "centrifugal effect" to make $||\boldsymbol{w}_{t+1}||_2$ larger than $||\boldsymbol{w}_t||_2$; while $-\eta\lambda\boldsymbol{w}_t$ (red line in Fig.1) is always in the opposite direction of $\boldsymbol{w}_t$, providing "centripetal effect" to make $||\boldsymbol{w}_{t+1}||_2$ smaller than $||\boldsymbol{w}_t||_2$. Due to this "tug of war" between "centrifugal effect" and "centripetal effect", the norm of weight will change accordingly. Though the norm of scale-invariant weight doesn't influence the loss at all (by definition), but it will affect the optimizing trajectory by changing the scale of gradient (Eq.4). Therefore, to decouple the relation between weight norm and gradient norm, a common way (van Laarhoven, 2017; Hoffer et al., 2017; Chiley et al., 2019) is to regard the

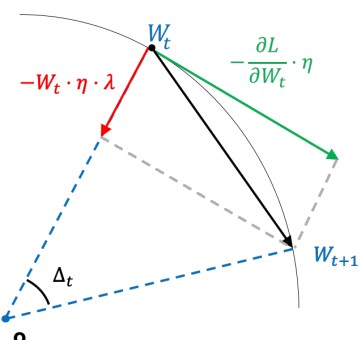

Figure 1: Illustration of optimization behavior with BN and WD. Angular update $\Delta_t$ represents the angle between the updated weight $\boldsymbol{w}_t$ and its former value $\boldsymbol{w}_{t+1}$.

learning dynamics of scale-invariant weight as a manifold learning process restricted on a unit sphere, using "effective learning rate", defined as $\eta/||\boldsymbol{w}_t||_2^2$. Akin to this reason, in this paper the learning dynamics of scale-invariant weight is viewed as a motion on a unit sphere, whose motion speed is determined by weight norm (in original weight space), gradient norm, and hyper-parameters. Therefore, we formally name it as *Spherical Motion Dynamcis (SMD)* and focus on discussing its unique properties.

**Concept of "Equilibrium"** Since "tug-of-war" in SMD determines the relative sizes of $||\boldsymbol{w}_{t+1}||_2$ and $||\boldsymbol{w}_t||_2$, a question naturally arises: what will happen if $||\boldsymbol{w}_{t+1}||_2 = ||\boldsymbol{w}_t||_2$? van Laarhoven (2017) discuss this question first; Chiley et al. (2019) named this state where $||\boldsymbol{w}_{t+1}||_2 = ||\boldsymbol{w}_t||_2$ in SMD as "*equilibrium*", and discuss its properties; Li & Arora (2020) derives a lemma about equilibrium in SGD with Momentum (SGDM); Li et al. (2020); Kunin et al. (2021) establish the

equilibrium in continuous approximation model. However, most early literature (van Laarhoven, 2017; Chiley et al., 2019; Li & Arora, 2020; Kunin et al., 2021) do not discuss a fundamental question: "Does equilibrium really occur in practice?" van Laarhoven (2017) intuitively explains that equilibrium is caused by convergence of optimization. But there exists a contradiction between the interpretation of van Laarhoven (2017) and traditional view of optimization: if equilibrium ($\|\boldsymbol{w}_t\|_2 = \|\boldsymbol{w}_{t+1}\|_2$) is caused by convergence of optimization, then gradient of loss $\partial\mathcal{L}/\partial\boldsymbol{w}\big|_{\boldsymbol{w}=\boldsymbol{w}_t}$ should be $\boldsymbol{0}$, which makes the balance of "centrifugal effect" and "centripetal effect" impossible to reach (since centripetal effect comes from $\partial\mathcal{L}/\partial\boldsymbol{w}\big|_{\boldsymbol{w}=\boldsymbol{w}_t}$). Therefore, van Laarhoven (2017); Chiley et al. (2019); Li & Arora (2020); Kunin et al. (2021) all essentially regard equilibrium as an assumption, and do not justify its existence in neither empirical nor theoretical aspects. "Equilibrium" was not even a phenomenon observed in practice, but only a concept until recently.

Recent work (Li et al., 2020) successfully exhibits the existence of equilibrium by formulating SGD in Eq.(5) via a Stochastic Differential Equation (SDE) in the continuous time limit. They theoretically prove equilibrium can be reached in SDE settings via convergence of $\|\boldsymbol{w}\|_t$, and the convergence time is $\mathcal{O}(1/(\lambda\eta))$ ($\lambda, \eta$ denote WD factor and learning rate respectively). However, due to the gap between discrete formulation of SGD and continuous formulation of SDE, theoretical results derived from SDE model can only provide intuitive understanding on empirical observations, some of which are even incorrect (will be discussed latter); Besides, SDE can hardly take SGD with momentum (Polyak, 1964) into account, which has become default setting in nearly all kinds of deep learning tasks. In summary, a thorough understanding on cause of "equilibrium" and its impact to learning dynamics of normalized neural network is still needed.

In this paper, we comprehensively reveal Spherical Motion Dynamics (SMD), i.e. the learning dynamics of normalized neural network using SGD(M) and weight decay (WD). Our analysis on SMD is directly established on discrete settings. We interpret why equilibrium can be reached in SMD in both theoretical and empirical aspects, and show how SMD affects the optimization trajectory of neural network. Specifically, our contributions are

- We introduce the assumptions which can lead to equilibrium in SMD, and justify their reasonableness by sufficient experiments. We also prove under given assumptions, equilibrium can be reached as weight norm approach to its theoretical value in a linear rate regime. Our theorem show equilibrium is a dynamic state in SMD, **convergence of weight norm is neither the precondition nor the consequence of equilibrium**. This conclusion almost refutes the assumptions of all previous related work (van Laarhoven, 2017; Chiley et al., 2019; Li & Arora, 2020; Kunin et al., 2021), and the conclusion of Li et al. (2020).

- We define a novel index, *angular update*, to measure the change of normalized neural network within a single iteration. We also derive its theoretical value in equilibrium. Our results show that angular update is better than norm of weight to indicate if equilibrium has been reached reached in SMD. Our empirical results further show angular update is an important index to reflect the effect of SMD and equilibrium;

- We verify our theorems on different computer vision tasks (including one of most challenging datasets ImageNet (Russakovsky et al., 2015) and MSCOCO (Lin et al., 2014)) with various networks structures. Experiments show the theoretical value of angular update and weight norm agree well with empirical observation. We also show how SMD influence the optimization trajectory of normalized neural network by controlling angular update.

We believe SMD is one of the key reason why learning dynamics of normalized neural network is not consistent with traditional optimization theory (Li et al., 2020). We think it is of great potential to take SMD and its equilibrium state into account while studying leaning dynamics of modern normalized neural network or designing novel efficient training strategy.

## 2 Theoretical results

In this section, we theoretically formulate Spherical Motion Dynamics (SMD) in discrete SGD/SGDM settings, and provide a precise description on "equilibrium" phenomenon; Then we prove equilibrium can be reached in SMD under specific assumptions; Finally a new index, whose theoretical value in equilibrium can be derived, is proposed to indicate the state of SMD.

A new definition need to be given at first. Eq.(4) implies though norm of scale-invariant weights does not affect the output of neural network, it can influence norm of gradients, thus we define *unit gradient* in order to decouple weight norm and gradient norm.

**Definition 2** (*Unit Gradient*)*. If $\boldsymbol{w}_t \neq \boldsymbol{0}$, $\tilde{\boldsymbol{w}} = \boldsymbol{w}/||\boldsymbol{w}||_2$, the unit gradient of $\partial\mathcal{L}/\partial\boldsymbol{w}|_{\boldsymbol{w}=\boldsymbol{w}_t}$ is $\partial\mathcal{L}/\partial\boldsymbol{w}|_{\boldsymbol{w}=\tilde{\boldsymbol{w}}_t}$.*

According to the definition of unit gradient, the unit gradient norm is independent of weight norm. Specifically, by setting $k$ as $1/||\boldsymbol{w}_t||_2$ in Eq.(4), the relation among weight norm, gradient and unit gradient is

$$\frac{\partial\mathcal{L}}{\partial\boldsymbol{w}}\Big|_{\boldsymbol{w}=\boldsymbol{w}_t} = \frac{1}{||\boldsymbol{w}_t||} \cdot \frac{\partial\mathcal{L}}{\partial\boldsymbol{w}}\Big|_{\boldsymbol{w}=\tilde{\boldsymbol{w}}_t}. \tag{6}$$

Now, we can depict equilibrium of SGD and SGDM in theorem 1, 2 respectively.

**Theorem 1.** *(**Equilibrium in SGD**) Assume the loss function is $\mathcal{L}(\boldsymbol{X};\boldsymbol{w})$ with scale-invariant weight $\boldsymbol{w}$, denote $\boldsymbol{g}_t = \frac{\partial\mathcal{L}}{\partial\boldsymbol{w}}\big|_{\boldsymbol{X}_t,\boldsymbol{w}_t}$, $\tilde{\boldsymbol{g}}_t = \boldsymbol{g}_t \cdot ||\boldsymbol{w}_t||_2$. Consider the update rule of SGD with weight decay,*

$$\boldsymbol{w}_{t+1} = \boldsymbol{w}_t - \eta \cdot (\boldsymbol{g}_t + \lambda\boldsymbol{w}_t) \tag{7}$$

*where $\lambda, \eta \in (0, 1)$. If the following assumptions hold:*

1) *$\lambda\eta \ll 1$ ($o(\lambda\eta)$ can be omitted);*

2) *Let $L_t = \mathbb{E}[||\tilde{\boldsymbol{g}}_t||_2^2|\boldsymbol{w}_t]$. $\exists V \in \mathbb{R}^+, \forall t \in \mathbb{N}^+, \mathbb{E}[(||\tilde{\boldsymbol{g}}_t||_2^2 - L_t)^2|\boldsymbol{w}_t] \leq V$;*

3) *$\forall t \in \mathbb{N}^+, L_t$ satisfies $|L_{t+1} - L_t| < 4\sqrt{V}(\lambda\eta)^{3/2}$;*

4) *$\exists l \in \mathbb{R}^+, \forall t \in \mathbb{N}^+, ||\tilde{\boldsymbol{g}}_t||_2^2 > l, l > 2[\frac{2\lambda\eta}{1-2\lambda\eta}]^2 L_t$.*

*Then $\exists B > 0, \forall t \in \mathbb{N}^+, w_t^* = \sqrt[4]{L_{t-1}\eta/(2\lambda)}$, we have*

$$\mathbb{E}[||\boldsymbol{w}_t||_2^2 - (w_t^*)^2]^2 \leq (1 - 2\lambda\eta)^t B + \frac{2V\eta^2}{l}. \tag{8}$$

**Remark 1.** *The theoretical value of weight norm $w_t^*$ in Theorem 1 is consistent with the magnitude of weight norm ($\mathcal{O}(\sqrt[4]{\eta/\lambda})$) in equilibrium in van Laarhoven (2017), though van Laarhoven (2017) assumes the equilibrium has been reached in advance, hence van Laarhoven (2017) cannot provide the approaching rate and scale of bias/variance. The vanishing term ($(1 - 2\lambda\eta)^t B$) in Eq.(8) is consistent with the mixing time $\mathcal{O}(1/(\lambda\eta))$ presented in Li et al. (2020).*

The proof can be seen in appendix. Assumption 1 is consistent with commonly used settings in practice (Goyal et al., 2017; He et al., 2017; Ma et al., 2018); Assumptions 2, 3, 4 all concern unit gradient: unit gradient norm should change smoothly (assumption 3) with bounded variance (assumption 2); besides, unit gradient norm should have a lower bound (assumption 4). We will see these assumptions can easily hold in practice in section 4.1.

**Remark 2.** *Note in theorem 1, we assume $||\tilde{\boldsymbol{g}}_t||_2^2$ has a uniform lowere bound $l$, while $\mathbb{E}[(||\tilde{\boldsymbol{g}}_t||_2^2 - L_t)^2|\boldsymbol{w}_t]$ has a uniform upper bound $V$. But these are too strong assumptions and may not hold in practice. We only use them for ease of demonstration and proof. In fact the lower bound of $||\tilde{\boldsymbol{g}}_t||_2^2$ and upper bound of $\mathbb{E}[(||\tilde{\boldsymbol{g}}_t||_2^2 - L_t)^2|\boldsymbol{w}_t]$ can be replaced with smoothly varying functions $l_t$ and $V_t$ respectively to remove unnecessary assumptions on uniformness. The "trend" to reach equilibirum is actually a local property, which can be inferred from the proof.*

**True meaning of "equilibrium": a dynamic state of SMD** Recall as we demonstrate in introduction, the concept of equilibrium is originally established on the assumption that weight norm is steady ($||\boldsymbol{w}_t||_2 = ||\boldsymbol{w}_{t+1}||_2$). But the assumption ($||\boldsymbol{w}_t||_2 = ||\boldsymbol{w}_{t+1}||_2$) is unrealistic due to the complex dynamics of training process and the variance of stochastic gradients. Now theorem 1 provides a realistic meaning of equilibrium in SGD settings: equilibrium is just a dynamic state of SMD, meaning $||\boldsymbol{w}_t||_2^2$ oscillates around the theoretical value $(w_t^*)^2$ determined by hyperparameters and unit gradient norm. Its variance is bounded by $2V\eta^2/l$, which is relatively small comparing with $(w_t^*)^4$ because

$$\frac{2V\eta^2}{l}/(w_t^*)^4 = \frac{4V\lambda\eta}{L_{t-1}l} = \mathcal{O}(\lambda\eta) \ll 1. \tag{9}$$

Aside form the stochastic behavior of $||\boldsymbol{w}_t||_2^2$, the "dynamic state" also reflects in the variation of the theoretical value $(w_t^*)^2$. Because $(w_t^*)^2$ is determined by $L_t$, which is allowed to change smoothly across the whole training process in assumption 2 (See more discussion in appendix). In summary, the sign of equilibrium is neither the convergence of weight norm $||\boldsymbol{w}_t||_2^2$ (van Laarhoven, 2017; Chiley et al., 2019) nor the convergence of $||\boldsymbol{w}_t||_2$ in expectation (Li et al., 2020). The real sign of equilibrium is whether $\mathbb{E}||\boldsymbol{w}_t||_2^2$ is close to its theoretical value $(w_t^*)^2$.

Theorem 1 also shows the dynamic equilibrium can be reached in a linear rate regime when vanishing term is larger than constant term in Eq.(8). The approaching rate is only determined by predefined parameters $\lambda, \eta$. Moreover, based on the proof of theorem 1, the cause of equilibrium is independent of optimization process at all, which implies the possibility that equilibrium can be reached long before the convergence of loss function, which refutes the conjections that "equilibirum" a equilibirum distribution of function space.

Now we extend theorem 1 to momentum case. SGDM is more complex than SGD since momentum is not always perpendicular to the weight, hence we need to modify assumptions.

**Theorem 2.** *(Equilibrium in SGDM) Considering the update rule of SGDM (heavy ball method (Polyak, 1964)):*

$$\boldsymbol{v}_t = \alpha \boldsymbol{v}_{t-1} + \boldsymbol{g}_t + \lambda \boldsymbol{w}_t \tag{10}$$

$$\boldsymbol{w}_{t+1} = \boldsymbol{w}_t - \eta \boldsymbol{v}_t \tag{11}$$

*where $\lambda, \eta \in (0, 1), \alpha \in (\frac{1}{2}, 1)$. If following assumptions hold:*

5) *$\lambda\eta \ll 1$, $\lambda\eta < (1 - \sqrt{\alpha})^2$;*

6) *Define $h_t = ||\boldsymbol{g}_t||_2^2 + 2\alpha\langle \boldsymbol{v}_{t-1}, \boldsymbol{g}_t \rangle$, $\tilde{h}_t = h_t \cdot ||\boldsymbol{w}_t||_2^2$, $L_t = \mathbb{E}[\tilde{h}_t|\boldsymbol{w}_t]$. $\exists V \in \mathbb{R}^+, \forall t \in \mathbb{N}^+$, $\mathbb{E}[(\tilde{h}_t - L_t)^2|\boldsymbol{w}_t] \le V$;*

7) *$\forall t \in \mathbb{N}^+$, $L_t$ satisfies $|L_{t+1} - L_t| < 4\sqrt{V}(\lambda\eta)^{3/2}$;*

8) *$\exists l \in \mathbb{R}^+, \forall t \in \mathbb{N}^+, \tilde{h}_t > l > 2[\frac{6\lambda\eta}{(1-\alpha)^3(1+\alpha)-8\lambda\eta(1-\alpha)}]^2 L_t$;*

*then $\exists B, C > 0$, $C$ only depends on $\alpha$, $w_t^* = \sqrt[4]{L_{t-1}\eta/(\lambda(1-\alpha)(2 - \lambda\eta/(1+\alpha)))}$, we have*

$$\mathbb{E}[||\boldsymbol{w}_t||_2^2 - (w_t^*)^2]^2 \le (1 - \frac{2\lambda\eta}{1-\alpha})^t B + \frac{V\eta^2}{l}C, \tag{12}$$

**Remark 3.** *So far, no other work rigorously prove equilibrium can be reached in SGDM. The most relevant work (Li et al., 2020) only provides a conjecture on convergence rate of weight norm in SGDM. By regarding SGDM as SGD with larger learning rate, they guess that the mixing time to reach equilibrium in SGDM case should be $\mathcal{O}(1/(\lambda\eta))$, same order as mixing time in SGD case. Their conjecture cannot provide further insight on difference between SGD and SGDM. While our results (vanishing terms in Eq.(8), (12) respectively) clearly reflect the difference: the approaching rate of SGDM should be $1/(1 - \alpha)$ times larger than rate of SGD with same $\eta\lambda$. $\alpha$ is usually set as 0.9 in practice, hence SGDM can reach equilibrium condition much faster than SGD.*

Proof can be seen in appendix. Like assumption 1, assumption 5 also holds for commonly used hyperparameter settings; Assumption 6, 7, 8 concerns not unit gradient norm $||\tilde{\boldsymbol{g}}_t||_2^2$ but an adjusted value $\tilde{h}_t$ which dominates the expectation and variance of $||\boldsymbol{w}_t||_2^2$. We empirically find the expectation of $\langle \boldsymbol{v}_{t-1}, g_t \rangle$ is very close to 0, therefore the behavior of $\tilde{h}_t$ is similar to that of $||\tilde{\boldsymbol{g}}_t||_2^2$ (see Figure 3(d)). We leave theoretical analysis on $\tilde{h}_t$ as future work. The experiments on justification of assumptions 6, 7, 8 can be seen in Figure 3. Comparing with Eq.(8) and Eq.(12), we can infer with same $\eta, \lambda$, SGDM can reach equilibrium state much faster than SGD, but it may have a larger variance, our experiments also verify our claim (see Figure 3(b), 3(e)).

We have derived the theoretical value of weight norm in equilibrium, it allows us to check if equilibrium has been reached in practice. But the theoretical value of weight norm still relies on the expectation of unit gradient norm, which is not easy to compute in practice. Besides, weight norm is of little value for studying normalized models since their weight is scale-invariant. Hence we introduce an new and meaningful index, *angular update*, to reflect the effect of SMD and equilibrium.

**Definition 3** (*Angular Update*). *Let $\boldsymbol{w}_t$ denote a scale-invariant weight from a neural network at iteration $t$, then **angular update** $\Delta_t$ is defined as*

$$\Delta_t = \measuredangle(\boldsymbol{w}_t, \boldsymbol{w}_{t+1}) = arccos\left(\frac{\langle \boldsymbol{w}_t, \boldsymbol{w}_{t+1}\rangle}{||\boldsymbol{w}_t|| \cdot ||\boldsymbol{w}_{t+1}||}\right), \tag{13}$$

*where $\measuredangle(\cdot, \cdot)$ denotes the angle between two vectors, $\langle \cdot, \cdot \rangle$ denotes the inner product.*

Angular update has a concrete geometric meaning (see illustration in Figure 1): it is exactly the geodesic distance between $\tilde{\boldsymbol{w}}_t$ and $\tilde{\boldsymbol{w}}_{t+1}$ on $\mathcal{S}^{p-1}$, where $\tilde{\boldsymbol{w}}_t = \boldsymbol{w}_t/||\boldsymbol{w}_t||_2$, $\tilde{\boldsymbol{w}}_{t+1} = \boldsymbol{w}_{t+1}/||\boldsymbol{w}_{t+1}||_2$. Comparing with the Euclidean distance $||\boldsymbol{w}_{t+1} - \boldsymbol{w}_t||_2$, angular update $\Delta_t$ better reflects the effective update of the scale-invariant weight $\boldsymbol{w}_t$ on its intrinsic domain $\mathcal{S}^{p-1}$. Angular update is determined by the relative sizes of gradient norm and weight norm, while the relative sizes of gradient/weight norm are influenced by SMD, hence angular update is strongly affected by SMD. The following theorem exhibits the behavior of angular update when equilibrium is reached in SMD.

**Theorem 3.** *(**Theoretical value of Angular Update**) In SGD(SGDM) case, if assumptions in theorem 1(2) hold, $\eta^2 \ll 1$, $t$ is sufficiently large so that vanishing terms in Eq.(8), (12) can be omitted, then with probability at least $1 - \sqrt[3]{\frac{V}{L_t l}}$ we have*

$$|\Delta_t - \sqrt{\frac{2\lambda\eta}{1+\alpha}}| < \mathcal{O}(\sqrt[3]{\frac{V}{L_t l}}). \tag{14}$$

*In SGD case, $\alpha = 0$.*

**Remark 4.** *Results of SGD and SGDM case are summarized in Eq.(14) in order to highlight the connection between SGD and SGDM. Theoretical value of angular update in Theorem 3 is partially consistent with previous works (Chiley et al., 2019; Li & Arora, 2020; Li et al., 2020; Kunin et al., 2021), detailed discussion is buried in appendix. Note bias term in right side of Eq.(14) is of $\mathcal{O}(\sqrt[3]{V/L_t l})$, which is too large comparing with its empirical value (see Figure 3(c), 3(f)), we leave it as a future work to improve the bound in Eq.(14).*

Proof is in appendix. According to theorem 3, the theoretical value of angular update in equilibrium only depends on hyper-parameters: learning rate $\eta$, WD factor $\lambda$, and momentum factor $\alpha$. Hence comparing with behavior of weight norm, angular update provides an easier way to check whether equilibrium is reached. Since equilibrium can be reached in a linear rate regime as theorem 1, 2 demonstrate, theorem 3 implies update efficiency of scale-invariant weights within a single step eventually will be determined only by predefined hyperparameters, regardless other attributes of the weights (shape, size, position in network structure, or effects from other weights).

## 3 The Role of Angular Update in Sphereical Motion Dynamics

Far beyond just an index indicating whether equilibrium is reached, angular update is also a important way by which SMD can affect the optimization process of normalized neural network. In this section, we will provide an intuitive explanation on the role of angular update in folklore view of landscape exploration.

The optimization process of gradient descent method is like an expedition to find the bottom of a basin in loss landscape (i.e. a local minimum). It is well known that gradient descent with constant learning rate can always converge to the minimum in strong convex problem as the gradient norm will converge to zero (Boyd et al., 2004); while SGD with constant learning rate can not converge to the minimum certainly, because noise of gradients can make trajectory "escape from local minimum" Zhu et al. (2018).

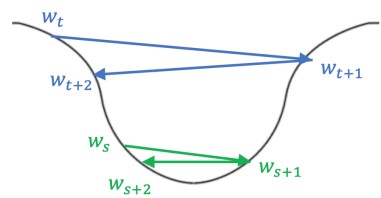

Figure 2: Illustration of local minimum and angular updates. Blue arrow indicates a single step with larger angular update, while green arrow indicates a single step with smaller angular update.

However, the behavior of SGD(M) in equilibrium is unique: due to the scale invariant property, the loss landscape is defined on a unit sphere $\mathcal{S}^{p-1}$, so does the basin (local mimium). Then based on the geometrical meaning of angular update, angular update denotes a "single step length" on the loss landscape. According to theorem 3, when equilibrium state has been reached in SMD, angular update is approximated equal to $\sqrt{2\lambda\eta/(1+\alpha)}$. Hence with constant $\eta$, $\lambda$, and $\alpha$, the optimization trajectory is "stuck" at the top of the basin, unable to reach the bottom (converge/reduce loss), just like the illustration in Figure 2. The value of angular update in equilibrium is totally determined by hyper-parameters, there is no way the optimization trajectory can move to the bottom of basin in equilibrium due to overly large single step length. Based on the analysis above, we have following conjectures: if the trajectory stays in the same basin with different values of angular update in equilibirum, and momentum factor is fixed, then

1. Larger angular update leads to higher training loss and lower test accuracy in equilibrium;

2. Even with different learning rate and weight decay, same angular update leads to similar training loss/test accuracy in equilibrium;

We will verify our conjectures in following empirical study. Note the influence of momentum is different from learning rate and weight decay: even momentum can decrease the value of angular update as theorem 3 implies, it still can increase the oscillating range of the trajectory in the basin, resulting in larger training/testing loss.

## 4   Experiments

In this section, we verify our theorems and conjectures on SMD and equilibrium by empirical study. We show the equilibrium phenomenon described in our theorems really occurs in various computer vision tasks including ImageNet (Russakovsky et al., 2015) and MSCOCO (Lin et al., 2014). We also analyze an interesting phenomenon as an example to show how SMD can affect training process in a way different from traditional view on optimization of neural network.

### 4.1   Fixed learning rate

First we verify our theorems on ImageNet (Russakovsky et al., 2015) with fixed learning rate. We use Resnet50 (He et al., 2016) as the baseline model. In SGDM case, the momentum factor is $\alpha = 0.9$. Form Figure 3(b), 3(e), 3(c), 3(f), we can see empirical value of $||\boldsymbol{w}_t||_2^2$ and $\Delta_t$ differ from their theoretical value respectively at very beginning, because the initialized value of weight norm is handcrafted, far away from the theoretical value in equilibrium. After several iterations, empirical values of weight norm and angular update agree with their theoretical values very well, which implies equilibrium has been reached. We also observe SGDM can achieve equilibrium much faster than SGD. According to Eq.(8), (12), the underlying reason might be with same learning rate $\eta$ and WD factor $\lambda$, approaching rate of SGDM ($\frac{\lambda\eta}{1-\alpha}$) is larger than approaching rate of SGD ($\lambda\eta$). Results in Figure 3 also prove our claim that equilibrium is a dynamical state: even equilibrium has been reached, $\mathbb{E}||\tilde{\boldsymbol{g}}||_2^2$ ($\mathbb{E}\tilde{h}_t$) constantly increase, $||\boldsymbol{w}_t||_2^2$ increases accordingly, $||\boldsymbol{w}_t||_2^2$ and $\Delta_t$ always oscillate around their theoretical values respectively, showing equilibrium state maintains in SMD.

Then we train Resnet50 on Imagenet with different hyperparameters to verify our conejctures. The results are shown in Figure 4. We can see the experimental results strongly support our conjectures: in SGD case, whatever learning rate $\eta$ and WD factor $\lambda$ is, they have similar angular update in equilibrium, as well as similar training/testing curves; when angular update is larger, both training loss and test error are larger. These phenomena are even more obvious in SGDM case, when momentum factor $\alpha$ is fixed. Besides, the role of momentum in SMD is exactly as we speculate: it can decrease angular update while increase training/testing loss.

### 4.2   Multi-stage learning rate

Now we study the behavior of angular update with SGDM and multi-stage learning rate schedule on Imagenet (Russakovsky et al., 2015) and MSCOCO (Lin et al., 2014). In ImageNet classification task, we still use Resnet50 as baseline model. The training settings rigorously follow Goyal et al. (2017); In MSCOCO experiment, we conduct experiments on Mask-RCNN (He et al., 2017) benchmark

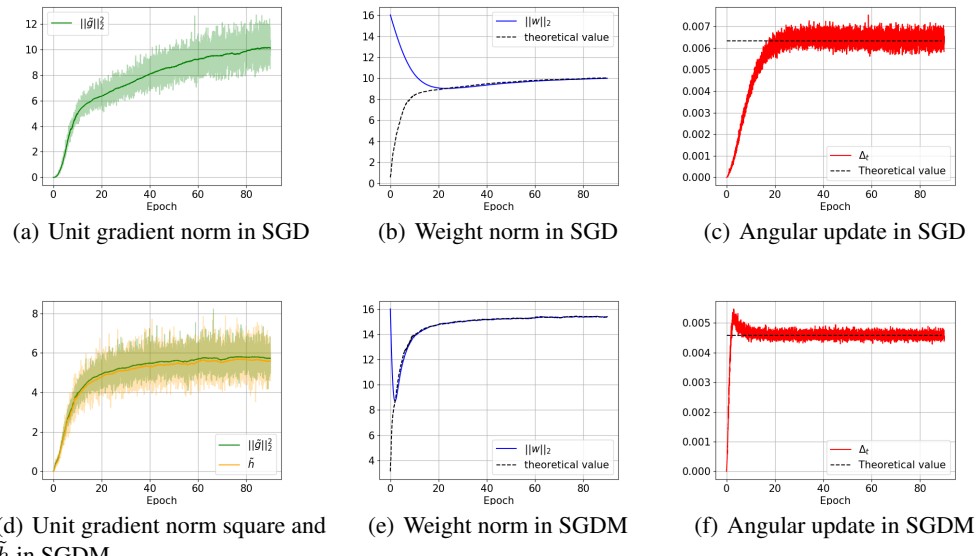

Figure 3: Performance of *layer.2.0.conv2* from Resnet50 in SGD and SGDM, respectively. In (a), (d), semitransparent line represents the raw value of $||\tilde{g}_t||_2^2$ or $\tilde{h}_t$, while solid line represents the averaged value within consecutive 200 iterations to estimate the expectation of $||\tilde{g}_t||_2^2$ or $\tilde{h}_t$ conditioning on $t$; In (b), (e), blue solid lines represents the raw value of weight norm $||w_t||_2$, while dashed line represent the theoretical value of weight norm computed in Theorem 1, 2 respectively. To compute the theoretical value of weight norm, we use the estimated $\mathbb{E}||\tilde{g}||_2^2$ and $\mathbb{E}\tilde{h}$ (solid lines) in (a) and (d) respectively; In (c), (f), red lines represent raw value of angular update during training, dashed lines represent the theoretical value of angular update computed by $\sqrt{2\lambda\eta}$ and $\sqrt{2\lambda\eta/(1+\alpha)}$ respectively.

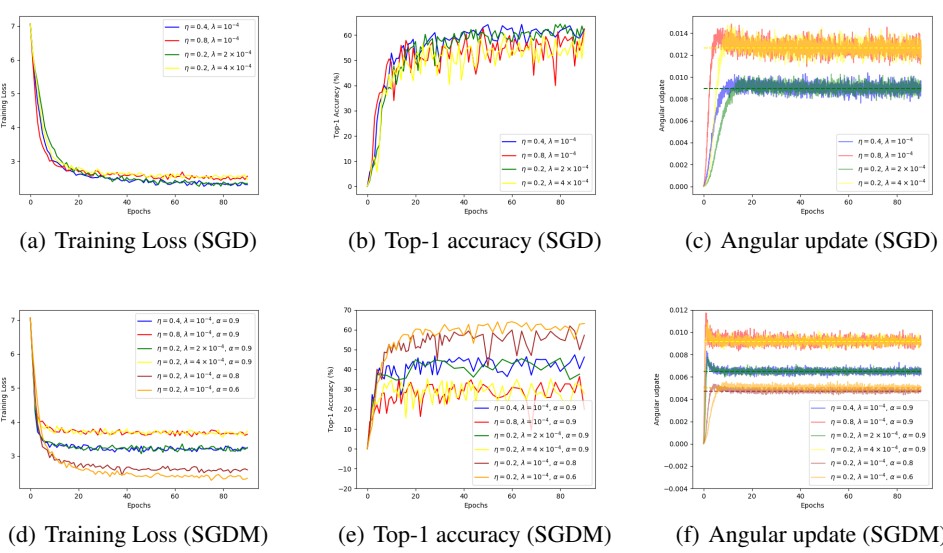

Figure 4: Resnet50 are trained on Imagenet with different hyper-parameters. The angular update is computed by the weight in *LAYER.2.0.CONV2* of Resnet50. Dashed lines in (c), (f) represent the theoretical value of angular update by $\sqrt{\frac{2\lambda\eta}{1+\alpha}}$.

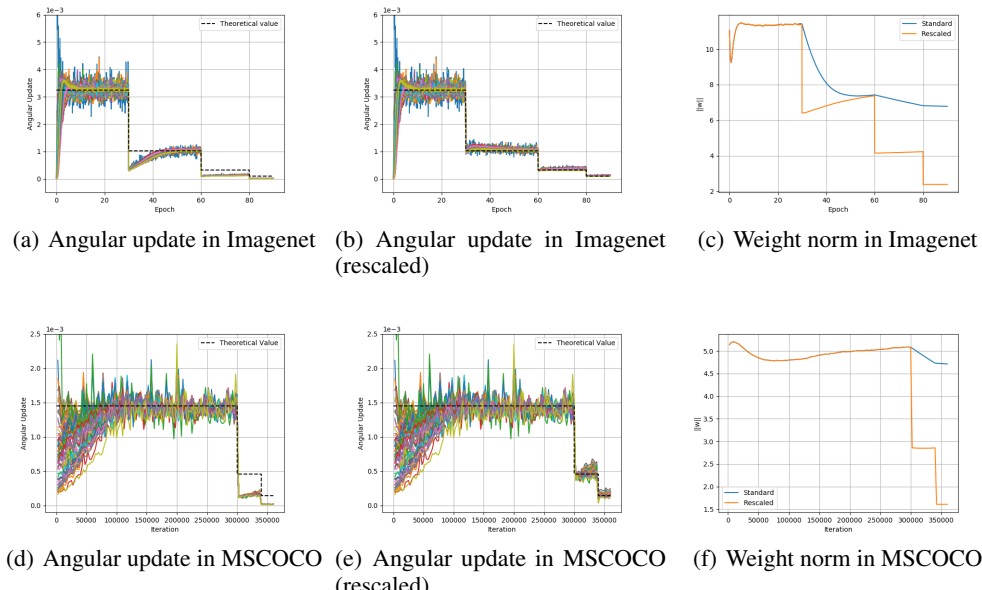

(a) Angular update in Imagenet

(b) Angular update in Imagenet (rescaled)

(c) Weight norm in Imagenet

(d) Angular update in MSCOCO

(e) Angular update in MSCOCO (rescaled)

(f) Weight norm in MSCOCO

Figure 5: In (a),(b),(d),(e), solid lines with different colors represent raw value of angular update from all convolution layers; In (a), (d), training setting rigorously follows Goyal et al. (2017); He et al. (2019) respectively; In (b), (e), weight norm is divided by $\sqrt[4]{10}$ as long as learning rate is divided by $10$; In (c), (f), weight norm is computed on *layer.1.0.conv2* in Resnet50 backbone. Blue line represent original settings, orange lines represent rescaled settings.

using a Feature Pyramid Network (FPN) (Lin et al., 2017), ResNet50 backbone and SyncBN (Peng et al., 2018) following the 4x setting in He et al. (2019).

There appears to be a mismatch between theorems and empirical observations in Figure 5(a), 5(d): angular update $\Delta_t$ in the last two learning rate stages is smaller than its theoretical value. This mismatch can be well interpreted by our theory: according to Theorem 1, 2, when equilibrium state is reached, theoretical value of weight norm $||\boldsymbol{w}_t||_2$ satisfies $||\boldsymbol{w}_t||_2 \propto \sqrt[4]{\frac{\eta}{\lambda}}$. However, when learning rate is divided by $k$, equilibrium state is broken, theoretical value of weight norm $||\boldsymbol{w}_t||_2$ in the new equilibrium state is $\sqrt[4]{1/k}$ times smaller. But new equilibrium cannot be reached immediately (see Figure 5(c), 5(f)), following corollary gives the least number of iterations to reach new equilibrium.

**Corollary 3.1.** *In SGD case with learning rate $\eta$, WD factor $\lambda$, if learning rate is divided by $k$, and unit gradient norm remains unchanged, then at least $\lceil \log(k)/(2\lambda\eta) \rceil$ iterations are required to reach the new equilibrium state; In SGDM case with momentum coefficient $\alpha$, then at least $\lceil [\log(k)(1-\alpha)]/(2\lambda\eta) \rceil$ iterations are required to reach the new equilibrium state.*

Corollary 3.1 implies SGD/SGDM with smaller learning rate requires more iterations to reach new equilibrium state. Hence, in second learning rate stage in Imagenet experiments, angular update $\Delta_t$ can reach its new theoretical value within 15 epochs. But in last two learning rate stages of Imagenet/MSCOCO experiments, SGDM cannot completely reach new equilibrium by the end of training. As a result, we observe empirical value of $\Delta_t$ is smaller than its theoretical value. Based on our theorem, we can bridge the gap by skipping the intermediate process between old equilibrium and new one. Specifically, when learning rate is divided by $k$, norm of scale-invariant weight is also divided by $\sqrt[4]{k}$, SGDM can reach new equilibrium immediately in new learning rate stage. Experiments((b),(e) in Figure 5) show this simple strategy can make angular update $\Delta_t$ always close to its theoretical value across the whole training process though learning rate changes.

## 4.3 "Pseduo overfitting" caused by learning rate decay

"Dropping test preformance(accuracy)" phenomenon is often explained as a result of "overfitting" issue, which refers to the phenomenon where a trained model can fit training data very well, but fails to fit additional data for validation or prediction. To handle such issues, regularization methods

like early-stopping (Prechelt, 1998) and dropout (Srivastava et al., 2014) are proposed. But here we show in some cases "dropping test preformance" phenomena is just a "pseduo" overfitting, caused by learning rate decay.

Let's see the following experiment in which Resnet18 (He et al., 2016) is trained on CI-FAR10 (Krizhevsky & Geoffrey, 2009) with standard settings (see experiment settings in appendix) Figure 6 presents the training curves of the experiments. It can be seen from figure 6(b)(blue line inside red ellipse) that with standard implementation of SGDM, test accuracy severely drops during the 2nd learning rate stage. It seems a typical overfitting issue. However, two other phenomena cannot be well interpreted by overfitting: 1) test accuracy *only* severely decreases during 2nd learning rate stage, it performs normally in the 3rd, 4th learning rate stages (blue lines in Figure6(b)); 2) When other optimization method like Adam (Kingma & Ba, 2015) is used, test accuracy does not drop apparently in any learning rate stages (green line in Figure6(b)).

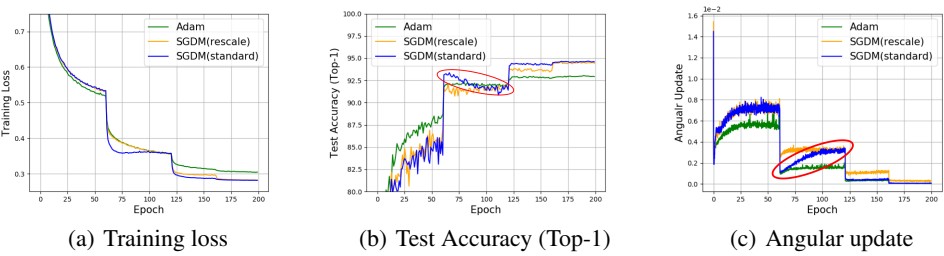

| (a) Training loss | (b) Test Accuracy (Top-1) | (c) Angular update |
|---|---|---|

Figure 6: Training curves of Resnet18 trained on CIFAR10 (averaged across 5 seeds). Angular update is from layer1.0.conv1 of Resnet18.

Now SMD can provide a more reasonable interpretation: we have demonstrated the conjectures on the relationship between angular update and performance of model (Sec. 3), and the fact that learning rate decay can make angular update increase (Sec. 4.2) above, so it's reasonable to speculate that learing rate decay is the cause of the temporary "dropping test accuracy" phenomenon. See illustration in Figure 7: after shrinking learning rate, smaller angular update allows the optimizaton trajectory to move towards the bottom of the basin, then angular

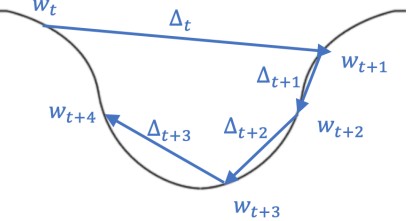

Figure 7: Increasing angular caused by learning rate decay.

update will increase, forcing trajectory to "escape" from the bottom, resulting in dorpping test performance. If this interpretation is ture, then rescaling strategy introduced in Sec. 4.2 should avoid "pseduo dropping test performance" by eliminating increasing angular update phenomenon. Further experiments support our speculation (Figure 6). They also explain why dropping test performance cannot be seen in Adam: we track the angular update with Adam, and no increasing angular update phenomenon is observed. Please see more discussion in appendix.

## 5  Conclusion

In this paper, we comprehensively reveal the learning dynamics of normalized neural network with SGD/SGDM and weight decay (WD), named as Spherical Motion Dynamics (SMD). With mild assumptions, we strictly prove SMD will reach equilibrium state in a linear regime. We also propose a novel index, angular update, to depict the state of SMD, and derive its theoretical property in equilibrium. Most importantly, we show our theorem is widely valid, they can be verified on challenging computer vision tasks, beyond synthetic datasets. Besides, we show SMD can dramatically effect the optimization of neural network by controlling angular update in practice. We believe our results on SMD make an important step to understand the mechanism of deep neural networks, and can inspire new deep learning techniques.

## Acknowledgments and Disclosure of Funding

This work is supported by The National Key Research and Development Program of China (No. 2017YFA0700800) and Beijing Academy of Artificial Intelligence (BAAI). We really appreciate the valuable suggestions on proof of the theorems from Dr. Donghao Wang and Dr. Siyi Yang.

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
