# Appendix: Spherical Motion Dynamics

**Ruosi Wan**
Megvii Technology
wanruosi@megvii.com

**Zhanxing Zhu**
University of Edinburgh
zhanxing.zhu@ed.ac.uk
zhanxing.zhu@pku.edu.cn

**Xiangyu Zhang**
Megvii Technology
zhangXiangyu@megvii.com

**Jian Sun**
Megvii Technology
sunjian@megvii.com

## A  Intuitive Description on Spherical Motion Dynamics

In introduction of the main text, we only provide a brief statement on the concept of Spherical Motion Dynamics (SMD) due to the limitation of the length. In this section we give a detailed description on intuition of Spherical Motion Dynamics and its relevant concept, effective learning rate.

**Effective learning rate**   Since Batch Normalization (Ioffe & Szegedy, 2015) becomes an indispensable module of popular network structures, the scale of the weight norm does not affect the output of unit at all, Euclidean distance defined in weight space completely fails to measure the evolving of DNN during learning process. As a result, learning rate $\eta$ cannot properly measure update efficiency of normalized DNN. To deal with such issue, van Laarhoven (2017); Hoffer et al. (2018); Zhang et al. (2019b) propose "effective learning rate" as a substitute for learning rate to measure update efficiency of normalized neural network using stochastic gradient descent (SGD), defined as

$$\eta_{eff} = \frac{\eta}{||\boldsymbol{w}||_2^2}. \tag{1}$$

Now we show why effective learning rate is defined like Eq.(1). A typical SGD update rule without weight decay(WD) is

$$\boldsymbol{w}_{t+1} = \boldsymbol{w}_t - \eta \frac{\partial \mathcal{L}}{\partial \boldsymbol{w}}\Big|_{\boldsymbol{w}=\boldsymbol{w}_t}, \tag{2}$$

if $||\boldsymbol{w}_t||_2 = ||\boldsymbol{w}_{t+1}||_2$, then dividing both side of Eq.(7) by $||\boldsymbol{w}_t||_2$, combining with the definition of unit gradient, and let $\tilde{\boldsymbol{w}}_t = \boldsymbol{w}_t/||\boldsymbol{w}_t||_2$, we have

$$
\begin{aligned}
\tilde{\boldsymbol{w}}_{t+1} &= \tilde{\boldsymbol{w}}_t - \frac{\eta}{||\boldsymbol{w}_t||_2^2} \frac{\partial \mathcal{L}}{\partial \boldsymbol{w}}\Big|_{\boldsymbol{w}=\tilde{\boldsymbol{w}}_t} \\
&= \tilde{\boldsymbol{w}}_t - \eta_{eff} \cdot \frac{\partial \mathcal{L}}{\partial \boldsymbol{w}}\Big|_{\boldsymbol{w}=\tilde{\boldsymbol{w}}_t}.
\end{aligned}
\tag{3}
$$

Eq.(3) shows effective learning rate can be viewed as learning rate of SGD defined on the intrinsic domain $\mathcal{S}^{p-1}$.

Next we show the connection between effective learning rate and angular update. According to the definition of angular update, $\partial \mathcal{L}/\partial \boldsymbol{w}$ is perpendicular to weight $\boldsymbol{w}$ (see Figure 1 in main text), combining with Eq.(3) and (1) we have

$$\tan(\Delta_t) = \frac{\eta}{||\boldsymbol{w}_t||} \cdot ||\frac{\partial \mathcal{L}}{\partial \boldsymbol{w}}\Big|_{\boldsymbol{w}=\boldsymbol{w}_t}||_2 = \eta_{eff} \cdot ||\frac{\partial \mathcal{L}}{\partial \boldsymbol{w}}\Big|_{\boldsymbol{w}=\tilde{\boldsymbol{w}}_t}||_2. \tag{4}$$

35th Conference on Neural Information Processing Systems (NeurIPS 2021).

If angular update $\Delta_t$ is small enough, it can be approximated by first order Taylor series expansion of $\tan(\Delta_t) = \Delta_t + \mathcal{O}(\Delta_t^2)$, therefore we have

$$\Delta_t \approx \tan(\Delta_t) = \eta_{eff} \cdot ||\frac{\partial \mathcal{L}}{\partial \boldsymbol{w}}\Big|_{\boldsymbol{w}=\tilde{\boldsymbol{w}}_t}||_2. \tag{5}$$

It can be seen from Eq.(5) effective learning rate multiplying unit gradient norm equals angular update. Comparing with original definition of learning rate $\eta$, effective learning rate is not deterministic, its value depends on the variation of weight norm. Even in equilibrium state of SMD where weight norm has its own theoretical value that can be explicitly computed (theorem 1 in main text), the theoretical value of effective learning rate still depends on unit gradient norm. On the hand, our results in main text showing the theoretical value of angular update in equilibrium is only determined by predefined hyper parameters (learning rate $\eta$, WD factor $\lambda$). Therefore, we suggest to directly study the behavior of angular update instead of effective learning in SMD.

**Necessity of Weight Decay**    Another deduction from Eq.(2) is that weight norm always increases because

$$||\boldsymbol{w}_{t+1}||_2^2 = ||\boldsymbol{w}_t||_2^2 + (\eta||\frac{\partial \mathcal{L}}{\partial \boldsymbol{w}}\Big|_{\boldsymbol{w}=\boldsymbol{w}_t}||_2)^2 > ||\boldsymbol{w}_t||_2^2. \tag{6}$$

Recall lemma 1 in main text implies gradient norm is inversely proportional to weight norm if unit gradient is fixed, hence increasing weight norm can shrink gradient norm as well as effective learning rate in Eq.(1). Zhang et al. (2019b) states the potential risk that GD/SGD without WD but BN will converge to a stationary point (where gradient norm is very small) not by reducing loss but by increasing norm of weight. Arora et al. (2019) proves that full gradient descent can avoid the risk of vanishing effective learning rate, and converge to a stationary point defined on $\mathcal{S}^{p-1}$, but their results still require sophisticated learning rate decay schedule in SGD case. Besides, practical implementation suggests trained neural networks without WD often suffer from poor generalization (Zhang et al., 2019b; Bengio & LeCun, 2007; Lewkowycz & Gur-Ari, 2020).

**Tug-of-war in Spherical Motion Dynamics**    Now we intuitively derive the tug-of-war between the centripetal effect from WD, and the centrifugal effect from the gradient perpendicular to weights. Considering the update rule of SGD with WD:

$$\boldsymbol{w}_{t+1} = \boldsymbol{w}_t - \eta(\frac{\partial \mathcal{L}}{\partial \boldsymbol{w}}\Big|_{\boldsymbol{w}=\boldsymbol{w}_t} + \lambda \boldsymbol{w}_t). \tag{7}$$

The squared norm of updated weight is

$$||\boldsymbol{w}_{t+1}||_2^2 = (1 - \lambda\eta)^2 ||\boldsymbol{w}_t||_2^2 + (\eta||\frac{\partial \mathcal{L}}{\partial \boldsymbol{w}}\Big|_{\boldsymbol{w}=\boldsymbol{w}_t}||_2)^2 \tag{8}$$

Comparing with Eq.(6), Eq.(8) implies WD provides direction of updates tending to reduce weight norm, hence Chiley et al. (2019); Zhang et al. (2019b) point out the possibility that weight norm can be steady, but do not explain this clearly. Here we demonstrate the mechanism deeper: By Eq.(8), we have

$$
\begin{aligned}
&||\boldsymbol{w}_{t+1}||_2 - ||\boldsymbol{w}_t||_2 \\
=&\sqrt{(1-\lambda\eta)^2||\boldsymbol{w}_t||_2^2 + \eta^2||\frac{\partial \mathcal{L}}{\partial \boldsymbol{w}}\Big|_{\boldsymbol{w}=\boldsymbol{w}_t}||_2^2} - ||\boldsymbol{w}_t||_2 \\
=&\frac{(1-\lambda\eta)^2||\boldsymbol{w}_t||_2^2 + \eta^2||\frac{\partial \mathcal{L}}{\partial \boldsymbol{w}}\Big|_{\boldsymbol{w}=\boldsymbol{w}_t}||_2^2 - ||\boldsymbol{w}_t||_2^2}{\sqrt{(1-\lambda\eta)^2||\boldsymbol{w}_t||_2^2 + \eta^2||\frac{\partial \mathcal{L}}{\partial \boldsymbol{w}}\Big|_{\boldsymbol{w}=\boldsymbol{w}_t}||_2^2} + ||\boldsymbol{w}_t||_2}.
\end{aligned}
\tag{9}
$$

When $\eta\lambda \ll 1$, we have

$$
\begin{aligned}
&(1-\lambda\eta)^2||\boldsymbol{w}_t||_2^2 + \eta^2||\frac{\partial \mathcal{L}}{\partial \boldsymbol{w}}\Big|_{\boldsymbol{w}=\boldsymbol{w}_t}||_2^2 - ||\boldsymbol{w}_t||_2^2 \\
=&-2\lambda\eta||\boldsymbol{w}_t||_2^2 + \eta^2||\frac{\partial \mathcal{L}}{\partial \boldsymbol{w}}\Big|_{\boldsymbol{w}=\boldsymbol{w}_t}||_2^2 + \mathcal{O}(\lambda\eta)
\end{aligned}
\tag{10}
$$

$$\sqrt{(1-\lambda\eta)^2||\boldsymbol{w}_t||_2^2 + \eta^2||\frac{\partial \mathcal{L}}{\partial \boldsymbol{w}}\Big|_{\boldsymbol{w}=\boldsymbol{w}_t}||_2^2} + ||\boldsymbol{w}_t||_2 = 2||\boldsymbol{w}_t||_2 + \mathcal{O}(\lambda\eta). \tag{11}$$

Therefore, we have

$$||\boldsymbol{w}_{t+1}||_2 - ||\boldsymbol{w}_t||_2$$

$$= -\lambda\eta||\boldsymbol{w}_t||_2 + \frac{\eta^2}{2||\boldsymbol{w}_t||_2}||\frac{\partial\mathcal{L}}{\partial\boldsymbol{w}}\Big|_{\boldsymbol{w}=\boldsymbol{w}_t}||_2^2 + \mathcal{O}(\lambda\eta) \quad (12)$$

$$\approx -\lambda\eta||\boldsymbol{w}_t||_2 + \frac{\eta^2}{2||\boldsymbol{w}_t||_2^3}||\frac{\partial\mathcal{L}}{\partial\boldsymbol{w}}\Big|_{\boldsymbol{w}=\tilde{\boldsymbol{w}}_t}||_2^2.$$

Eq.(12) implies if unit gradient norm ($||\frac{\partial\mathcal{L}}{\partial\boldsymbol{w}}\Big|_{\boldsymbol{w}=\tilde{\boldsymbol{w}}_t}||_2$) is steady, "centripetal force" ($-\lambda\eta||\boldsymbol{w}_t||_2$) is proportional to $||\boldsymbol{w}_t||_2$, while "centrifugal force" ($\frac{\eta^2}{2||\boldsymbol{w}_t||_2^3}||\frac{\partial\mathcal{L}}{\partial\boldsymbol{w}}\Big|_{\boldsymbol{w}=\tilde{\boldsymbol{w}}_t}||_2^2$) is inversely proportional to $||\boldsymbol{w}_t||_2^3$. As a result, the dynamics of weight norm is like a spherical motion in physics: overly large weight norm makes centripetal force larger than centrifugal force, leading to decreasing weight norm; while too small weight norm makes centripetal force smaller than centrifugal force, resulting in increasing weight norm. At last, equilibrium state will be reached if the number of iterations is sufficiently large.

Let us see what will happen if equilibrium is reached in this tug-of-war. By Eq.(12), we have

$$\lambda\eta||\boldsymbol{w}_t||_2 = \frac{\eta^2}{2||\boldsymbol{w}_t||_2^3}||\frac{\partial\mathcal{L}}{\partial\boldsymbol{w}}\Big|_{\boldsymbol{w}=\tilde{\boldsymbol{w}}_t}||_2^2. \quad (13)$$

Rewrite Eq.(13), we have

$$\frac{\eta}{||\boldsymbol{w}_t||^2} \cdot ||\frac{\partial\mathcal{L}}{\partial\boldsymbol{w}}\Big|_{\boldsymbol{w}=\tilde{\boldsymbol{w}}_t}||_2 = \sqrt{2\lambda\eta}. \quad (14)$$

Recall $\frac{\eta}{||\boldsymbol{w}_t||^2}$ is effective learning rate $\eta_{eff}$. Combining with (14) and (5), we have

$$\Delta_t \approx \sqrt{2\lambda\eta}. \quad (15)$$

$\sqrt{2\lambda\eta}$ is exactly the theoretical value of angular update $\Delta_t$ in SGD case (Theorem 3 in main text).

Note the intuitive description on SMD is only suitable for SGD case. In momentum case, the tug-and-tar between effect of weight decay and gradient of loss is not as clear as that in pure SGD case. But our theorem 2/3 in main text shows SGDM can also obtain equilibrium, and the theoretical value of angular update is influenced by momentum factor $\alpha$ besides learning rate $\eta$ and weight decay factor $\lambda$.

## B   Related Work

In this section, we additionally review literatures about normalization techniques and weight decay.

**Normalization techniques**   Batch normalization (BN (Ioffe & Szegedy, 2015)) is proposed to deal with gradient vanishing/explosion, and accelerate the training of DNN. Rapidly, BN has been widely used in almost all kinds of deep learning tasks. Aside from BN, more types of normalization techniques have been proposed to remedy the defects of BN (Ioffe, 2017; Wu & He, 2018; Chiley et al., 2019; Yan et al., 2020) or to achieve better performance (Ba et al., 2016; Ulyanov et al., 2016; Salimans & Kingma, 2016; Shao et al., 2019; Singh & Krishnan, 2020). Though extremely effective, the mechanism of BN still remains as a mystery. Existing works attempt to analyze the function of BN: Ioffe & Szegedy (2015) claims BN can reduce the Internal Covariance Shift (ICS) of DNN; Santurkar et al. (2018) argue that the effectiveness of BN is not related to ICS, but the smoothness of normalized network; Luo et al. (2019) shows BN can be viewed as an implicit regularization technique; Cai et al. (2019) proves that with BN orthogonal least square problem can converge at linear rate; Dukler et al. (2020) proves weight normalization can speed up training in a two-layer ReLU network.

**Weight decay**   Weight decay (WD) is well-known as $l_2$ regularization, or ridge regression, in statistics. WD is also found to be extreme effective when applied in deep learning tasks. Krizhevsky & Geoffrey (2009) shows WD sometimes can even improve training accuracy not just generalization

performance; Zhang et al. (2019b) show WD can regularize the input-output Jacobian norm and reduce the effective damping coefficient; Li et al. (2020a) discusses the disharmony between WD and weight normalization. A more recent work Lewkowycz & Gur-Ari (2020) empirically finds the number of SGD steps $T$ until a model achieves maximum performance satisfies $T \propto \frac{1}{\lambda\eta}$, where $\lambda, \eta$ are weight decay factor and learning rate respectively, they interpret this phenomenon under the view of Neural Tangent Kernel (Jacot et al., 2018), showing that weight decay can accelerate the training process. Notice their result has no connection with equilibrium condition discussed in this work. Our results shows the cause of equilibrium condition can be reached long before neural network gets its highest performance.

**Equilibrium** Since the scale invariant property caused by normalization makes euclidean metrics of weight meaningless, researchers start to study the behavior of effective learning rate. van Laarhoven (2017); Chiley et al. (2019) estimate the magnitude of effective learning rate under equilibrium assumptions in SGD case; Hoffer et al. (2018) quantify effective learning rate without equilibrium assumption; Arora et al. (2019) proves that without WD, normalized neural network still can converge using fixed/decaying learning rate in Gradient Descent(GD)/SGD cases respectively; Zhang et al. (2019b) shows WD can increase effective learning rate; Li & Arora (2020) proves standard multi-stage learning rate schedule with BN and WD is equivalent to an exponential increasing learning rate schedule without WD. As a proposition, Li & Arora (2020) quantifies the magnitude of effective learning rate in SGDM case. But none of them have ever discussed why equilibrium condition can be reached. A recent work Li et al. (2020b) studies the convergence of effective learning rate by SDE, proving that the convergence time is of $\mathcal{O}(1/(\lambda\eta))$, where $\lambda, \eta$ are weight decay factor and learning rate respectively. Kunin et al. (2021) also depicts the equilibrium state by gradient flow.

**Adaptive learning rate method** Equilibrium state of Sperical Motion Dynamics makes all the relative update ($||\Delta\boldsymbol{w}_t||_2/||\boldsymbol{w}_t||_2$) of weights from different layers same and only determined by predefined hyper-parameters. Though this interesting phenomenon has not been widely accepted by optimization community (due to lack of justification on equilibrium), a bunch of large batch training methods (You et al., 2017, 2019) have adopted this "fixed relative update" motivation to design optimization algorithm for deep models. They do not connect the "fixed relative update" motivation with equilibrium phenomenon at all, but it is still worth being mentioned.

## C   Proof of Theorems

**Remark C.1.** *In the rest of context, we will use the following conclusions multiple times:* $\forall \delta, \varepsilon \in \mathbb{R}$, *if* $|\delta| \ll 1, |\varepsilon| \ll 1$, *then we have:*

$$(1 + \delta)^2 = 1 + 2\delta + o(\delta), \tag{16}$$

$$\sqrt{1 + \delta} = 1 + \frac{\delta}{2} + o(\delta), \tag{17}$$

$$\frac{1}{1 + \delta} = 1 - \delta + o(\delta), \tag{18}$$

$$(1 + \delta)(1 + \varepsilon) = 1 + \delta + \varepsilon + o(\delta + \varepsilon). \tag{19}$$

### C.1   Proof of Lemma 1

**Lemma 1.** *If* $\boldsymbol{w}$ *is scale-invariant with respect to* $\mathcal{L}(\boldsymbol{w})$ *, then for all* $k > 0$, *we have:*

$$\langle \boldsymbol{w}_t, \frac{\partial \mathcal{L}}{\partial \boldsymbol{w}}\Big|_{\boldsymbol{w}=\boldsymbol{w}_t} \rangle = 0 \tag{20}$$

$$\frac{\partial \mathcal{L}}{\partial \boldsymbol{w}}\Big|_{\boldsymbol{w}=k\boldsymbol{w}_t} = \frac{1}{k} \cdot \frac{\partial \mathcal{L}}{\partial \boldsymbol{w}}\Big|_{\boldsymbol{w}=\boldsymbol{w}_t}. \tag{21}$$

*Proof.* Given $\boldsymbol{w}_0 \in \mathbb{R}^p \backslash \{\boldsymbol{0}\}$, since $\forall k > 0, \mathcal{L}(\boldsymbol{w}_0) = \mathcal{L}(k\boldsymbol{w}_0)$, then we have

$$\frac{\partial \mathcal{L}(\boldsymbol{w})}{\partial \boldsymbol{w}}\Big|_{\boldsymbol{w}=\boldsymbol{w}_0} = \frac{\partial \mathcal{L}(k\boldsymbol{w})}{\partial \boldsymbol{w}}\Big|_{\boldsymbol{w}=\boldsymbol{w}_0} = \frac{\partial \mathcal{L}(\boldsymbol{w})}{\partial \boldsymbol{w}}\Big|_{\boldsymbol{w}=k\boldsymbol{w}_0} \cdot k \tag{22}$$

$$\frac{\partial \mathcal{L}(k\boldsymbol{w})}{\partial k}\bigg|_{\boldsymbol{w}=\boldsymbol{w}_0} = \langle \frac{\partial \mathcal{L}(\boldsymbol{w})}{\partial \boldsymbol{w}}\bigg|_{\boldsymbol{w}=k\boldsymbol{w}_0}, \boldsymbol{w}_0 \rangle = \frac{1}{k} \cdot \langle \frac{\partial \mathcal{L}(\boldsymbol{w})}{\partial \boldsymbol{w}}\bigg|_{\boldsymbol{w}=\boldsymbol{w}_0}, \boldsymbol{w}_0 \rangle = 0 \tag{23}$$

$\square$

## C.2 Proof of Theorem 1

**Lemma C.1.** *If the sequence $\{x_t\}_{t=1}^{\infty}$ satisfies*

$$x_t \geq \alpha x_{t-1} + \frac{L}{x_{t-1}}, \tag{24}$$

*where $x_1 > 0$, $L > 0$, $\alpha > 1/2$*

Then, we have

$$x_t \geq \sqrt{\frac{L}{1-\alpha}} - \alpha^{t-1}|\sqrt{\frac{L}{1-\alpha}} - x_1| \tag{25}$$

*Proof.* If $x_t \geq \sqrt{\frac{L}{1-\alpha}}$, since $\sqrt{\frac{L}{1-\alpha}} \geq \sqrt{\frac{L}{\alpha}}$, and $f(x) = \alpha x + L/x$ is monotonically increasing on $[\sqrt{L/\alpha}, \infty)$. Then we have

$$\begin{aligned} x_{t+1} &\geq \alpha x_t + \frac{L}{x_t} \\ &\geq \alpha\sqrt{\frac{L}{1-\alpha}} + \frac{L}{\sqrt{L/(1-\alpha)}} \\ &= \sqrt{\frac{L}{1-\alpha}}, \end{aligned} \tag{26}$$

which means $\forall k > t, x_k \geq \sqrt{L/(1-\alpha)}$.

If $0 < x_t < \sqrt{L/(1-\alpha)}$, then

$$\begin{aligned} \sqrt{\frac{L}{1-\alpha}} - x_{t+1} &\leq (\alpha - \frac{\sqrt{L(1-\alpha)}}{x_t})(\sqrt{\frac{L}{1-\alpha}} - x_t) \\ &< \alpha(\sqrt{\frac{L}{1-\alpha}} - x_t). \end{aligned} \tag{27}$$

Therefore, if $0 < x_T < \sqrt{L/(1-\alpha)}$, by induction it's easy to $\forall t \in [1, T-1]$, we have

$$0 < x_t < x_{t+1} \leq x_T < \sqrt{\frac{L}{1-\alpha}}, \tag{28}$$

$$(\sqrt{\frac{L}{1-\alpha}} - x_T) < \alpha^{T-1}(\sqrt{\frac{L}{1-\alpha}} - x_1). \tag{29}$$

Summarizing Eq.(26), Eq.(29), we have

$$x_t \geq \sqrt{\frac{L}{1-\alpha}} - \alpha^{t-1}|\sqrt{\frac{L}{1-\alpha}} - x_1| \tag{30}$$

$\square$

**Theorem 1.** *(Equilibrium in SGD) Assume the loss function is $\mathcal{L}(\boldsymbol{X}; \boldsymbol{w})$ with scale-invariant weight $\boldsymbol{w}$, $\boldsymbol{g}_t = \frac{\partial \mathcal{L}}{\partial \boldsymbol{w}}\big|_{\boldsymbol{X}_t, \boldsymbol{w}_t}$, $\tilde{\boldsymbol{g}}_t = \boldsymbol{g}_t \cdot ||\boldsymbol{w}_t||_2$. Consider the update rule of SGD with weight decay,*

$$\boldsymbol{w}_{t+1} = \boldsymbol{w}_t - \eta \cdot (\boldsymbol{g}_t + \lambda \boldsymbol{w}_t) \tag{31}$$

*where $\lambda, \eta \in (0, 1)$. If the following assumptions hold:*

    *1) $\lambda\eta \ll 1$ ($o(\lambda\eta)$ can be omitted);*

2) *Let $L_t = \mathbb{E}[||\tilde{\boldsymbol{g}}_t||_2^2|\boldsymbol{w}_t]$. $\exists V \in \mathbb{R}^+$, $\forall t \in \mathbb{N}^+$, $\mathbb{E}[(||\tilde{\boldsymbol{g}}_t||_2^2 - L_t)^2|\boldsymbol{w}_t] \leq V$;*

3) *$\forall t \in \mathbb{N}^+$, $L_t$ satisfies $|L_{t+1} - L_t| < 4\sqrt{V}(\lambda\eta)^{3/2}$;*

4) *$\exists l \in \mathbb{R}^+$, $\forall t \in \mathbb{N}^+$, $||\tilde{\boldsymbol{g}}_t||_2^2 > l$, $l > 2[\frac{2\lambda\eta}{1-2\lambda\eta}]^2 L_t$.*

*Then $\exists B > 0$, $\forall t \in \mathbb{N}^+$, $w_t^* = \sqrt[4]{L_{t-1}\eta/(2\lambda)}$, we have*

$$\mathbb{E}[||\boldsymbol{w}_t||_2^2 - (w_t^*)^2]^2 \leq (1 - 2\lambda\eta)^t B + \frac{2V\eta^2}{l(1 - 2\lambda\eta)}. \tag{32}$$

*Proof.* Since $\langle \boldsymbol{w}_t, \boldsymbol{g}_t \rangle = 0$, by Eq.(31) we have:

$$||\boldsymbol{w}_{t+1}||_2^2 = (1 - \eta\lambda)^2||\boldsymbol{w}_t||_2^2 + \frac{||\tilde{\boldsymbol{g}}_t||_2^2\eta^2}{||\boldsymbol{w}_t||_2^2} \tag{33}$$

Let $x_t$ denote $||\boldsymbol{w}_t||_2^2$, $G_t$ denote $||\tilde{\boldsymbol{g}}_t||_2^2$ and omit $\mathcal{O}((\eta\lambda)^2)$ part. Then according to assumptions 1), 2), 3), $G_t > l$, $\mathbb{E}[G_t|x_t] = L_t$, $\mathbb{E}[(G_t - L_t)^2|x_t] < V$, Eq.(33) can be rewritten as

$$x_{t+1} = (1 - 2\lambda\eta)x_t + \frac{G_t\eta^2}{x_t}. \tag{34}$$

The rest work is to prove the sequence $\{x_t\}_{t=1}^\infty$ will approach to another sequence $\{x_t^*\}_{t=1}^\infty$ in expectation, where $x_t^* = \sqrt{L_{t-1}\eta/(2\lambda)}$.

**Step 1, derive a lower bound of $x_t$ using Lemma C.1 and assumption 4.**

According to Lemma C.1, the lower bound of $x_t$ is

$$x_t > \sqrt{\frac{l\eta}{2\lambda}} - (1 - 2\lambda\eta)^t|x_0 - \sqrt{\frac{l\eta}{2\lambda}}|. \tag{35}$$

Eq.(35) implies when $t > \mathcal{T}(\lambda, \eta, l, x_0)$, where

$$\mathcal{T}(\lambda, \eta, l, x_0) = [1 + \frac{log((\sqrt{2} - 1)\sqrt{l\eta/(4\lambda)}) - log(|x_0 - \sqrt{l\eta/(4\lambda)}|)}{log(1 - 2\lambda\eta)}], \tag{36}$$

we have

$$x_t > \sqrt{\frac{l\eta}{4\lambda}}. \tag{37}$$

**Step 2, derive the relation between $\mathbb{E}(x_t - x_{t+1}^*)^2$ and $\mathbb{E}(x_{t+1} - x_{t+1}^*)^2$ using lower bound of $x_t$ and assumption 2, 4.**

Since $x_{t+1}^* = \sqrt{L_t\eta/(2\lambda)}$, Expanding $\mathbb{E}(x_{t+1} - x_{t+1}^*)^2$ we have

$$\begin{aligned} \mathbb{E}[(x_{t+1} - x_{t+1}^*)^2|x_t] &= \mathbb{E}[((1 - 2\lambda\eta - \frac{L_t\eta^2}{x_t x_{t+1}^*})(x_t - x_{t+1}^*) + \frac{(G_t - L_t)\eta^2}{x_t})^2|x_t] \\ &= (1 - 2\lambda\eta - \frac{L_t\eta^2}{x_t x_{t+1}^*})^2(x_t - x_{t+1}^*)^2 + \frac{\mathbb{E}[(G_t - L_t)^2|x_t]\eta^4}{x_t^2}. \end{aligned} \tag{38}$$

Now we need to prove

$$0 < 1 - 2\lambda\eta - \frac{L_t\eta^2}{x_t x_{t+1}^*} < 1 - 2\lambda\eta \tag{39}$$

when $t$ is sufficiently large. According to Eq.(36),(37), when $t > \mathcal{T}(\lambda, \eta, l, x_0)$, $x_t > \sqrt{\frac{l\eta}{4\lambda}}$, $x_{t+1}^* = \sqrt{L_t\eta/(2\lambda)}$, then we have

$$1 - 2\lambda\eta - \sqrt{\frac{2L_t}{l}} \cdot 2\lambda\eta < 1 - 2\lambda\eta - \frac{L_t\eta^2}{x_{t+1}^* x_t}. \tag{40}$$

Combining with assumption 4 in Theorem 1, we have

$$0 < 1 - 2\lambda\eta - \sqrt{\frac{2L_t}{l}} \cdot 2\lambda\eta < 1 - 2\lambda\eta - \frac{L_t}{x_{t+1}^* x_t}. \tag{41}$$

Combining with Eq.(37),(38),(41), if $t > \mathcal{T}(\lambda, \eta, l, x_0)$, we have

$$\mathbb{E}[(x_{t+1} - x_{t+1}^*)^2 | x_t] < (1 - 2\lambda\eta)^2 (x_t - x_{t+1}^*)^2 + \frac{4V\eta^3\lambda}{l}. \tag{42}$$

Considering the expectation with respect to the distribution of $x_t$, we have

$$\mathbb{E}(x_{t+1} - x_{t+1}^*)^2 < (1 - 2\lambda\eta)^2 \mathbb{E}(x_t - x_{t+1}^*)^2 + \frac{4V\eta^3\lambda}{l}. \tag{43}$$

**Step 3, derive the relation between $\mathbb{E}(x_{t+1} - x_{t+1}^*)^2$ and $\mathbb{E}(x_t - x_t^*)^2$ using assumption 4 and results in step 2**

Note the theoretical value at iteration $t$ is $x_t^*$ not $x_{t+1}^*$. Now we see at iteration $t$,

$$|x_t - x_{t+1}^*| \le |x_t - x_t^*| + |x_t^* - x_{t+1}^*|. \tag{44}$$

Since $\forall t, G_t > l, L_t = \mathbb{E}[G_t | x_t] > l$, we have

$$|x_t^* - x_{t+1}^*| \le (\sqrt{L_{t-1}} - \sqrt{L_t}) \cdot \sqrt{\frac{\eta}{2\lambda}} = \frac{|L_{t-1} - L_t|}{\sqrt{L_{t-1}} + \sqrt{L_t}} \cdot \sqrt{\frac{\eta}{2\lambda}} \le \frac{|L_{t-1} - L_t|}{2\sqrt{l}} \cdot \sqrt{\frac{\eta}{2\lambda}}. \tag{45}$$

Combining with assumption 3, we have

$$|x_t - x_{t+1}^*| \le |x_t - x_t^*| + \frac{\sqrt{2V}\eta^2\lambda}{\sqrt{l}} \tag{46}$$

Now considering the expectation w.r.t. distribution of $x_t$, we have

$$\mathbb{E}(x_t - x_{t+1}^*)^2 \le \mathbb{E}[|x_t - x_t^*| + \frac{\sqrt{2V}\eta^2\lambda}{\sqrt{l}}]^2 \le [\sqrt{\mathbb{E}(x_t - x_t^*)^2} + \frac{\sqrt{2V}\eta^2\lambda}{\sqrt{l}}]^2 \tag{47}$$

Now we consider two cases separately: if $\lambda\eta\sqrt{\mathbb{E}(x_t - x_t^*)^2} > \frac{\sqrt{2V}\eta^2\lambda}{\sqrt{l}}$, then

$$\mathbb{E}(x_t - x_{t+1}^*)^2 \le [\sqrt{\mathbb{E}(x_t - x_t^*)^2} + \frac{\sqrt{2V}\eta^2\lambda}{\sqrt{l}}]^2 \le (1 + \lambda\eta)^2 \mathbb{E}(x_t - x_t^*)^2. \tag{48}$$

Combining with Eq.(43), we have

$$\mathbb{E}(x_{t+1} - x_{t+1}^*)^2 < (1 - 2\lambda\eta)\mathbb{E}(x_t - x_t^*)^2 + \frac{4V\eta^3\lambda}{l}. \tag{49}$$

Else if $\lambda\eta\sqrt{\mathbb{E}(x_t - x_t^*)^2} < \frac{\sqrt{2V}\eta^2\lambda}{\sqrt{l}}$, i.e. $\mathbb{E}(x_t - x_t^*)^2 < \frac{2V\eta^2}{l}$, then by Eq.(47), we have

$$\mathbb{E}(x_t - x_{t+1}^*)^2 \le [\sqrt{\mathbb{E}(x_t - x_t^*)^2} + \frac{\sqrt{2V}\eta^2\lambda}{\sqrt{l}}]^2 < (1 + \lambda\eta)^2 \frac{2V\eta^2}{l}. \tag{50}$$

Combining with Eq.(43), we have

$$\mathbb{E}(x_{t+1} - x_{t+1}^*)^2 < (1 - 2\lambda\eta)\frac{2V\eta^2}{l} + \frac{4V\eta^3\lambda}{l} = \frac{2V\eta^2}{l}. \tag{51}$$

Summary Eq.(49), (51), we have

$$\mathbb{E}(x_{t+1} - x_{t+1}^*)^2 < max\{(1 - 2\lambda\eta)\mathbb{E}(x_t - x_t^*)^2 + \frac{4V\eta^3\lambda}{l}, \frac{2V\eta^2}{l}\}. \tag{52}$$

**Step 4, derive the upper bound of $\mathbb{E}(x_{t+1} - x_{t+1}^*)^2$ given $t$**

According to Eq.(52), by applying deduction method, when $t > \mathcal{T}(\lambda, \eta, l, x_0)$, we have

$$\mathbb{E}(x_t - x_t^*)^2 \leq (1 - 2\lambda\eta)^{t-\mathcal{T}} \mathbb{E}(x_\mathcal{T} - x_\mathcal{T}^*)^2 + \frac{2V\eta^2}{l} \tag{53}$$

Now we have derived the upper bound of $\mathbb{E}(x_{t+1} - x_{t+1}^*)^2$ when $t \geq \mathcal{T}(\lambda, \eta, l, x_0)$, we need to take $t < \mathcal{T}(\lambda, \eta, l, x_0)$ into account. To do that, we just need to prove $B = max\{(1 - 2\lambda\eta)^{-t} \mathbb{E}(x_t - x_t^*)^2 | t = 1, 2, ..., \mathcal{T}$ exists, where $\mathcal{T} = \mathcal{T}(\lambda, \eta, l, x_0)$. Specifically, we need to prove $\forall t \in \{0, 1, 2, ..., \mathcal{T}, \mathbb{E}(x_t - x_t^*)^2$ is finite. According to Lemma C.1, since $G_t > l$, $x_t \geq min\{x_0, \sqrt{\frac{l\eta}{2\lambda}}\}$, we have

$$\mathbb{E}x_t^2 = \mathbb{E}[(1 - 2\lambda\eta)^2 x_{t-1}^2 + 2(1 - 2\lambda\eta)G_t + \frac{G_t^2}{x_{t-1}^2}]$$
$$< (1 - 4\lambda\eta)\mathbb{E}x_{t-1}^2 + 2(1 - 2\lambda\eta)L_t + \frac{V + L_t^2}{min(x_0^2, l\eta/(2\lambda))}. \tag{54}$$

Eq.(54) implies if $\mathbb{E}x_{t-1}^2 < \infty$, then $\mathbb{E}x_t^2 < \infty$. Hence by deduction methods, $\forall t \in [0, \mathcal{T}(\lambda, \eta, l, x_0)]$, $\mathbb{E}x_t^2$ is finite. Next, we can bounded $\mathbb{E}(x_t - x^*)^2$ by

$$\mathbb{E}(x_t - x^*)^2$$
$$< \mathbb{E}x_t^2 + 2x_t^* |\mathbb{E}x_t| + (x_t^*)^2$$
$$< \mathbb{E}x_t^2 + 2x_t^* \sqrt{\mathbb{E}x_t^2} + (x_t^*)^2 \tag{55}$$
$$< \infty.$$

Therefore, $B$ exists, and satisfies: when $t \leq \mathcal{T}(\lambda, \eta, l, x_0)$, we have

$$\mathbb{E}(\boldsymbol{x}_t - x^*)^2 = (1 - 2\lambda\eta)^t \cdot (1 - 2\lambda\eta)^{-t} \mathbb{E}(\boldsymbol{x}_t - x_t^*)^2 < (1 - 2\lambda\eta)^t B < (1 - 2\lambda\eta)^t B + \frac{2V\eta^2}{l}; \tag{56}$$

when $t > \mathcal{T}(\lambda, \eta, l, x_0)$, we have

$$\mathbb{E}(x_t - x_t^*)^2 \leq (1 - 2\lambda\eta)^{t-\mathcal{T}} \mathbb{E}(x_\mathcal{T} - x_\mathcal{T}^*)^2 + \frac{2V\eta^2}{l} \leq (1 - 2\lambda\eta)^t B + \frac{2V\eta^2}{l}; \tag{57}$$

In summary, $\forall t > 0$, we have

$$\mathbb{E}(x_t - x^*)^2 < (1 - 2\lambda\eta)^t B + \frac{2V\eta^2}{l}. \tag{58}$$

$\square$

## C.3 Proof of Theorem 2

**Lemma C.2.** *Assume $\alpha, \beta, \varepsilon \in (0, 1)$, where $\beta \ll 1$. Denote $diag(1 - \frac{2\beta}{1-\alpha}, \alpha, \alpha^2 + \frac{2\alpha^2}{1-\alpha}\beta)$ as $\boldsymbol{\Lambda}$, $\boldsymbol{k} = (\frac{1}{(1-\alpha)^2}, -\frac{2\alpha}{(1-\alpha)^2}, \frac{\alpha^2}{(1-\alpha)^2})^T$, $\boldsymbol{e} = (1, 1, 1)^T$. If $\varepsilon < \frac{1}{3}[\frac{1-\alpha^2}{\beta} - \frac{8}{1-\alpha}]$, then $\forall \boldsymbol{d} \in \mathbb{R}^p$, we have*

$$||(\boldsymbol{\Lambda} - \varepsilon\beta(1 - \alpha)^2 \boldsymbol{k}\boldsymbol{e}^T)\boldsymbol{d}||_2^2 < (1 - \frac{4\beta}{1 - \alpha})||\boldsymbol{d}||_2^2 \tag{59}$$

*Proof.* Omit $O(\beta^2)$ part, we have

$$||(\boldsymbol{\Lambda} - \varepsilon\beta(1 - \alpha)^2 \boldsymbol{k}\boldsymbol{e}^T)\boldsymbol{d}||_2^2$$
$$= (1 - \frac{4\beta}{1 - \alpha})d_1^2 + \alpha^2 d_2^2 + \alpha^4(1 + \frac{4\beta}{1 - \alpha})d_3^2 \tag{60}$$
$$- 2\varepsilon\beta(d_1 + d_2 + d_3)(d_1 - 2\alpha^2 d_2 + \alpha^4 d_3)$$

First, we need to estimate the lower bound of $(d_1 + d_2 + d_3)(d_1 - 2\alpha^2 d_2 + \alpha^4 d_3)$ by

$$
\begin{aligned}
&(d_1 + d_2 + d_3)(d_1 - 2\alpha^2 d_2 + \alpha^4 d_3) \\
=&\frac{[d_1 + (1 - 2\alpha^2)d_2]^2}{2} + \frac{[d_1 + (1 + \alpha^4)d_3]^2}{2} \\
&- (1/2 + 2\alpha^4)d_2^2 - \frac{1 + \alpha^8}{2} + (\alpha^4 - 2\alpha^2)d_2 d_3 \\
\geq&-(\frac{1}{2} + 2\alpha^4 + \frac{\alpha^4}{2})d_2^2 - (\frac{\alpha^8}{2} + \frac{\alpha^4}{2} - 2\alpha^2 + \frac{5}{2})d_3^2 \\
\geq&-3d_2^2 - \frac{5}{2}d_3^2.
\end{aligned}
\tag{61}
$$

Then combining Eq.(60), (61), we have

$$
\begin{aligned}
&||(\boldsymbol{\Lambda} - \varepsilon\beta(1 - \alpha)^2\boldsymbol{k}\boldsymbol{e}^T)\boldsymbol{d}||_2^2 \\
\leq&(1 - \frac{4\beta}{1 - \alpha})d_1^2 + (\alpha^2 + 3\beta\varepsilon)d_2^2 \\
&+(\alpha^4 + \frac{4\beta\alpha^4}{1 - \alpha} + \frac{5\beta\varepsilon}{2})d_3^2.
\end{aligned}
\tag{62}
$$

Since $\varepsilon < \frac{1}{3}[\frac{1-\alpha^2}{\beta} - \frac{8}{1-\alpha}]$, we have

$$
\alpha^2 + 3\beta\varepsilon < 1 - \frac{4\beta}{1 - \alpha},
\tag{63}
$$

$$
\alpha^4 + \frac{4\beta\alpha^4}{1 - \alpha} + \frac{5\beta\varepsilon}{2} < 1 - \frac{4\beta}{1 - \alpha}.
\tag{64}
$$

Hence, by Eq.(62) we have

$$
||(\boldsymbol{\Lambda} - \varepsilon\beta(1 - \alpha)^2\boldsymbol{k}\boldsymbol{e}^T)\boldsymbol{d}||_2^2 < (1 - \frac{4\beta}{1 - \alpha})||\boldsymbol{d}||_2^2
\tag{65}
$$

$\square$

**Theorem 2.** *(Equilibrium in SGDM) Considering the update rule of SGDM (heavy ball method (Polyak, 1964)):*

$$
\begin{aligned}
\boldsymbol{v}_t &= \alpha\boldsymbol{v}_{t-1} + \boldsymbol{g}_t + \lambda\boldsymbol{w}_t \\
\end{aligned}
\tag{66}
$$

$$
\begin{aligned}
\boldsymbol{w}_{t+1} &= \boldsymbol{w}_t - \eta\boldsymbol{v}_t
\end{aligned}
\tag{67}
$$

*where $\lambda, \eta \in (0, 1), \alpha \in (\frac{1}{2}, 1)$. If following assumptions hold:*

5) *$\lambda\eta \ll 1$, $\lambda\eta < (1 - \sqrt{\alpha})^2$;*

6) *Define $h_t = ||\boldsymbol{g}_t||_2^2 + 2\alpha\langle\boldsymbol{v}_{t-1}, \boldsymbol{g}_t\rangle$, $\tilde{h}_t = h_t \cdot ||\boldsymbol{w}_t||_2^2$, $L_t = \mathbb{E}[\tilde{h}_t|\boldsymbol{w}_t]$. $\exists V \in \mathbb{R}^+, \forall t \in \mathbb{N}^+$, $\mathbb{E}[(\tilde{h}_t - L_t)^2|\boldsymbol{w}_t] \leq V$;*

7) *$\forall t \in \mathbb{N}^+$, $L_t$ satisfies $|L_{t+1} - L_t| < 4\sqrt{V}(\lambda\eta)^{3/2}$;*

8) *$\exists l \in \mathbb{R}^+, \forall t \in \mathbb{N}^+, \tilde{h}_t > l > 2[\frac{6\lambda\eta}{(1-\alpha)^3(1+\alpha)-8\lambda\eta(1-\alpha)}]^2 L_t$, ;*

*then $\exists B, C > 0$, $C$ only depends on $\alpha$, $w_t^* = \sqrt[4]{L_{t-1}\eta/(\lambda(1 - \alpha)(2 - \lambda\eta/(1 + \alpha)))}$, we have*

$$
\mathbb{E}[||\boldsymbol{w}_t||_2^2 - (w_t^*)^2]^2 \leq (1 - \frac{2\lambda\eta}{1 - \alpha})^t B + \frac{V\eta^2}{l}C,
\tag{68}
$$

*Proof.* The update rule is

$$
\begin{aligned}
\boldsymbol{w}_{t+1} &= \boldsymbol{w}_t - \eta\boldsymbol{v}_t \\
&= \boldsymbol{w}_t - \eta(\alpha\boldsymbol{v}_{t-1} + \frac{\tilde{\boldsymbol{g}}_t}{||\boldsymbol{w}_t||} + \lambda\boldsymbol{w}_t) \\
&= \boldsymbol{w}_t - \eta(\alpha\frac{\boldsymbol{w}_{t-1} - \boldsymbol{w}_t}{\eta} + \frac{\tilde{\boldsymbol{g}}_t}{||\boldsymbol{w}_t||} + \lambda\boldsymbol{w}_t) \\
&= (1 - \eta\lambda + \alpha)\boldsymbol{w}_t - \alpha\boldsymbol{w}_{t-1} - \boldsymbol{g}_t\eta.
\end{aligned}
\tag{69}
$$

Derive the update of weight norm by Eq.(69), we have

$$
\begin{aligned}
&||\boldsymbol{w}_{t+1}||_2^2 \\
=&(1-\eta\lambda+\alpha)^2||\boldsymbol{w}_t||_2^2 - 2\alpha(1+\alpha-\eta\lambda)\langle\boldsymbol{w}_t,\boldsymbol{w}_{t-1}\rangle \\
&+ \alpha^2||\boldsymbol{w}_{t-1}||_2^2 + ||\boldsymbol{g}_t||_2^2\eta^2 + 2\langle\alpha\boldsymbol{w}_{t-1},\boldsymbol{g}_t\eta\rangle \\
=&(1-\eta\lambda+\alpha)^2||\boldsymbol{w}_t||_2^2 - 2\alpha(1+\alpha-\eta\lambda)\langle\boldsymbol{w}_t,\boldsymbol{w}_{t-1}\rangle \\
&+ \alpha^2||\boldsymbol{w}_{t-1}||_2^2 + ||\boldsymbol{g}_t||_2^2\eta^2 + 2\langle\alpha(\boldsymbol{w}_t+\eta\boldsymbol{v}_{t-1}),\boldsymbol{g}_t\eta\rangle \\
=&(1-\eta\lambda+\alpha)^2||\boldsymbol{w}_t||_2^2 - 2\alpha(1+\alpha-\eta\lambda)\langle\boldsymbol{w}_t,\boldsymbol{w}_{t-1}\rangle \\
&+ \alpha^2||\boldsymbol{w}_{t-1}||_2^2 + \frac{\tilde{h}_t\eta^2}{||\boldsymbol{w}_t||_2^2}.
\end{aligned}
\tag{70}
$$

Derive the update of inner product by Eq.(69) and the fact that $\tilde{\boldsymbol{g}}_t$ is perpendicular to $\boldsymbol{w}_t$, we have

$$
\langle\boldsymbol{w}_{t+1},\boldsymbol{w}_t\rangle = (1+\alpha-\lambda\eta)||\boldsymbol{w}_t||^2 - \alpha\langle\boldsymbol{w}_t,\boldsymbol{w}_{t-1}\rangle.
\tag{71}
$$

Now Eq.(70), (71) can be formulated as a 3 dimensional iterative map, let $\boldsymbol{X}_t, \boldsymbol{A}, \boldsymbol{e}$ denote:

$$
\boldsymbol{X}_t = \begin{pmatrix} a_t \\ b_t \\ c_t \end{pmatrix} = \begin{pmatrix} ||\boldsymbol{w}_t||_2^2 \\ \langle\boldsymbol{w}_t,\boldsymbol{w}_{t-1}\rangle \\ ||\boldsymbol{w}_{t-1}||_2^2 \end{pmatrix},
\tag{72}
$$

$$
\boldsymbol{A} = \begin{pmatrix} (1+\alpha-\lambda\eta)^2 & -2\alpha(1+\alpha-\lambda\eta) & \alpha^2 \\ 1+\alpha-\lambda\eta & -\alpha & 0 \\ 1 & 0 & 0 \end{pmatrix},
\tag{73}
$$

$$
\boldsymbol{e} = \begin{pmatrix} 1 \\ 0 \\ 0 \end{pmatrix},
\tag{74}
$$

$$
\tag{75}
$$

respectively, then we have

$$
\boldsymbol{X}_{t+1} = \boldsymbol{A}\boldsymbol{X}_t + \frac{\tilde{h}_t\eta^2}{\boldsymbol{e}^T\boldsymbol{X}_t}\boldsymbol{e}.
\tag{76}
$$

The rest process is to prove $\{\boldsymbol{X}_t\}$ will approach to another sequence $\{\boldsymbol{X}_t^*\}$, where $\boldsymbol{X}_t^*$ denotes the solution of equation

$$
\boldsymbol{X} = \boldsymbol{A}\boldsymbol{X} + \frac{L_{t-1}\eta^2}{\boldsymbol{e}^T\boldsymbol{X}}\boldsymbol{e}
\tag{77}
$$

**Step 1, simplify the iterative equation Eq.(76) by eigenvalue decomposition.**

When $\lambda\eta < (1-\sqrt{\alpha})^2$, the eigen value of $\boldsymbol{A}$ are all real number, and explicity computed as

$$
\begin{aligned}
\lambda_1 =& \frac{(1+\alpha-\lambda\eta)^2 + (1+\alpha-\lambda\eta)\sqrt{(1+\alpha-\lambda\eta)^2-4\alpha}}{2} \\
&-\alpha \\
=& 1 - \frac{2\lambda\eta}{1-\alpha} + \mathcal{O}(\lambda^2\eta^2), \\
\lambda_2 =& \alpha, \\
\lambda_3 =& \frac{(1+\alpha-\lambda\eta)^2 - (1+\alpha-\lambda\eta)\sqrt{(1+\alpha-\lambda\eta)^2-4\alpha}}{2} \\
&-\alpha \\
=& \alpha^2 + \frac{2\alpha^2}{(1-\alpha)}\lambda\eta + \mathcal{O}(\lambda^2\eta^2),
\end{aligned}
$$

and they satisfy

$$
0 < \lambda_3 < \lambda_2 = \alpha < \lambda_1 < 1.
\tag{78}
$$

We omit $\mathcal{O}(\lambda^2\eta^2)$ in following derivation. Considering a eigenvalue decomposition of $\boldsymbol{A}$

$$\boldsymbol{S}^{-1}\boldsymbol{A}\boldsymbol{S} = \boldsymbol{\Lambda}, \tag{79}$$

where $\boldsymbol{\Lambda}$ is a diagonal matrix whose diagonal elements are the eigen value of $\boldsymbol{A}$; the column vector of $\boldsymbol{S}$ is the eigen vectors of $\boldsymbol{A}$, note the fromulation of $\boldsymbol{S}, \boldsymbol{\Lambda}$ are not unique. Specifically, we set $\boldsymbol{\Lambda}, \boldsymbol{S}$ as Eq.(80), (81), the inverse of $\boldsymbol{S}$ exists, and can be explicitly expressed as Eq.(82).

$$\boldsymbol{\Lambda} = \begin{pmatrix} \lambda_1 & 0 & 0 \\ 0 & \lambda_2 & 0 \\ 0 & 0 & \lambda_3 \end{pmatrix}, \tag{80}$$

$$\boldsymbol{S} = \begin{pmatrix} 1 & 1 & 1 \\ \frac{1+\alpha-\lambda\eta}{\alpha+\lambda_1} & \frac{1+\alpha-\lambda\eta}{\alpha+\lambda_2} & \frac{1+\alpha-\lambda\eta}{\alpha+\lambda_3} \\ \frac{1}{\lambda_1} & \frac{1}{\lambda_2} & \frac{1}{\lambda_3} \end{pmatrix}. \tag{81}$$

$$\boldsymbol{S}^{-1} = \begin{pmatrix} \frac{(\alpha+\lambda_1)\lambda_1}{(\lambda_1-\alpha)(\lambda_1-\lambda_3)} & -\frac{2\lambda_1(\alpha+\lambda_1)(\alpha+\lambda_3)}{(\lambda_1-\lambda_3)(\lambda_1-\alpha)(1+\alpha-\beta)} & \frac{\lambda_1\lambda_3(\alpha+\lambda_1)}{(\lambda_1-\alpha)(\lambda_1-\lambda_3)} \\ -\frac{2\alpha^2}{(\lambda_1-\alpha)(\alpha-\lambda_3)} & \frac{2\alpha(\alpha+\lambda_1)(\alpha+\lambda_3)}{(\lambda_1-\alpha)(\alpha-\lambda_3)(1+\alpha-\beta)} & -\frac{2\alpha\lambda_1\lambda_3}{(\lambda_1-\alpha)(\alpha-\lambda_3)} \\ \frac{(\alpha+\lambda_3)\lambda_3}{(\alpha-\lambda_3)(\lambda_1-\lambda_3)} & -\frac{2\lambda_3(\alpha+\lambda_3)(\alpha+\lambda_1)}{(\lambda_1-\lambda_3)(\alpha-\lambda_3)(1+\alpha-\beta)} & \frac{(\alpha+\lambda_3)\lambda_1\lambda_3}{(\alpha-\lambda_3)(\lambda_1-\lambda_3)} \end{pmatrix}. \tag{82}$$

**Remark C.2.** *The computing process of eigenvalue decomposition of $\boldsymbol{A}$ is straightforward and tedious, so we omit them. Besides, notice $\boldsymbol{A}$ is not symmetric, its eigenvalue decomposition is not equivalent to its SVD(singular value decomposition). The eigen vectors of $\boldsymbol{A}$ are not necessarily unit or orthogonal to each other*

Let $\boldsymbol{Y}_t = \boldsymbol{S}^{-1}\boldsymbol{X}_t$, combining with Eq.(76), we have

$$\boldsymbol{Y}_{t+1} = \boldsymbol{\Lambda}\boldsymbol{Y}_t + \frac{\tilde{h}_t\eta^2}{(\boldsymbol{S}^T\boldsymbol{e})^T\boldsymbol{Y}_t}\boldsymbol{S}^{-1}\boldsymbol{e}. \tag{83}$$

Combining with Eq.(81) and Eq.(82), and set $\boldsymbol{Y}_t = (\tilde{a}_t, \tilde{b}_t, \tilde{c}_t)^T$, Eq.(83) can explicitly expressed as

$$\tilde{a}_{t+1} = \lambda_1\tilde{a}_t + \frac{\tilde{h}_t\eta^2}{\tilde{a}_t + \tilde{b}_t + \tilde{c}_t} \cdot \frac{(\alpha+\lambda_1)\lambda_1}{(\lambda_1-\alpha)(\lambda_1-\lambda_3)}, \tag{84}$$

$$\tilde{b}_{t+1} = \alpha\tilde{b}_t - \frac{\tilde{h}_t\eta^2}{\tilde{a}_t + \tilde{b}_t + \tilde{c}_t} \cdot \frac{2\alpha^2}{(\lambda_1-\alpha)(\alpha-\lambda_3)}, \tag{85}$$

$$\tilde{c}_{t+1} = \lambda_3\tilde{c}_t + \frac{\tilde{h}_t\eta^2}{\tilde{a}_t + \tilde{b}_t + \tilde{c}_t} \cdot \frac{(\alpha+\lambda_3)\lambda_3}{(\alpha-\lambda_3)(\lambda_1-\lambda_3)}. \tag{86}$$

**Step 2. derive the lower bound of $||\boldsymbol{w}_t||_2^2 = \tilde{a}_t + \tilde{b}_t + \tilde{c}_t$ like step 1 in theorem 1**

To do that, we need to prove following inequations (Eq.(87) - Eq.(91)) by mathematical induction method

$$\tilde{b}_t < 0, \tag{87}$$

$$\tilde{c}_t > 0, \tag{88}$$

$$(\alpha - \lambda_1)\tilde{b}_t > (\lambda_1 - \lambda_3)\tilde{c}_t, \tag{89}$$

$$\tilde{a}_t + \tilde{b}_t + \tilde{c}_t > 0, \tag{90}$$

$$\tilde{a}_{t+1} + \tilde{b}_{t+1} + \tilde{c}_{t+1}, > \lambda_1(\tilde{a}_t + \tilde{b}_t + \tilde{c}_t) + \frac{\tilde{h}_t\eta^2}{\tilde{a}_t + \tilde{b}_t + \tilde{c}_t}. \tag{91}$$

According to Eq.(72), start point $\boldsymbol{X}_1$ satisfies

$$\boldsymbol{X}_1 = (||\boldsymbol{w}_1||_2^2, \langle \boldsymbol{w}_1, \boldsymbol{w}_0 \rangle, ||\boldsymbol{w}_0||_2^2)^T = (a_1, a_1, a_1)^T \tag{92}$$

where $\boldsymbol{w}_0 = \boldsymbol{w}_1$, $a_1 = ||\boldsymbol{w}_1||_2^2$. Then $\boldsymbol{Y}_1 = \boldsymbol{S}^{-1}\boldsymbol{X}_1$. Combining with Eq.(85), (86), we have

$$\tilde{b}_1 = -\frac{2\alpha^2\lambda\eta}{(\lambda_1 - \alpha)(\alpha - \lambda_3)}a_1, \tag{93}$$

$$\tilde{c}_1 = \frac{\lambda_3(\lambda_3 + \alpha)(1 - \alpha + \lambda\eta)}{(\lambda_3 - \alpha)(\lambda_1 - \lambda_3)(1 + \alpha - \lambda\eta)}\left(\frac{1 - \alpha - \lambda\eta}{1 - \alpha + \lambda\eta} - \lambda_1\right)a_1, \tag{94}$$

Since

$$\lambda_3 < \alpha \tag{95}$$

$$\frac{1 - \alpha - \lambda\eta}{1 - \alpha + \lambda\eta} - \lambda_1 = \mathcal{O}(\lambda^2\eta^2) \tag{96}$$

then we have

$$(\alpha - \lambda_1)\tilde{b}_1 = \mathcal{O}(\lambda\eta) > \mathcal{O}(\lambda^2\eta^2) = (\lambda_1 - \lambda_3)\tilde{c}_1. \tag{97}$$

Besides

$$\tilde{a}_1 + \tilde{b}_1 + \tilde{c}_1 = \boldsymbol{e}^T\boldsymbol{S}\boldsymbol{Y}_1 = \boldsymbol{e}^T\boldsymbol{X}_1 = a_1 > 0. \tag{98}$$

Sum Eq.(84), Eq.(85), Eq.(86), set $t = 1$, and combine with Eq.(97) we have

$$\begin{aligned}
\tilde{a}_2 + \tilde{b}_2 + \tilde{c}_2 &= \lambda_1\tilde{a}_1 + \alpha\tilde{b}_1 + \lambda_3\tilde{c}_1 + \frac{\tilde{h}_1\eta^2}{\tilde{a}_1 + \tilde{b}_1 + \tilde{c}_1}, \\
&\geq \lambda_1(\tilde{a}_1 + \tilde{b}_1 + \tilde{c}_1) + \frac{\tilde{h}_1\eta^2}{\tilde{a}_1 + \tilde{b}_1 + \tilde{c}_1}.
\end{aligned} \tag{99}$$

Therefore, for $t = 1$, Eq.(87) - (91) holds.

Suppose for $t = T$, Eq. (87) - (91)) hold. Then by Eq.(85), (86), we have $\tilde{b}_{T+1} < 0, \tilde{a}_{T+1} > 0$, so Eq.(87), (88) hold for $t = T + 1$;

Combining with Eq.(85), (86), (89), we have

$$\begin{aligned}
&(\alpha - \lambda_1)\tilde{b}_{T+1} \\
&= \alpha(\alpha - \lambda_1)\tilde{b}_T + \frac{\tilde{h}_t\eta^2}{\tilde{a}_T + \tilde{b}_T + \tilde{c}_T} \cdot \frac{2\alpha^2}{(\alpha - \lambda_3)} \\
&> \lambda_3(\lambda_1 - \lambda_3)\tilde{c}_T + \frac{\tilde{h}_t\eta^2}{\tilde{a}_T + \tilde{b}_T + \tilde{c}_T} \cdot \frac{(\alpha + \lambda_3)\lambda_3}{(\alpha - \lambda_3)} \\
&= (\lambda_1 - \lambda_3)\tilde{c}_{T+1},
\end{aligned} \tag{100}$$

thus Eq.(89) holds for $t = T + 1$.

Sum Eq.(84), Eq.(85), Eq.(86) for $t = T + 1$, due to Eq.(100) we have

$$\begin{aligned}
&\tilde{a}_{T+2} + \tilde{b}_{T+2} + \tilde{c}_{T+2} \\
&= \lambda_1\tilde{a}_{T+1} + \alpha\tilde{b}_{T+1} + \lambda_3\tilde{c}_{T+1} + \frac{\tilde{h}_t\eta^2}{\tilde{a}_{T+1} + \tilde{b}_{T+1} + \tilde{c}_{T+1}} \\
&> \lambda_1(\tilde{a}_{T+1} + \tilde{b}_{T+1} + \tilde{c}_{T+1}) + \frac{\tilde{h}_t\eta^2}{\tilde{a}_{T+1} + \tilde{b}_{T+1} + \tilde{c}_{T+1}},
\end{aligned} \tag{101}$$

Eq.(91) holds for $t = T + 1$.

Combining with the fact that $\tilde{a}_T + \tilde{b}_T + \tilde{c}_T > 0$, when $t = T$ in Eq.(91), we have

$$\begin{aligned}
&\tilde{a}_{T+1} + \tilde{b}_{T+1} + \tilde{c}_{T+1} \\
&> \lambda_1(\tilde{a}_T + \tilde{b}_T + \tilde{c}_T) + \frac{\tilde{h}_t\eta^2}{\tilde{a}_T + \tilde{b}_T + \tilde{c}_T} \\
&> 0
\end{aligned} \tag{102}$$

Hence Eq.(90) holds for $t = T + 1$.

In summary, Eq.(87) - (91) hold $\forall t > 0$. Eq.(91) allow us to estimate the lower bound of $\tilde{a}_t + \tilde{b}_t + \tilde{c}_t$ when $t$ is sufficient large. Define $\mathcal{T}(\alpha, \lambda, \eta, l, a_1)$ as Eq.(103)

$$
\begin{aligned}
&\mathcal{T}(\alpha, \lambda, \eta, l, a_1) \\
&= [1 + \frac{log((\sqrt{2}-1)\sqrt{\frac{l\eta}{4\lambda(1-\alpha)}}) - log(|x_0 - \sqrt{\frac{l\eta}{4\lambda(1-\alpha)}}|)}{log(1 - \frac{2\lambda\eta}{1-\alpha})}]
\end{aligned}
\tag{103}
$$

According to Lemma C.1, when $t > \mathcal{T}(\alpha, \lambda, \eta, l, a_1)$, we have

$$
\tilde{a}_t + \tilde{b}_t + \tilde{c}_t \geq \sqrt{\frac{l\eta}{2(1-\lambda_1)}} = \sqrt{\frac{l\eta(1-\alpha)}{4\lambda}} + \mathcal{O}(\lambda\eta)
\tag{104}
$$

**Step 3, derive the relation between $\mathbb{E}||Y_t - Y_{t+1}^*||_2^2$ and $\mathbb{E}||Y_{t+1} - Y_{t+1}^*||_2^2$ like step 2 in theorem 1.**

$Y_{t+1}^* = (\tilde{a}_{t+1}^*, \tilde{b}_{t+1}^*, \tilde{c}_{t+1}^*)^T$ denotes the solution of

$$
Y = \Lambda Y + \frac{L_t \eta^2}{(S^T e)^T Y} S^{-1} e.
\tag{105}
$$

Recall $a_{t+1} = \tilde{a}_{t+1} + \tilde{b}_{t+1} + \tilde{c}_{t+1}$, so $a_{t+1}^* = \tilde{a}_{t+1}^* + \tilde{b}_{t+1}^* + \tilde{c}_{t+1}^* > 0$, then $a_{t+1}^*$ is computed as

$$
a_{t+1}^* = \tilde{a}_{t+1}^* + \tilde{b}_{t+1}^* + \tilde{c}_{t+1}^* = \sqrt{\frac{L_t \eta}{\lambda(1-\alpha)(2 - \frac{\lambda\eta}{1+\alpha})}}.
\tag{106}
$$

According to assumption 6, $\mathbb{E}\tilde{h}_t = L_t$, then we have:

$$
Y_{t+1} - Y_{t+1}^* = (\Lambda - \frac{L_t \eta^2}{a_t a_{t+1}^*} k e^T)(Y_t - Y_{t+1}^*) + \frac{(\tilde{h}_t - L_t)\eta^2}{a_t} k
\tag{107}
$$

where $k = (k_1, k_2, k_3)^T = S^{-1} e$. In the following context, we will omit the $\mathcal{O}(\lambda^2 \eta^2)$ part since $\lambda\eta \ll 1$. $k_1, k_2, k_3$ can be approximated by first order Taylor expansion as

$$
k_1 = \frac{1}{(1-\alpha)^2} + \mathcal{O}(\lambda\eta),
\tag{108}
$$

$$
k_2 = -\frac{2\alpha}{(1-\alpha)^2} + \mathcal{O}(\lambda\eta),
\tag{109}
$$

$$
k_3 = \frac{\alpha^2}{(1-\alpha)^2} + \mathcal{O}(\lambda\eta).
\tag{110}
$$

By Eq.(107), the conditional expected distance is derived as

$$
\mathbb{E}[||Y_{t+1} - Y_{t+1}^*||_2^2 | Y_t] = ||(\Lambda - \frac{L_t}{a_t a_{t+1}^*} k e^T)(Y_t - Y_{t+1}^*)||_2^2 + \frac{\mathbb{E}[(\tilde{h}_t - L_t)^2 | Y_t]\eta^4}{a_t^2}||k||_2^2.
\tag{111}
$$

Now we derive the upper bound of Eq.(111). We have known that if $t > 1 + \mathcal{T}(\alpha, \lambda, \eta, l, a_1)$, Eq.(104) shows the lower bound of $a_t$, Eq.(106) shows the value of $a_{t+1}^*$, therefore we have

$$
\frac{L_t \eta^2}{a_t a_{t+1}^*} < \sqrt{\frac{2L_t}{l}} \cdot 2\lambda\eta.
\tag{112}
$$

According to assumption 8, we can derive

$$
\sqrt{\frac{2L}{l}} < \frac{(1-\alpha)^2}{3}[\frac{1-\alpha^2}{\beta} - \frac{8}{1-\alpha}].
\tag{113}
$$

Combining with Eq.(112), (113), according to lemma C.2, omitting $\mathcal{O}(\lambda^2 \eta^2)$ we have

$$
||(\Lambda - \frac{L}{a_t a_{t+1}^*} k e^T)(Y_t - Y_{t+1}^*)||_2^2 \leq (1 - \frac{4\lambda\eta}{1-\alpha})||Y_t - Y_{t+1}^*||_2^2.
\tag{114}
$$

Assumption 5 in theorem shows the $\mathbb{E}[(\tilde{h}_t - L_t)^2 | \boldsymbol{Y}_t] < V$, so combining with lower bound of $a_t$ in Eq.(104) we can derive

$$\frac{\mathbb{E}[(\tilde{h}_t - L)^2 | \boldsymbol{Y}_t]}{a_t^2}\eta^4 < \frac{4V\eta^3\lambda}{l(1-\alpha)}. \tag{115}$$

Combining with Eq.(111), (114), (115), we can derive the upper bound of conditional expected distance in Eq.(111) given timestamp $t$

$$\mathbb{E}[||\boldsymbol{Y}_{t+1} - \boldsymbol{Y}_{t+1}^*||_2^2 | \boldsymbol{Y}_t] < (1 - \frac{4\lambda\eta}{1-\alpha})||\boldsymbol{Y}_t - \boldsymbol{Y}_{t+1}^*||_2^2 + \frac{4V\eta^3\lambda||\boldsymbol{k}||_2^2}{l(1-\alpha)}. \tag{116}$$

Then the upper bound of expected distance is

$$\mathbb{E}||\boldsymbol{Y}_{t+1} - \boldsymbol{Y}_{t+1}^*||_2^2 < (1 - \frac{4\lambda\eta}{1-\alpha})\mathbb{E}||\boldsymbol{Y}_t - \boldsymbol{Y}_{t+1}^*||_2^2 + \frac{4V\eta^3\lambda||\boldsymbol{k}||_2^2}{l(1-\alpha)}. \tag{117}$$

**Step 4, derive the relation between $\mathbb{E}||\boldsymbol{Y}_t - \boldsymbol{Y}_t^*||_2^2$ and $\mathbb{E}||\boldsymbol{Y}_{t+1} - \boldsymbol{Y}_{t+1}^*||_2^2$ like step 3 in theorem 1.**

Given time step $t$, we have

$$||\boldsymbol{Y}_t - \boldsymbol{Y}_{t+1}^*||_2 \le ||\boldsymbol{Y}_t - \boldsymbol{Y}_t^*||_2 + ||\boldsymbol{Y}_t^* - \boldsymbol{Y}_{t+1}^*||_2. \tag{118}$$

Here we need to estimate the upper bound of $||\boldsymbol{Y}_t^* - \boldsymbol{Y}_{t+1}^*||_2$. Recall $\boldsymbol{Y}_{t+1}^*$ satisfies

$$\boldsymbol{Y}_{t+1}^* = \Lambda\boldsymbol{Y}_{t+1}^* + \frac{L_t\eta^2}{(\boldsymbol{S}^T\boldsymbol{e})^T\boldsymbol{Y}_{t+1}^*}\boldsymbol{S}^{-1}\boldsymbol{e}, \tag{119}$$

, $\boldsymbol{k} = \boldsymbol{S}^{-1}\boldsymbol{e}$, by Eq.(106)

$$a_{t+1}^* = \tilde{a}_{t+1}^* + \tilde{b}_{t+1}^* + \tilde{c}_{t+1}^* = (\boldsymbol{S}^T\boldsymbol{e})^T\boldsymbol{Y}_{t+1}^* = \sqrt{\frac{L_t\eta}{\lambda(1-\alpha)(2 - \frac{\lambda\eta}{1+\alpha})}}. \tag{120}$$

Therefore, $\boldsymbol{Y}_{t+1}^*$ can be explicitly computed as

$$\boldsymbol{Y}_{t+1}^* = \eta\sqrt{L_t\lambda\eta(1-\alpha)(2 - \frac{\lambda\eta}{1+\alpha})}(\boldsymbol{I} - \Lambda)^{-1}\boldsymbol{k}. \tag{121}$$

Then we have

$$||\boldsymbol{Y}_t^* - \boldsymbol{Y}_{t+1}^*||_2^2 = (\sqrt{L_{t-1}} - \sqrt{L_t})^2\lambda\eta^3(1-\alpha)(2 - \frac{\lambda\eta}{1+\alpha})||(\boldsymbol{I} - \Lambda)^{-1}\boldsymbol{k}||_2^2. \tag{122}$$

By assumption 7, and using the fact that $L_t = \mathbb{E}[\tilde{h}_t | \boldsymbol{w}_t] > l$ we have

$$|\sqrt{L_{t-1}} - \sqrt{L_t}| = \frac{|L_{t-1} - L_t|}{\sqrt{L_{t-1}} + \sqrt{L_t}} \le \frac{|L_{t-1} - L_t|}{2\sqrt{l}} \le 2\sqrt{\frac{V(\lambda\eta)^3}{l}}. \tag{123}$$

On the other hand,

$$(\boldsymbol{I} - \Lambda)^{-1} = diag(\frac{1}{1-\lambda_1}, \frac{1}{1-\lambda_2}, \frac{1}{1-\lambda_3}), \tag{124}$$

where $\frac{1}{1-\lambda_3} < \frac{1}{1-\lambda_2} < \frac{1}{1-\lambda_1} = \frac{1-\alpha}{2\lambda\eta} + \mathcal{O}(\lambda\eta)$. Therefore

$$||(\boldsymbol{I} - \Lambda)^{-1}\boldsymbol{k}||_2^2 \le \frac{(1-\alpha)^2}{4\lambda^2\eta^2}||\boldsymbol{k}||_2^2 + \mathcal{O}(\lambda\eta) \tag{125}$$

Combining Eq.(122), (123), (125), we have

$$||\boldsymbol{Y}_t^* - \boldsymbol{Y}_{t+1}^*||_2^2 \le \frac{2V(1-\alpha)^3\lambda^2\eta^4}{l}||\boldsymbol{k}||_2^2 + \mathcal{O}(\lambda^3\eta^4) < \frac{2V\lambda^2\eta^4}{l(1-\alpha)^2}||\boldsymbol{k}||_2^2. \tag{126}$$

i.e.

$$||\boldsymbol{Y}_t^* - \boldsymbol{Y}_{t+1}^*||_2 < \sqrt{\frac{2V}{l}}\frac{\lambda\eta^2||\boldsymbol{k}||_2}{1-\alpha}. \tag{127}$$

Hence combining with Eq.(118), we have

$$||\boldsymbol{Y}_t - \boldsymbol{Y}_{t+1}^*||_2 \leq ||\boldsymbol{Y}_t - \boldsymbol{Y}_t^*||_2 + \sqrt{\frac{2V}{l}} \frac{\lambda \eta^2 ||\boldsymbol{k}||_2}{1 - \alpha}. \tag{128}$$

Now considering expectation w.r.t $\boldsymbol{w}_t$, we have

$$\mathbb{E}||\boldsymbol{Y}_t - \boldsymbol{Y}_{t+1}^*||_2^2 \leq \mathbb{E}[||\boldsymbol{Y}_t - \boldsymbol{Y}_t^*||_2 + \sqrt{\frac{2V}{l}} \frac{\lambda \eta^2 ||\boldsymbol{k}||_2}{1 - \alpha}]^2 \leq [\sqrt{\mathbb{E}||\boldsymbol{Y}_t - \boldsymbol{Y}_t^*||_2^2} + \sqrt{\frac{2V}{l}} \frac{\lambda \eta^2 ||\boldsymbol{k}||_2}{1 - \alpha}]^2. \tag{129}$$

If $\sqrt{\mathbb{E}||\boldsymbol{Y}_t - \boldsymbol{Y}_t^*||_2^2} \geq \sqrt{\frac{2V}{l}} \eta ||\boldsymbol{k}||_2$, then by Eq.(129), we have

$$\mathbb{E}||\boldsymbol{Y}_t - \boldsymbol{Y}_{t+1}^*||_2^2 \leq (1 + \frac{\lambda \eta}{1 - \alpha})^2 \mathbb{E}||\boldsymbol{Y}_t - \boldsymbol{Y}_t^*||_2^2. \tag{130}$$

Then combining with Eq.(117), we have

$$\mathbb{E}||\boldsymbol{Y}_{t+1} - \boldsymbol{Y}_{t+1}^*||_2^2 < (1 - \frac{2\lambda \eta}{1 - \alpha})\mathbb{E}||\boldsymbol{Y}_t - \boldsymbol{Y}_t^*||_2^2 + \frac{4V\eta^3 \lambda ||\boldsymbol{k}||_2^2}{l(1 - \alpha)}. \tag{131}$$

Else if $\sqrt{\mathbb{E}||\boldsymbol{Y}_t - \boldsymbol{Y}_t^*||_2^2} < \sqrt{\frac{2V}{l}} \eta ||\boldsymbol{k}||_2$, then by Eq.(129), we have

$$\mathbb{E}||\boldsymbol{Y}_t - \boldsymbol{Y}_{t+1}^*||_2^2 \leq (1 + \frac{\lambda \eta}{1 - \alpha})^2 \frac{2V\eta^2 ||\boldsymbol{k}||_2^2}{l}. \tag{132}$$

Then combining with Eq.(117), we have

$$\mathbb{E}||\boldsymbol{Y}_{t+1} - \boldsymbol{Y}_{t+1}^*||_2^2 \leq \frac{2V\eta^2 ||\boldsymbol{k}||_2^2}{l}. \tag{133}$$

Summing Eq.(131), (133), we have

$$\mathbb{E}||\boldsymbol{Y}_{t+1} - \boldsymbol{Y}_{t+1}^*||_2^2 \leq max\{(1 - \frac{2\lambda \eta}{1 - \alpha})\mathbb{E}||\boldsymbol{Y}_t - \boldsymbol{Y}_t^*||_2^2 + \frac{4V\eta^3 \lambda ||\boldsymbol{k}||_2^2}{l(1 - \alpha)}, \frac{2V\eta^2 ||\boldsymbol{k}||_2^2}{l}\}. \tag{134}$$

**Step 5 derive the upper bound of $\mathbb{E}||Y_{t+1} - Y_{t+1}^*||_2^2$ given $t$ like step 4 in theorem 1**

According to Eq.(134), by applying deduction method, when $t > \mathcal{T} = \mathcal{T}(\alpha, \lambda, \eta, l, a_1)$, we have

$$\mathbb{E}||\boldsymbol{Y}_t - \boldsymbol{Y}_t^*||_2^2 < (1 - \frac{2\lambda \eta}{1 - \alpha})^{t - \mathcal{T}} \mathbb{E}||\boldsymbol{Y}_{\mathcal{T}} - \boldsymbol{Y}_{\mathcal{T}}^*||_2^2 + \frac{2V\eta^2 ||\boldsymbol{k}||_2^2}{l}. \tag{135}$$

Similar to proof in SGD case, we need to take $t \leq \mathcal{T}(\alpha, \lambda, \eta, l, a_1)$ into account. To do that, we need to prove $\tilde{B} = max\{(1 - \frac{2\lambda \eta}{1 - \alpha})^{-t} \mathbb{E}||\boldsymbol{Y}_t - \boldsymbol{Y}_t^*||_2^2\}, t = 1, 2, ...\mathcal{T}(\alpha, \lambda, \eta, l, a_0)\}$ exists by showing $\mathbb{E}||\boldsymbol{Y}_t - \boldsymbol{Y}_t^*||_2^2$ is finite for $t = 1, ..., \mathcal{T}(\alpha, \lambda, \eta, l, a_0)$.

According to lemma C.1, combining with Eq.(91), we have

$$a_t = (\boldsymbol{S}^T \boldsymbol{e})^T \boldsymbol{Y}_t = \tilde{a}_t + \tilde{b}_t + \tilde{c}_t > \min(a_0, \sqrt{\frac{l\eta(1 - \alpha)}{4\lambda}}). \tag{136}$$

Hence $t = 0, 1, ..., \mathcal{T}(\lambda, \eta, l, a_1) - 1$, we have

$$
\begin{aligned}
\mathbb{E}||\boldsymbol{Y}_{t+1}||_2^2 &= \mathbb{E}||\boldsymbol{\Lambda Y}_t + \frac{\tilde{h}_t \eta^2}{(\boldsymbol{S}^T \boldsymbol{e})^T \boldsymbol{Y}_t} \boldsymbol{S}^{-1} \boldsymbol{e}||_2^2 \\
&\leq 2\mathbb{E}||\boldsymbol{\Lambda Y}_t||_2^2 + 2||\frac{\tilde{h}_t \eta^2}{(\boldsymbol{S}^T \boldsymbol{e})^T \boldsymbol{Y}_t} \boldsymbol{S}^{-1} \boldsymbol{e}||_2^2 \\
&\leq 2\lambda_1^2 \mathbb{E}||\boldsymbol{Y}_t||_2 + 2\frac{L\eta^2}{\min(a_0, \sqrt{\frac{l\eta(1 - \alpha)}{4\lambda}})} ||\boldsymbol{S}^{-1} \boldsymbol{e}||_2^2.
\end{aligned} \tag{137}
$$

By mathematical deduction, if $\mathbb{E}||\boldsymbol{Y}_0||_2^2 = ||\boldsymbol{Y}_0||_2^2$ is finite, then

$$\mathbb{E}||\boldsymbol{Y}_t||_2^2 \leq \infty, t = 1, ..., \mathcal{T}(\lambda, \eta, l, a_1). \tag{138}$$

Further more, by Cauchy inequality, we have

$$\mathbb{E}||\boldsymbol{Y}_t - \boldsymbol{Y}_t^*||_2^2 \leq 2\mathbb{E}||\boldsymbol{Y}_t||_2^2 + 2||\boldsymbol{Y}_t^*||_2^2 < \infty. \tag{139}$$

Hence $\tilde{B}$ exists, and we can prove $\forall t > 0$

$$\mathbb{E}||\boldsymbol{Y}_t - \boldsymbol{Y}_t^*||_2^2 < (1 - \frac{2\lambda\eta}{1-\alpha})^t \tilde{B} + \frac{2V\eta^2||\boldsymbol{k}||_2^2}{l}. \tag{140}$$

If $t > \mathcal{T}$, by Eq.(135), we have

$$\begin{aligned}
\mathbb{E}||\boldsymbol{Y}_t - \boldsymbol{Y}_t^*||_2^2 &< (1 - \frac{2\lambda\eta}{1-\alpha})^{t-\mathcal{T}}\mathbb{E}||\boldsymbol{Y}_\mathcal{T} - \boldsymbol{Y}_\mathcal{T}^*||_2^2 + \frac{2V\eta^2||\boldsymbol{k}||_2^2}{l} \\
&= (1 - \frac{2\lambda\eta}{1-\alpha})^t \cdot (1 - \frac{2\lambda\eta}{1-\alpha})^{-\mathcal{T}}\mathbb{E}||\boldsymbol{Y}_\mathcal{T} - \boldsymbol{Y}_\mathcal{T}^*||_2^2 + \frac{2V\eta^2||\boldsymbol{k}||_2^2}{l} \\
&\leq (1 - \frac{2\lambda\eta}{1-\alpha})^t \tilde{B} + \frac{2V\eta^2||\boldsymbol{k}||_2^2}{l}.
\end{aligned} \tag{141}$$

Else if $t \leq \mathcal{T}$, we have

$$\begin{aligned}
\mathbb{E}||\boldsymbol{Y}_t - \boldsymbol{Y}_t^*||_2^2 &= (1 - \frac{2\lambda\eta}{1-\alpha})^t \cdot (1 - \frac{2\lambda\eta}{1-\alpha})^{-t}\mathbb{E}||\boldsymbol{Y}_t - \boldsymbol{Y}_t^*||_2^2 \\
&\leq (1 - \frac{2\lambda\eta}{1-\alpha})^t \tilde{B} \\
&< (1 - \frac{2\lambda\eta}{1-\alpha})^t \tilde{B} + \frac{2V\eta^2||\boldsymbol{k}||_2^2}{l}
\end{aligned} \tag{142}$$

**Step 6, derive the upper bound of $\mathbb{E}(||\boldsymbol{w}_t||_2^2 - (w_t^*)^2)^2$ by $\mathbb{E}||\boldsymbol{Y}_t - \boldsymbol{Y}_t^*||_2^2$.**

Recall $||\boldsymbol{w}_t||_2^2 = a_t = \tilde{a}_t + \tilde{b}_t + \tilde{c}_t$, $(w_t^*)^2 = a_t^* = \tilde{a}_t^* + \tilde{b}_t^* + \tilde{c}_t^*$, therefore by Cauchy inequality, combining with Eq.(140)

$$\mathbb{E}[||\boldsymbol{w}_t||_2^2 - (w^*)^2]^2 \leq 3\mathbb{E}||\boldsymbol{Y}_t - \boldsymbol{Y}_t^*||_2^2 < 3(1 - \frac{2\lambda\eta}{1-\alpha})^t \tilde{B} + \frac{6V\eta^2||\boldsymbol{k}||_2^2}{l} \tag{143}$$

Set $B = 3\tilde{B}$, $C = 6||\boldsymbol{k}||_2^2$, note $\boldsymbol{k} = \boldsymbol{S}^{-1}\boldsymbol{e}$ only depends on $\alpha$, we can obtain

$$\mathbb{E}[||\boldsymbol{w}_t||_2^2 - (w^*)^2]^2 \leq (1 - \frac{2\lambda\eta}{1-\alpha})^t B + \frac{V\eta^2}{l}C. \tag{144}$$

$\square$

## C.4 Proof of Theorem 3

**Theorem 3.** *(Theoretical value of Angular Update) In SGD(SGDM) case, if assumptions in theorem 1(2) hold, $\eta^2 \ll 1$, $t$ is sufficiently large so that vanishing terms in Eq.(32), (68) can be omitted, then with probability at least $1 - \sqrt[3]{\frac{V}{L_t l}}$ we have*

$$|\Delta_t - \sqrt{\frac{2\lambda\eta}{1+\alpha}}| < \mathcal{O}(\sqrt[3]{\frac{V}{L_t l}}). \tag{145}$$

*In SGD case, $\alpha = 0$.*

*Proof.* According to Eq.(32), when $t$ is sufficiently large, we have (omit $\mathcal{O}(\lambda\eta^3)$

$$\mathbb{E}|||\boldsymbol{w}_t||_2^2 - (w_t^*)^2|^2 \leq \frac{2V\eta^2}{l}. \tag{146}$$

where $w_t^* = \sqrt[4]{L_{t-1}\eta/(2\lambda)}$. Then we have $\forall \delta > 0$,

$$\delta^2 Pr(|\,||\boldsymbol{w}_t||_2^2 - \sqrt{\frac{L_{t-1}\eta}{2\lambda}}| > \delta) \leq \mathbb{E}|\,||\boldsymbol{w}_t||_2^2 - \sqrt{\frac{L_{t-1}\eta}{2\lambda}}|^2 \leq \frac{2V\eta^2}{l}. \tag{147}$$

Eq.(147) implies $\forall \delta > 0$

$$Pr(|\,||\boldsymbol{w}_t||_2^2 - \sqrt{\frac{L_{t-1}\eta}{2\lambda}}| > \delta) \leq \frac{2V\eta^2}{l\delta^2}. \tag{148}$$

Then by Eq.(148), we have $\forall \delta > 0$

$$Pr(|\,||\boldsymbol{w}_t||_2^2 - \sqrt{\frac{L_{t-1}\eta}{2\lambda}}| < \delta, |\,||\boldsymbol{w}_{t+1}||_2^2 - \sqrt{\frac{L_t\eta}{2\lambda}}| < \delta)$$

$$\geq 1 - Pr(|\,||\boldsymbol{w}_t||_2^2 - \sqrt{\frac{L_{t-1}\eta}{2\lambda}}| > \delta) - Pr(|\,||\boldsymbol{w}_{t+1}||_2^2 - \sqrt{\frac{L_t\eta}{2\lambda}}| > \delta) \tag{149}$$

$$\geq 1 - \frac{4V\eta^2}{l\delta^2}.$$

On the other hand, by update manner of SGD in Eq.(31), we have

$$\langle \boldsymbol{w}_{t+1}, \boldsymbol{w}_t \rangle = (1 - \lambda\eta)||\boldsymbol{w}_t||_2^2, \tag{150}$$

then we can compute $\cos^2 \Delta_t^2$ by

$$\cos^2 \Delta_t = \frac{\langle \boldsymbol{w}_{t+1}, \boldsymbol{w}_t \rangle^2}{||\boldsymbol{w}_t||_2^2 \cdot ||\boldsymbol{w}_{t+1}||_2^2} = (1 - 2\lambda\eta)\frac{||\boldsymbol{w}_t||_2^2}{||\boldsymbol{w}_{t+1}||_2^2}. \tag{151}$$

According to the definition of $\Delta_t$, $\Delta_t \geq 0$, and $\Delta_t$ is very close to 0, we have

$$\Delta_t \approx \sin \Delta_t \tag{152}$$

$$= \sqrt{1 - \cos^2 \Delta_t} \tag{153}$$

$$= \sqrt{1 - (1 - \lambda\eta)^2 \frac{||\boldsymbol{w}_t||_2^2}{||\boldsymbol{w}_{t+1}||_2^2}} \tag{154}$$

$$= \sqrt{1 - (1 - \lambda\eta)^2 \frac{x_t}{x_{t+1}}} \tag{155}$$

where $x_t, x_{t+1}$ denotes $||\boldsymbol{w}_t||_2^2, ||\boldsymbol{w}_{t+1}||_2^2$ respectively as in Eq. (34). Assume $x_t, x_{t+1}$ are close to $x_{t+1}^* = \sqrt{\frac{L_t\eta}{2\lambda}}$, the second order Taylor expansion of Eq.(155) at $x_t = x_{t+1} = x_{t+1}^*$ is

$$\Delta_t = \sqrt{2\lambda\eta} + \frac{(1 - \lambda\eta)^2}{2\sqrt{2\lambda\eta}} \cdot \frac{1}{x_{t+1}^*} \cdot [(x_{t+1} - x_{t+1}^*) - (x_t - x_{t+1}^*)]$$

$$+ \mathcal{O}((x_{t+1} - x_{t+1}^*)^2 + (x_t - x_{t+1}^*)^2) \tag{156}$$

Then by Eq.(156), we have,

$$|\Delta_t - \sqrt{2\lambda\eta}| \leq \frac{1}{2\sqrt{L_t\eta}} \cdot (|x_{t+1} - x_{t+1}^*| + |x_t - x_t^*| + |x_t^* - x_{t+1}^*|)$$

$$+ \mathcal{O}(|x_{t+1} - x_{t+1}^*|^2 + |x_t - x_t^*|^2 + |x_t^* - x_{t+1}^*|^2) \tag{157}$$

By Eq.(45), we have

$$|x_t^* - x_{t+1}^*| < \sqrt{\frac{2V}{l}}\lambda\eta^2. \tag{158}$$

Assume $\exists \delta > 0, |x_{t+1} - x_{t+1}^*| < \delta, |x_t - x_t^*| < \delta$, then we have

$$|\Delta_t - \sqrt{2\lambda\eta}| \leq \frac{\delta}{\eta\sqrt{L_t}} + \sqrt{\frac{V}{2L_t\eta}}\lambda\eta + \mathcal{O}(\delta^2 + \frac{2V\lambda^2\eta^4}{l}) \tag{159}$$

Set $\delta = 2\eta\sqrt{L_t}\sqrt[3]{\frac{V}{L_t l}}$ in Eq.(149) and Eq.(159), with probability $1 - \sqrt[3]{\frac{V}{L_t l}}$, $|\|\boldsymbol{w}_t\|_2^2 - \sqrt{\frac{L_{t-1}\eta}{2\lambda}}| < \eta\sqrt{L_t}\sqrt[3]{\frac{2V}{Ll}}$, $|\|\boldsymbol{w}_{t+1}\|_2^2 - \sqrt{\frac{L_t\eta}{2\lambda}}| < \eta\sqrt{L_t}\sqrt[3]{\frac{2V}{L_t l}}$, therefore we have

$$|\Delta_t - \sqrt{2\lambda\eta}| \leq 2\sqrt[3]{\frac{V}{L_t l}} + \sqrt{\frac{V}{2L_t\eta}}\lambda\eta + \mathcal{O}(\eta^2(\frac{V}{L_t l})^{\frac{2}{3}}) = \mathcal{O}(\sqrt[3]{\frac{V}{L_t l}}). \tag{160}$$

In SGDM case, if $t$ is sufficiently large so that vanishing term in Eq.(140) can be omitted, we have

$$\mathbb{E}\|\boldsymbol{Y}_{t+1} - \boldsymbol{Y}_{t+1}^*\|_2^2 \leq \frac{2V\eta^2\|\boldsymbol{k}\|_2^2}{l}, \tag{161}$$

where $\boldsymbol{Y}_{t+1}$ is defined as Eq.(83) , $\boldsymbol{Y}_{t+1}^*$ is defined as Eq.(105), $\boldsymbol{k}$ is defined in Eq.(107). Similar to Eq.(148), $\forall \delta > 0$, we have

$$Pr(\|\boldsymbol{Y}_{t+1} - \boldsymbol{Y}_{t+1}^*\|_2 \geq \delta) \leq \frac{2V\eta^2\|\boldsymbol{k}\|_2^2}{l\delta^2}. \tag{162}$$

Hence we have

$$Pr(\|\boldsymbol{Y}_t - \boldsymbol{Y}_{t+1}^*\|_2 < \delta) \geq 1 - \frac{2V\eta^2\|\boldsymbol{k}\|_2^2}{l\delta^2}. \tag{163}$$

On the other hand, in SGDM case, the angular update can be computed by

$$\Delta_t = \sin(\Delta_t) \tag{164}$$

$$= \sqrt{1 - \cos^2\Delta_t} \tag{165}$$

$$= \sqrt{1 - \frac{\langle \boldsymbol{w}_t, \boldsymbol{w}_{t+1}\rangle^2}{\|\boldsymbol{w}_t\|_2^2 \cdot \|\boldsymbol{w}_{t+1}\|_2^2}} \tag{166}$$

$$= \sqrt{1 - \frac{b_{t+1}^2}{c_{t+1}a_{t+1}}}, \tag{167}$$

where $(a_{t+1}, b_{t+1}, c_{t+1}) = (\|\boldsymbol{w}_{t+1}\|_2^2, \langle \boldsymbol{w}_t, \boldsymbol{w}_{t+1}\rangle, \|\boldsymbol{w}_t\|_2^2)$. Assume $(a_{t+1}, b_{t+1}, c_{t+1})$ is close to $(a_{t+1}^*, b_{t+1}^*, c_{t+1}^*)$, where $\boldsymbol{X}_{t+1}^* = (a_{t+1}^*, b_{t+1}^*, c_{t+1}^*)^T$, $a_{t+1}^* = c_{t+1}^* = (\sqrt{\frac{L_t\eta}{\lambda(1-\alpha)(2-\frac{\lambda\eta}{1+\alpha})}}, b_{t+1}^* = \frac{1+\alpha-\lambda\eta}{1+\alpha}a_{t+1}^*$. Then the first order Taylor series expansion of Eq.(167) at $(a_{t+1} = a_{t+1}^*, b_{t+1} = b_{t+1}^*, c_{t+1} = c_{t+1}^*)$ is

$$\begin{aligned}\Delta_t &= \sqrt{1 - \frac{b_{t+1}^2}{a_{t+1}c_{t+1}}}\\ &= \sqrt{\frac{2\lambda\eta}{1+\alpha}} + \frac{\sqrt{1+\alpha}}{2\sqrt{2\lambda\eta}} \cdot (1 - \frac{\lambda\eta}{1+\alpha})^2\\ &\quad \cdot [\frac{a_{t+1} - a_{t+1}^*}{a_{t+1}^*} - \frac{2(b_{t+1} - b_{t+1}^*)}{b_{t+1}^*} + \frac{c_{t+1} - c_{t+1}^*}{c_{t+1}^*}]\\ &\quad + \mathcal{O}((a_{t+1} - a_{t+1}^*)^2 + (b_{t+1} - b_{t+1}^*)^2 + (c_{t+1} - c_{t+1}^*)^2).\end{aligned} \tag{168}$$

Now substituting $\boldsymbol{X}_{t+1} = (a_{t+1}, b_{t+1}, c_{t+1})$ with $\boldsymbol{Y}_{t+1} = (\tilde{a}_{t+1}, \tilde{b}_{t+1}, \tilde{c}_{t+1})$ defined in Eq.(83),(84),(85), (86), Eq.(156) can be explicitly rewritten as (computing process is straightforward and tedious so we omit it)

$$\begin{aligned}\Delta_t &= \sqrt{\frac{2\lambda\eta}{1+\alpha}} + \frac{\sqrt{1+\alpha}}{2\sqrt{2\lambda\eta}}(1 - \frac{\lambda\eta}{1+\alpha})^2\\ &\quad \cdot \frac{1}{a_{t+1}^*}\frac{(1-\alpha)^2}{\alpha^2}(\tilde{c}_{t+1} - \tilde{c}_{t+1}^*) + \mathcal{O}(\lambda\eta)\\ &\quad + \mathcal{O}(\|\boldsymbol{Y}_{t+1} - \boldsymbol{Y}_{t+1}^*\|_2^2),\end{aligned} \tag{169}$$

Omit $\mathcal{O}(\lambda\eta)$, if $\exists \delta > 0$, $||\boldsymbol{Y}_t - \boldsymbol{Y}^*||_2 < \delta$, we have

$$
\begin{aligned}
&|\Delta_t - \sqrt{\frac{2\lambda\eta}{1+\alpha}}| \\
&\leq \frac{\sqrt{1+\alpha}}{2\sqrt{2\lambda\eta}}(1 - \frac{\lambda\eta}{1+\alpha})^2 \cdot \frac{1}{a_{t+1}^*}\frac{(1-\alpha)^2}{\alpha^2}\delta + \mathcal{O}(\delta^2) \\
&\leq \frac{(1-\alpha)^2\sqrt{1-\alpha^2}}{2\alpha^2\eta\sqrt{L_t}}\delta + \mathcal{O}(\delta^2).
\end{aligned}
\tag{170}
$$

Now set $\delta = \sqrt{2L_t}\eta||\boldsymbol{k}||_2 \sqrt[3]{\frac{V}{L_t l}}$ in Eq.(163) and Eq.(170), with at least probability $1 - \sqrt[3]{\frac{V}{L_t l}}$, we have

$$
|\Delta_t - \sqrt{\frac{2\lambda\eta}{1+\alpha}}| \leq \frac{(1-\alpha)^2\sqrt{2(1-\alpha^2)}}{2\alpha^2}||\boldsymbol{k}||_2\sqrt[3]{\frac{V}{L_t l}} + \mathcal{O}(\eta^2(\frac{V}{L_t l})^{\frac{2}{3}}) \leq \mathcal{O}(\sqrt[3]{\frac{V}{L_t l}})
\tag{171}
$$

Summarize Eq.(160) and Eq.(171), for SGD/SGDM, if $T$ is sufficiently large, with at least $1 - \sqrt[3]{\frac{V}{L_t l}}$ probability, we have

$$
|\Delta_t - \sqrt{\frac{2\lambda\eta}{1+\alpha}}| \leq \mathcal{O}(\sqrt[3]{\frac{V}{L_t l}}).
\tag{172}
$$

$\square$

## C.5    Proof of Corollary 3.1

*Proof of Corollary 3.1.* In SGD case, by Eq.(34), we have

$$
||\boldsymbol{w}_{t+1}||_2^2 > (1 - 2\lambda\eta)||\boldsymbol{w}_t||_2^2,
\tag{173}
$$

which means

$$
||\boldsymbol{w}_{t+T}||_2^2 > (1 - 2\lambda\eta)^T||\boldsymbol{w}_t||_2^2.
\tag{174}
$$

On the other hand, we know that when $\eta$ is divided by $k$, $||\boldsymbol{w}_t||_2^2$ should be divided by $\sqrt{k}$ to reach the new equilibrium state, therefore we have

$$
\frac{||\boldsymbol{w}_{t+T}||^2}{||\boldsymbol{w}_t||^2} = \frac{1}{\sqrt{k}} > (1 - 2\lambda\eta)^T.
\tag{175}
$$

Since $\lambda\eta \ll 1$, $\log(1 - 2\lambda\eta) \approx -2\lambda\eta$, thus

$$
T > \frac{\log(k)}{4\lambda\eta}.
\tag{176}
$$

.

In SGDM case, by Eq.(101), we have

$$
||\boldsymbol{w}_{t+1}||_2^2 > (1 - \frac{2\lambda\eta}{1-\alpha})||\boldsymbol{w}_t||_2^2,
\tag{177}
$$

Similar to SGD case, we have

$$
T > \frac{\log(k)(1-\alpha)}{4\lambda\eta}.
\tag{178}
$$

$\square$

.

# D    Modeling Equilibrium by Stochastic Differential Equation

Li et al. (2020b) models the concept of equilibrium by Stochastic Differential Equation (SDE) in the continuous time limit. In this section, we discuss the connection between continuous form and discrete form of equilibrium.

In Li et al. (2020b), SGD can be approximated by

$$d\boldsymbol{w}_t = -\eta\lambda\boldsymbol{w}_t dt - \eta\nabla\mathcal{L}(\boldsymbol{w}_t)dt + \eta\boldsymbol{\Sigma}_{\boldsymbol{w}_t}^{1/2}d\boldsymbol{B}_t \tag{179}$$

where $\eta$, $\lambda$ denote learning rate and WD factor respectively in SGD. To properly approximate learning dynamics of normalized neural network, besides the condition that $\mathcal{L}$ is scale invariant w.r.t. $\boldsymbol{W}_t$, additional property of $\boldsymbol{\Sigma}_{\boldsymbol{w}_t}$ is required:

1. $\boldsymbol{\Sigma}_{c\boldsymbol{w}} = c^{-2}\boldsymbol{\Sigma}_{\boldsymbol{w}}$ for any $c > 0$;
2. $\boldsymbol{w}^T\boldsymbol{\Sigma}_{\boldsymbol{w}}\boldsymbol{w} = 0$;

The key results is

**Lemma D.1** (Li et al. (2020b)). *If $\sigma^2 \leq Tr(\Sigma_{\bar{\boldsymbol{w}}_t}) \leq (1 + \epsilon)\sigma^2$ for all $\boldsymbol{w}_t$ encountered in the trajectory, then*

$$\gamma_t = e^{-4\lambda_e t}\gamma_0 + (1 + \mathcal{O}(\epsilon))\frac{\sigma^2}{2\lambda_e}(1 - e^{-4\lambda_e t}), \tag{180}$$

*where $\gamma_t = ||\boldsymbol{w}_t||^4/\eta^2$, $\lambda_e = \lambda\eta$, $\bar{\boldsymbol{w}}_t = \boldsymbol{w}_t/||\boldsymbol{w}_t||_2$.*

Li et al. (2020b) states that the sign of equilibrium is the convergence of $\gamma_t$, and the convergence rate of equilibrium is determined by so called intrinsic learning rate $\lambda_e = \lambda\eta$.

Let's see the theoretical value of weight norm in continuous time limit, by Lemma D.1, we have

$$||\boldsymbol{w}_t||_2^4 = e^{-4\lambda\eta t}||\boldsymbol{w}_0||_2^4 + (1 + \mathcal{O}(\epsilon))\frac{\sigma^2\eta}{2\lambda}(1 - e^{-4\lambda_e t}). \tag{181}$$

When $\epsilon$ is small, Eq.(181) implies the theoretical value of weight norm in equilibrium is $\tilde{w}^* = \sqrt[4]{\sigma^2\eta/(2\lambda)}$. Comparing with the theoretical value of weight norm ($w_t^* = \sqrt[4]{L_{t-1}\eta/(2\lambda)}$) in our results (Theorem 1), the formulation of $\tilde{w}^*$ is very similar to $w_t^*$. The only difference is that $\tilde{w}^*$ relies on the scale of noise $\sigma^2$ while $w_t^*$ relies on expected square norm of unit gradient $L_{t-1} = \mathbb{E}||\boldsymbol{g}_{t-1}||_2^2$.

At first glance, the theoretical value $\tilde{w}^*$ of Lemma D.1 is counterintuitive: how is it possible that the scale of norm weight $||\boldsymbol{w}_t||_2$ only relies on the scale of noise at last? The formulation of $w^*$ and the proof of lemma D.1 all show that the evolving of weight norm is only driven by scale of unit gradient noise $Tr(\boldsymbol{\Sigma}_{\bar{\boldsymbol{w}}})$ and has no connection to full batch gradients $\nabla\mathcal{L}(\boldsymbol{w}_t)$. While our results in discrete form seems more reasonable: the theoretical value $w_t^*$ are determined by the behavior of unit gradient norm, which contains the information of both full batch unit gradients and noise of unit gradients, specifically we have

$$\mathbb{E}||\boldsymbol{g}_t||_2^2 = ||\mathbb{E}\boldsymbol{g}_t||_2^2 + \mathbb{E}||\boldsymbol{g}_t - \mathbb{E}\boldsymbol{g}_t||_2^2. \tag{182}$$

However, in many practical cases, the theoretical values of weight norm in continuous form and discrete form are consistent due to noisy dominated regime (Smith et al., 2020). Noisy dominated regime means the variance of gradients is much larger than the squared norm of full batch gradients, i.e.

$$||\mathbb{E}\boldsymbol{g}_t||_2^2 \ll \mathbb{E}||\boldsymbol{g}_t - \mathbb{E}\boldsymbol{g}_t||_2^2, \tag{183}$$

then we have

$$\mathbb{E}||\boldsymbol{g}_t||_2^2 \approx \mathbb{E}||\boldsymbol{g}_t - \mathbb{E}\boldsymbol{g}_t||_2^2, \tag{184}$$

which implies that in noisy dominated regime, $L_{t-1} \approx \sigma^2$. The evolving process of weight norm is indeed only determined by the scale of gradients' noise (see Figure 1).

Another connection is that in SDE settings, the convergence rate of weight norm is determined by $\lambda_e = \lambda\eta$. In other word, it will take

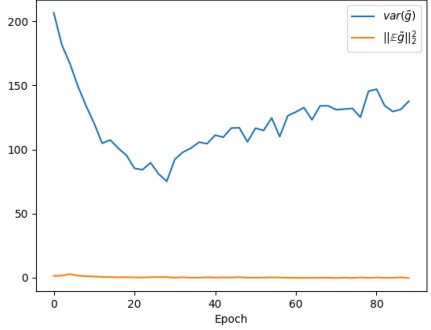

Figure 1: Expectation and variance of unit gradient norm. The data points are collected from the multi-stage Imagenet experiment in section 4.1 in main text.

$\mathcal{O}(1/(\lambda\eta))$ time to reach equilibrium. This result is consistent with the approaching rate in theorem 1.

Comparing with the discrete form of equilibrium, the weakness of continuous model is apparent: first, lemma D.1 does not take a more general situation into account, where the scale of gradients (including full batch gradients and its noise) can variate across the whole training process; second, it is hard to establish equilibrium state by SDE approximation when momentum method is applied. Therefore Li et al. (2020b) only provides conjectures and empirical observations to speculate equilibrium in SGDM. While in discrete settings, we can strictly prove the existence of equilibrium in SGDM case. Third, SDE approximation is not easy to depict the behavior of angular update (We exhibit the importance of angular update will be shown in next section). We notice recent works (Kunin et al., 2021; Tanaka & Kunin, 2021) use the term *angular velocity* in continuous model to describe the angular update in the discrete SGD and SGDM respectively. The theoretical results of angular velocity is consistent with our results. But their results about angular velocity still rely on the steady weight norm assumption (assume equilibrium state has been reached in advance), which weaken the significance of their results.

However, even current continuous models do not depict equilibrium phenomenon very well, continuous models own their unique advantages comparing with discrete models on studying the learning dynamics of complex function. So a more powerful theoretical tool or model which can better connect continuous models and discrete models is needed to study the effect SMD and equilibrium make to learning dynamics of normalized neural network.

## E    The Role of Momentum in Spherical Motion Dynamics

In the main text, we propose angular update to indicate equilibrium state of Spherical Motion Dynamics (SMD), derives its theoretical property in theorem, and give an example showing the behavior of angular update in SMD can significantly influence the training curve of neural network, and give a detailed description on the connection between angular update and the performance of normalized neural network.

If $\lambda\eta$ is fixed, increasing momentum factor will lead to smaller angular update (recall theorem 3), but resulting in larger training/testing loss. A reasonable interpretation is that SGDM is equivalent to SGD with larger learning rate (Smith et al., 2020; Mandt et al., 2017; Kidambi et al., 2018; Zhang et al., 2019a). Intuitively speaking, even though SGDM has smaller angular update, the existence of momentum makes its oscillation range much larger than its single step length on the loss landscape, preventing optimization trajectory from approaching the bottom of basin. In summary, the role of momentum is still not theoretically justified in non-convex problem, let alone taking equilibrium state of normalized neural network into account. We leave this as a future work.

## F    Supplemental details on "Pseduo Overfitting" by Spherical Motion Dynamics

First we introduce our experiment setting in section 4.3 in main text. Resnet18 (He et al., 2016) is trained on CIFAR10 (Krizhevsky & Geoffrey, 2009) for 200 epochs. Learning rate is divided by 5 at epoch 60, 120, 160 respectively; When using SGDM, learning rate is initialized as 0.1, WD factor is $5 \times 10^{-4}$, momentum factor is 0.9; When using Adam (Kingma & Ba, 2015), learning rate is initialized as $10^{-3}$, WD factor is $5 \times 10^{-4}$, first and second momentum factors are 0.9, 0.999 respectively.

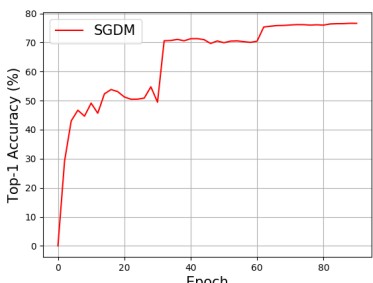

Figure 2: Test Accuracy (Top-1) of Resnet50 on Imagenet (same setting in Section 4.2 in main text).

We need to emphasize that though increasing angular update persistently can cause sub-optimum as our theorems suggest, the dropping test accuracy phenomenon is not common on various tasks. For example, when Resent50 is trained on Imagenet with multi-stage learning rate schedule, we cannot see apparent dropping test accuracy phenomenon (Figure 2). We can provide an intuitive explanation: to produce dropping test accuracy phenomenon, optimization trajectory needs to approach the bottom

of the basin rapidly with overly small angular update after learning rate decay, then increasing angular update will force it leaving away from the bottom of the basin, resulting in dropping test accuracy; However, if optimization trajectory moves to the bottom of the basin very slowly, then during the whole intermediate process to the new equilibrium state, optimization trajectory just moves towards the bottom of the basin. No extra "escaping" behavior occurs, no dropping test accuracy phenomenon can be seen. Therefore, we infer that dropping test accuracy phenomenon are more likely to be seen in simpler data experiments (like training resnet18 on CIFAR10), but hard to be seen in complex data experiments (like training resnet50 on Imagenet).

## G    Experiments on Synthetic Data

In this section we apply experiments on synthetic data to justify our claim that equilibrium is a dynamic state, and can be reached and maintain even if unit gradient norm constantly varies across the whole training process.

The proof of theorem implies square norm of weight is determined by the following iterative map:

$$x_{t+1} = (1 - 2\lambda\eta)x_t + \frac{L_t\eta^2}{x_t}, \tag{185}$$

where $\lambda, \eta \in (0, 1)$, $L_t$ denotes the square of unit gradient norm. Hence we simulate $x_t$ with different type of $\{L_t\}_{t=1}^{\infty}$. Results in Figure 3 shows as long as the local variance of square norm of unit gradient is not too much, and expectation of $L_t$ changes smoothly, weight norm can quickly converge to its theoretical value base on expectation of square norm of unit gradient.

We also simulate SGDM case by following iteration map

$$\boldsymbol{X}_{t+1} = \boldsymbol{A}\boldsymbol{X}_t + \frac{L_t\eta^2}{\boldsymbol{X}_t[0]} \cdot \boldsymbol{e}, \tag{186}$$

where $\boldsymbol{A}$, $\boldsymbol{X}_t$, $\boldsymbol{e}$ is defined as Eq.(72), (73), (74). Simulation results is shown in Figure 4.

### G.1    Complementary in Multi-Stage Learning Rate Schedule

In this section we present complementary results in Multi-Stage Learning Rate Schedule experiment.

The plots of weight norm (empirical and predicted values) in multi-learning rate stage is shown in Figure 5. We also present the test performance of resent50/maskrcnn on Imagenet/MSCOCO with multi-stage learning rate schedule mentioned in Section 5.2. We only provide complementary results for reference only. **We do not intend to prove the advantages or disadvantages of rescaling strategy here, it is beyond the discussion of this paper**.

|          | Top 1 Accuracy(%) |
|----------|-------------------|
| Standard | 76.25             |
| Rescale  | 76.27             |

Table 1: Performance of Resnet50 on Imagenet with multi-stage learning rate scheduler.

|          | $AP^{bbox}$ | $AP^{bbox}_{50}$ | $AP^{bbox}_{75}$ | $AP^{mask}$ | $AP^{mask}_{50}$ | $AP^{mask}_{7}$ |
|----------|-------------|------------------|------------------|-------------|------------------|-----------------|
| Standard | 39.25       | 58.88            | 42.95            | 35.16       | 56.09            | 37.48           |
| Rescale  | 38.38       | 57.98            | 42.31            | 34.49       | 55.27            | 36.71           |

Table 2: Performance of Resnet50 on Imagenet with multi-stage learning rate scheduler in Section.

### G.2    Rethinking Linear Scaling Principle in Spherical Motion Dynamics

In this section, we will discuss the effect of Linear Scaling Principle (LSP) under the view of SMD. Linear Scaling Principle is proposed by Goyal et al. (2017) to tune the learning rate $\eta$ with batch size $B$ by $\eta \propto B$. The intuition of LSP is if weights do not change too much within $k$ iterations, then $k$

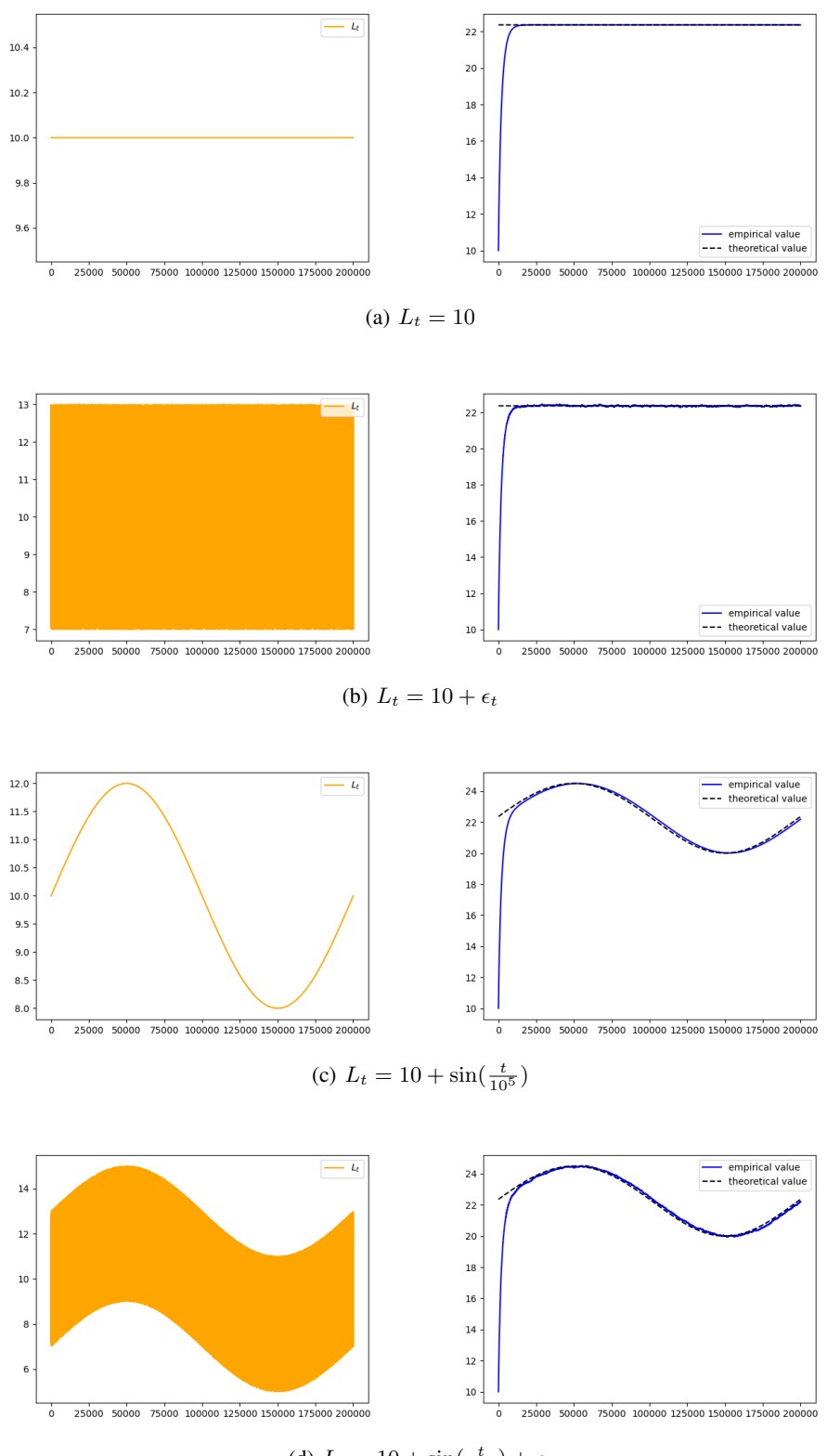

(a) $L_t = 10$

(b) $L_t = 10 + \epsilon_t$

(c) $L_t = 10 + \sin(\frac{t}{10^5})$

(d) $L_t = 10 + \sin(\frac{t}{10^5}) + \epsilon_t$

Figure 3: Simulation of SGD (Eq.(185)), $\eta = 0.1$, $\lambda = 0.001$, $x_0 = 10$, $\epsilon_t \sim \mathcal{U}(-3, 3)$. Orange lines represent the square of unit gradient norm; blue solid lines represent simulated value of weight norm square; black dashed lines represent theoretical value of weight norm square

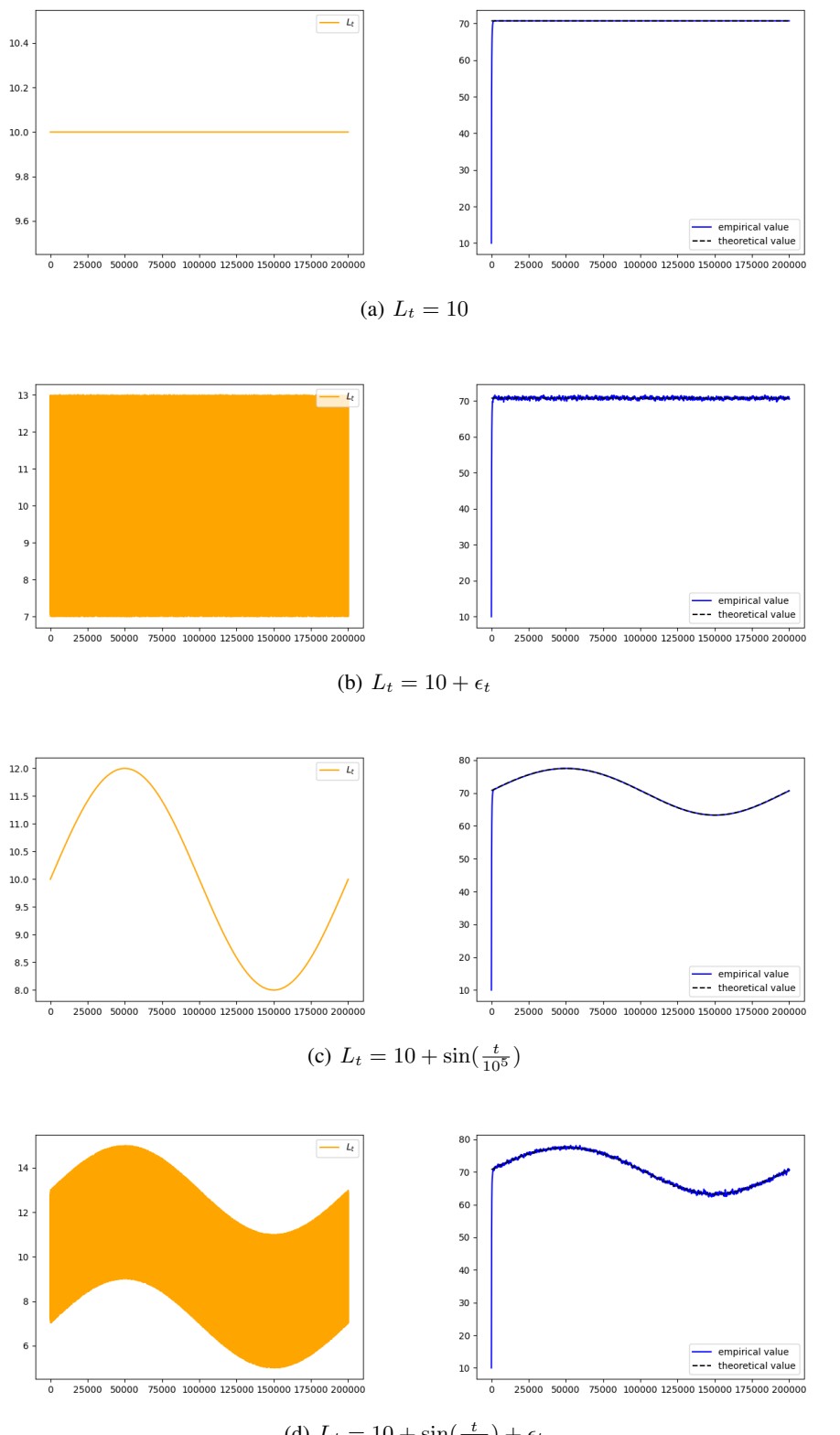

(a) $L_t = 10$

(b) $L_t = 10 + \epsilon_t$

(c) $L_t = 10 + \sin(\frac{t}{10^5})$

(d) $L_t = 10 + \sin(\frac{t}{10^5}) + \epsilon_t$

Figure 4: Simulation of SGDM (Eq.(186)), $\eta = 0.1$, $\lambda = 0.001$, $x_0 = 10$, $\epsilon_t \sim \mathcal{U}(-3, 3)$, $\alpha = 0.9$. Orange lines represent the square of unit gradient norm; blue solid lines represent simulated value of weight norm square; black dashed lines represent theoretical value of weight norm square

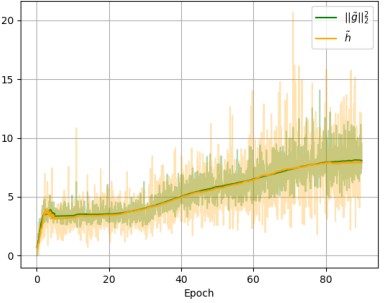

(a) $||\tilde{g}||_2^2$ and $\tilde{h}$ in standard settings

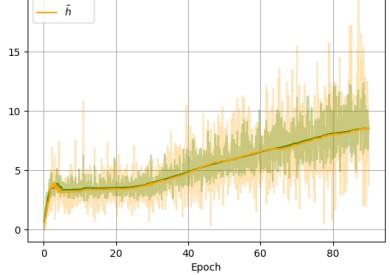

(b) $||\tilde{g}||_2^2$ and $\tilde{h}$ in rescaled settings

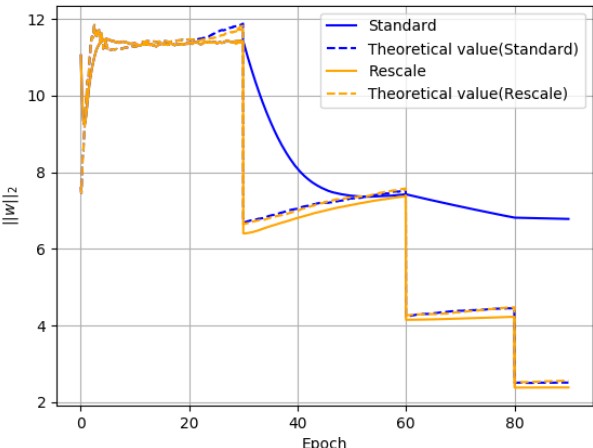

(c) Weight norm in Imagenet with theoretical value

Figure 5: $||\tilde{g}_t||_2^2$, $\tilde{h}_t$ and norm of weight from *layer.1.0.conv2* in Resnet50 backbone. (a),(b) present $||\tilde{g}||_2^2$, $\tilde{h}$ in multi-state learning rate schedule experiments discussed in Section 5.2. semitransparent line represents the raw value of $||\tilde{g}_t||_2^2$ and $\tilde{h}_t$, solid line represents the averaged value within consecutive 200 iterations to estimate the expectations $\mathbb{E}||\tilde{g}_t||_2^2$ and $\mathbb{E}\tilde{h}_t$. (c) presents the empirical value of weight norm and its theoretical value in standard and rescaled cases. The theoretical value is computed by estimated expectations $\mathbb{E}\tilde{h}_t$ in (a), (b) respectively.

iterations of SGD with learning rate $\eta$ and minibatch size $B$ (Eq.(187)) can be approximated by a single iteration of SGD with learning rate $k\eta$ and minibatch size $kB$ (Eq.(188):

$$\boldsymbol{w}_{t+k} = \boldsymbol{w}_t - \eta \sum_{j<k} (\frac{1}{B} \sum_{x \in \mathcal{B}_j} \frac{\partial \boldsymbol{L}}{\partial \boldsymbol{w}}\Big|_{\boldsymbol{w}_{t+j}, x} + \lambda \boldsymbol{w}_{t+j}), \tag{187}$$

$$\boldsymbol{w}_{t+1} = \boldsymbol{w}_t - k\eta (\frac{1}{kB} \sum_{j<k} \sum_{x \in \mathcal{B}_j} \frac{\partial \boldsymbol{L}}{\partial \boldsymbol{w}}\Big|_{\boldsymbol{w}_t, x} + \lambda \boldsymbol{w}_t). \tag{188}$$

Goyal et al. (2017) shows that combining with gradual warmup, LSP can enlarge the batch size up to $8192(256 \times 32)$ without severe degradation on ImageNet experiments.

LSP has been proven extremely effective in a wide range of applications. However, from the perspective of SMD, the angular update mostly relies on the pre-defined hyper-parameters, and it is hardly affected by batch size. To clarify the connection between LSP and SMD, we explore the learning dynamics of DNN with different batch size by conducting extensive experiments with ResNet50 on ImageNet, the training settings rigorously follow Goyal et al. (2017): momentum coefficient is $\alpha = 10^{-4}$; WD coefficient is $\lambda = 10^{-4}$; Batch size is denoted by $B$; learning rate is

initialized as $\frac{B}{256} \cdot 0.1$; Total training epoch is 90 epoch, and learning rate is divided by 10 at 30, 60, 80 epoch respectively.

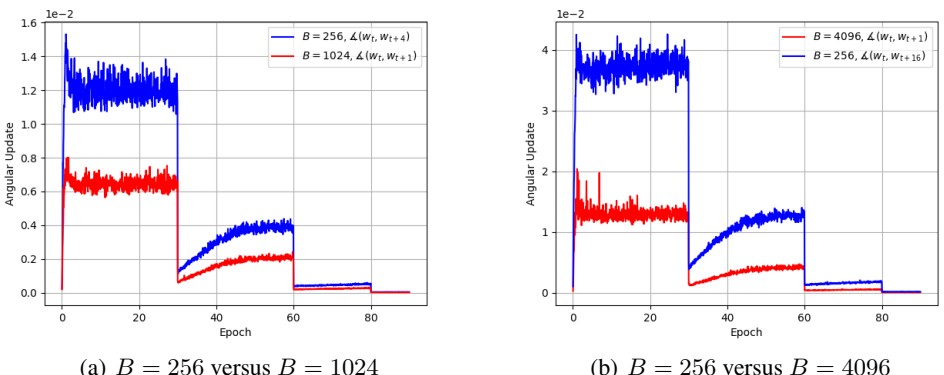

(a) $B = 256$ versus $B = 1024$      (b) $B = 256$ versus $B = 4096$

Figure 6: Angular update of weights from layer1.0.conv2 in ResNet50. The blue lines represent the angular update of weights within a single iteration in when batch settings is $B = 1024(4096)$; The red lines represent the accumulated angular update within 4(16) iterations in smaller batch setting($B = 256$).

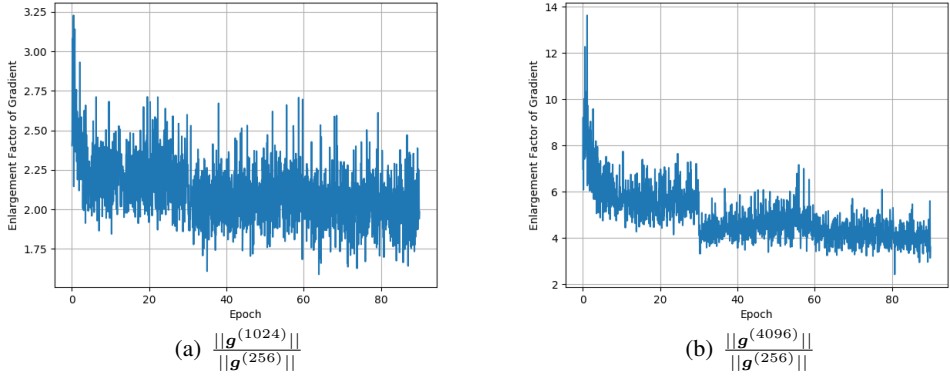

(a) $\frac{||\boldsymbol{g}^{(1024)}||}{||\boldsymbol{g}^{(256)}||}$      (b) $\frac{||\boldsymbol{g}^{(4096)}||}{||\boldsymbol{g}^{(256)}||}$

Figure 7: Enlargement ratio of gradients' norm of weights from layer1.0.conv2 when batch size increases . $||\boldsymbol{g}^{(k)}||$ represents the gradient's norm computed using $k$ samples(not average).

The results of experiments(Figure 6, 7) suggests that the assumption of LSP does not always hold in practice because of three reasons: first, the approximate equivalence between a single iteration in large batch setting, and multiple iterations in small batch setting can only hold in pure SGD formulation, but momentum method is far more commonly used; Second, according Theorem 2, the enlargement ratio of angular update is only determined by the increase factor of learning rate. Figure 6 shows in practice, the accumulated angular update $\measuredangle(\boldsymbol{w}_t, \boldsymbol{w}_{t+k})$ in small batch batch setting is much larger than angular update $\measuredangle(\boldsymbol{w}_t, \boldsymbol{w}_{t+1})$ of a single iteration in larger batch setting when using Linear Scaling Principle; Third, even in pure SGD cases, the enlargement of angular update still relies on the increase of learning rate, and has no obvious connection to the enlargement of gradient's norm when equilibrium condition is reached (see Figure 7).

In conclusion, though LSP usually works well in practical applications, SMD suggests we can find more sophisticated and reasonable schemes to tune the learning rate when batch size increases.

## G.3   Equilibrium condition with Different Network Structures

We also verify our theory on other commonly used network structures (MobileNet-V2 (Sandler et al., 2018), ShuffleNet-V2+ (Ma et al., 2018)) with standard training settings. The results is shown in Figure 8.

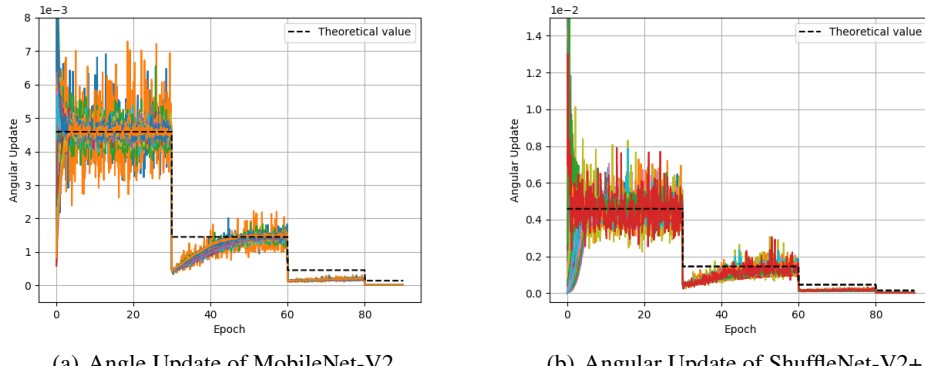

(a) Angle Update of MobileNet-V2    (b) Angular Update of ShuffleNet-V2+

Figure 8: The angular update $\mathbf{\Delta}_t$ of MobileNet-V2 (Sandler et al., 2018) and ShuffleNet-V2+ (Ma et al., 2018). The solid lines with different colors represent all scale-invariant weights from the model; The dash black line represents the theoretical value of angular update, which is computed by $\sqrt{\frac{2\lambda\eta}{1+\alpha}}$. Learning rate $\eta$ is initialized as $0.5$, and divided by $10$ at epoch $30$, $60$, $80$ respectively; WD coefficient $\lambda$ is $4 \times 10^{-5}$; Momentum parameter $\alpha$ is set as $0.9$.