# OpenReview forum: "Spherical Motion Dynamics: Learning Dynamics of Normalized Neural Network using SGD and Weight Decay"
_NeurIPS.cc/2021/Conference — NeurIPS 2021 Spotlight_

### Official Review · Reviewer_VeBT · 2021-07-04

**Rating:** 7
**Confidence:** 4

**Summary:**

In this paper, the dynamics of scale-invariant (SI) weights of normalized neural networks trained with SGD(M) + WD are studied. It is known that the scale-invariant parameters' intrinsic domain is a unit hypersphere, which the authors have effectively taken into account by introducing the "angular update" (AU) term, and the gradients w.r.t. SI parameters 1) scale inversely proportional to parameters norm and 2) are orthogonal to the radial direction in parameter space: the first property compels to introduce a unit gradient term and condition the dynamics of SI parameters on the intrinsic domain, the second one, together with weight decay, gives birth to the main concept of the paper &mdash; Spherical Motion Dynamics (SMD). Important attainment of this work is that, in contrast to the prior art, the notion of SMD equilibrium is not static (i.e., when $\Vert w_{t} \Vert_2 = \Vert w_{t+1} \Vert_2$) but represents a dynamic state when parameters norm oscillates around its theoretical value determined by hyperparameters and (generally varying) unit gradient norm. Interestingly, despite the dynamic nature of the equilibrium in terms of weight norm, the authors have shown that AU, in contrast, becomes approximately constant with its value only dependent on hyperparameters: learning rate, weight decay, and momentum factor. The authors corroborate their theoretical findings with experiments involving different computer vision tasks and networks architectures. They also provide an intriguing view on overfitting from the SMD perspective: when the learning rate drops, test error sometimes increases which could be interpreted as escaping from the local minimum due to an increase in the angular update.

**Limitations And Societal Impact:**

I appreciate that the authors have noted some limitations of their work and possible future research directions  (e.g., in Remark 3 and at the end of Section 4.2), however, I find several important flaws in this submission to be addressed (see Weaknesses above).

**Main Review:**

**Originality**
Despite a large volume of the related literature (well cited by the authors), I find the contributions of this paper novel and original. I put a special emphasis on the notion of angular update, dynamic interpretation of the equilibrium, formulation of the main results in discrete setting (instead of continuous SDE approximation), and generalization to the SGD with momentum case.

**Quality**
The submission is overall technically sound, although there are some concerns about the reasonableness of assumptions, experimental setup, and clarity of proofs (see Weaknesses below). The main claims are supported both by theoretical analysis and experimental results. All in all, I consider this submission a complete piece of work.

**Clarity**
Errata aside, this paper is clearly written and well organized. The authors provide enough details in the main text and appendix to reproduce the experiments and verify the claims.

**Significance**
The results of this work are important and provide valuable insight into the dynamics of learning of normalized neural networks significantly differing from previous works (see Originality). Notably, the authors state that the effective stepsize of SI weights will eventually be determined only by hyperparameters, regardless of any other attributes like shape, size, position, etc. While the results are indeed intriguing, their validity can be compromised by the limitations below.

**Weaknesses**
Below I provide a list of principal concerns about this submission. I would like the authors to address them primarily.
1. Here I question the assumptions made in Theorem 1, though similar concerns apply to Theorem 2 as well.
    1. Assumption 2 requires $V$ to be a *uniform* bound on the conditional squared unit gradient norm variance which can be a large value and lead to vacuous results (see below). Could this uniformness assumption on variance be weakened?
    2. Even more important, in Assumption 4 you set a *uniform* lower bound $l$ on the square unit gradient norm that, in addition, must not differ too much from the conditional expected squared unit gradient norm $L_t$ for *all* iterations. The second part does not allow the derived results to hold when the network converges to a minimum and the unit gradient norm tends to zero, so the lower bound is not valid anymore. I believe that might be the reason why the plots in Figure 2 depict only the beginning of training when the gradient norm is still high. Again, could this assumption be further weakened?
    3. Assumption 3 demands $L_t$ to change as $\mathcal{O}\left( (\lambda \eta)^{3/2} \right)$, i.e., very slowly since in assumption 1 you state that $\lambda \eta \ll 1$. I do not find this "smoothness of mean squared gradient norm dynamics" condition convincing enough, especially considering the possible heavy-tailness of the stochastic gradient noise yielding sudden jumps in gradient norm (see, e.g., https://arxiv.org/abs/1901.06053). Furthermore, a concurrent work (https://arxiv.org/abs/2106.15739) reveals that such jumps can in fact be regularly observed in normalized networks.
2. Considering that $V$ can be larger than one (see Figure 2a) and $L_t l$ is always below one (as a product of mean and min values), the value $\sqrt[3]{\frac{V}{L_tl}}$ can easily exceed one, hence the statement in Theorem 3 becomes vacuous: something can happen with probability larger than a negative value. I appreciate that the authors note in Remark 3 that their theoretical bound is too large compared to the empirical evidence, however, this issue may be of secondary importance. I expect the authors to reconsider the statement in Theorem 3. Would a formulation in terms of the *expected* angular update fit better (especially in light of the empirically observed behavior)?
3. In the process of reading, I encountered several questions about Figure 2, so some additional comments on it in the text would be appreciated.
    1. Why does the gradient norm start from zero and only grow? If I understand correctly, it does so due to the vanishing gradient effect for the first layers at the beginning of training and a limited number of iterations. However, usually, the total gradient norm (i.e., w.r.t. all network SI parameters) behaves differently, so does the behavior hold for all layers and the whole network? A brief comment on this would help a reader. While the lines for different layers are presented in Figure 3, more examples in the setting with a fixed learning rate in the appendix would also be beneficial.
    2. Does the behavior of angular updates remain stable after gradient norm saturation? Since the paper is about reaching a certain equilibrium state, adding at least one longer experiment would make the empirical results more convincing and even partially address the concerns about assumptions in Theorems 1 and 2.
4. About the “Overfitting” part:
    1. In line 331 of the main text, you write that "Adam has no equilibrium state like SGDM". Why so? Could you, please, comment more on this? A recent work https://arxiv.org/abs/2006.13382 could be relevant.
    2. The explanation in lines 324-328 of the main text seems vague. Rewriting this part using the intuition from Appendix F would make the text clearer.
    3. I suggest putting the bolded text from lines 493-494 of the appendix into the main text to avoid confusion that rescaling strategy helps generalization.

Minor concerns:
* When stating the main theorems, you use different iterations for defining certain important values in the main text and the appendix (compare, e.g., definitions of $w_t^*$ in line 148 of the main text and line 129 of the appendix). Also, heed the conditioning on $w_t$ when defining $L_t$ in lines 145 and 184 of the main text (compare, again, with lines 126 and 185 of the appendix, respectively). While I believe that the main claims are still valid, I would like the authors to fix these inconsistencies.
* I admit that I did not thoroughly follow all the proofs, however, I managed to find several inaccuracies. For instance, please explain what is going on in Eq. (38): e.g., what is $L$? Also, I believe you missed $2$ in Eq. (58), however, I still do not completely understand how it follows from equations (56) and (57), namely, why did you put $(1 - 2 \lambda \eta)$ in the denominator? I also suppose that the inequality signs must be $>$ in the second line of Eq. (149). Considering these and other issues, I advise you to revise your proofs more thoroughly.
* In Appendix E (and partially in the main text), you conjecture that larger angular updates lead to worse performance. However, I suppose that a more accurate formulation would be "*too* large angular updates *may* lead to worse performance" due to the following reasons: first, you consider only two AU values in Figure 3a-c in the appendix, which is not enough for making far-reaching conclusions; second, there exist plenty of works showing that larger learning rates can aid generalization by escaping from narrow minima (see, e.g., https://arxiv.org/abs/2003.03977 and references therein); third, the experiments with momentum, as noted in the text, contradict this conjecture (the explanation that momentum prevents optimization from reaching the bottom of the loss landscape is unconvincing since momentum is specifically tailored for improving convergence); finally, at the end of Appendix F you confirm that growth of AU is not always followed by test accuracy drop. I suggest relaxing the conjecture or provide more experiments supporting it.

---

**UPD**
After the discussion, I have found most of my concerns addressed and decided to increase my initial score. I also hope that the authors will improve their paper based on our discussion: formulate their theorems more accurately (e.g., modify the unnecessary uniformness of bounds), provide additional longer experiments with saturating gradients (or even periodic behavior), change the demonstration of some of their claims to evade misunderstanding (e.g., about "Adam has no equilibrium state like SGDM"), etc.

**Time Spent Reviewing:**

16-20

---

> ### Author Response · Authors · 2021-08-10
> **Response to Reviewer VeBT**
>
> Very appreciate for your affirmation and detailed comments, we will respond to your concerns one by one:
>
> > Weakness 1.1,1.2
>
> Yes, the uniformness assumption on both variance $V$ and lower bound $l$ can be weakened. Note that our results (theorem 1, theorem 2) are non-asymptotic, the approaching (to theoretical value) behavior of weight norm is ensured per iteration (see Eq.(43), Eq.(134) in appendix). It means we can use the “local” bound (upper bound of variance, and lower bound of unit gradient norm within several consecutive iterations) instead of “global” bound of the whole process to formulate the approaching behavior within a single iteration, then accumulate them. Mathematically speaking, we can substitute $V$, $l$ with a $t$-dependent sequential $V_t$, $l_t$. Then by figure 2(a)(d), even though $L_t$ is extremely small at beginning, $l_t$ and $V_t$ are also be small; when $L_t$ increases later, $l_t$ also increases. We take use of uniform bound $l$ and $V$ just in order to simplify the derivation, you can regard the approaching behavior depicted by theorem 1, 2 as a local process within one epoch. Of course we also can modify the theorem 1 and 2 to clarify the confusion.
>
> Besides, we need to clarify some misunderstanding of the reviewer:
>
> - There is no contradiction between “the network converges to a minimum” and “unit gradient norm does not tend to zero”. Please notice that the unit gradient norm is computed by a mini-batch gradient, “the network converges to a minimum” means the full-batch gradient is equal to zero, but the norm of mini-batch gradient still could be large due to large variance. Actually, it is really common that the mini-batch gradient norm is large even though the network has been well-trained (CIFAR10, Imagenet, COCO, etc). This phenomenon is called [Noisy dominated regime](https://arxiv.org/pdf/2006.15081.pdf), we have analyzed this in Sec D. in appendix. Of course, our theorems do not require that dynamics of SGD must be in noisy dominated regime to reach equilibrium. But imagenet experiments shown in Fig 2 are both in noisy dominated regime so they can satisfy the assumption of our theorems.
>
> - We do not only exhibit the beginning of training process. Please notice number of epochs in Fig 2 is exactly same as standard implementation of multi-state learning rate schedule. Besides, we also plots the unit gradient norm in multi-stage learning rate experiments (see Fig 7 (a), (b) in appendix). It can be seen that even the network has  approached to minimum by shrinking learning rate (angular update), the unit gradient norm still gets larger.
>
> > Weakness 1.3
>
> Please refer to the definition of $L_t$ in assumption 2) and 6) in theorem 1 and 2 respectively. We only assume the expected value of squared unit gradient norm changes smoothly (which means unit gradient norm owns finite second moment), and the squared unit gradient norm has lower bound. The assumptions of our theorem allow the occasional “jumping” phenomenon caused by large unit gradient norm. As for the possible heavy-tailness of the stochastic gradient noise, I notice both [Şimşekli et al., (2019)](https://arxiv.org/pdf/1901.06053.pdf) and [Lobacheva et al., (2021)](https://arxiv.org/pdf/2106.15739.pdf) do not conduct experiment in standard settings. [Lobacheva et al., (2021)](https://arxiv.org/pdf/2106.15739.pdf) also admits that  “jumping” phenomenon is uncommon with commonly used experiment settings. The unit gradient norm might have a proper upper bound  with standard training settings in practice. So our experiments do not meet jumping phenomenon as Fig 2 shows.
>
> > Weakness 2
>
> As we have responded to weakness 1.1,1.2, $V_t/(L_t * l_t)$ can be always smaller than 1 as Fig 2(a) (d) shows.  Of course $V_t/(L_t * l_t)$ is still too large (not necessarily close to zero). Rewritten the statement of theorem 3 into expected form cannot relax  the issue. In the original version of theorem 3, we presented the gap between expect value of angular update and its theoretical value, latter we found a severe flaw in the original statement of theorem 3: the gap is even larger than the theoretical value. This flaw is caused by too large bias term in Eq.(43), Eq.(134) in appendix (for weight norm, the scale of bias is relatively small; but it is too large when bounding the bias of angular update). The bias of angular update cannot be improved unless Eq.(43), Eq.(134) can be improved. We have not found a way to deal with this issue. We have to state result of theorem 3 in current version.
>
> > Weakness 3.1
>
> The (unit) gradient norm starts from zero only because we take use of zero-init trick mentioned by [Goyal et al., (2018)](https://arxiv.org/pdf/1706.02677.pdf). Specifically, **“for BN layers, the learnable scaling coefficient $\gamma$ is initialized to be 1, except for each residual block’s last BN where $\gamma$ is initialized to be 0”**, then the (unit) gradient norm of the convolutional layers in residual branches are all equal to zero at beginning. No convolutional layer really meets vanishing gradient issue.  If the zero-init trick is removed, the (unit) gradient norm will not be equal to zero at beginning. We have conducted the experiments without zero-init trick, we found this trick will not affect the performance of resnet50 on imagenet at all. We will add additional demonstrations in next version. As for the fact that unit gradient norm continues increasing across whole training process, it is indeed an interesting phenomenon, but it is beyond the discussion of our work. Our theorem cannot provide any insights on the global behavior of unit gradient norm. We can add additional results to see if the behavior of unit gradient norm is universal.
>
> > Weakness 3.2
>
> Both our theoretical results and experiments (Figure 3 in appendix) show angular update can reach its theoretical value and remains stable long before training loss/test accuracy remains stable, as long as the expectation of squared unit gradient norm changes smoothly. So we believe angular update can remain stable no matter whether gradient norm saturate. We will add longer experiment to enhance persuasion.
>
> > Weakness 4.1
>
> Sorry for the confusion, what we really want to express is that Adam does not have the “increasing angular update” phenomenon as SGD(M) (see Fig. 4(c)), so "dropping test accuracy" does not happen when Adam is used(see Fig. 4(b)). We admit it is haste to say Adam has no equilibrium state. Adam just does not have apparent SMD evolving pattern (see Sec A in appendix) like SGD(M), because the second order moving average momentum in Adam simultaneously break the perpendicular property and inverse-proportion property as Lemma 1 shows in main text. We will clarify the confusion in the main text, and add additional section to show the behavior of angular update in Adam in appendix. [Roburin et al., 2020](https://arxiv.org/pdf/2006.13382.pdf)  can be a good reference, thanks for your reminder.
>
> > Weakness 4.2, 4.3
>
> Thanks for your suggestions, we will improve it.
>
> > Minor concerns 1,2
>
> Sorry for these mistakes, it seems that we modify the proof in appendix but forgot modifying the main text and some notations in the proof accordingly. In first line of Eq.(38), $L_t$ should be $G_t$, $L$ should be $L_t$; It should be $2V\eta^2 / l$ in Eq.(58); In second line of Eq.(149), the inequality signs should be $>$. We will correct these mistakes.
>
> > Minor concern 3
>
> Thanks for your reminder, but there is a misunderstanding in your comment. Note that when we say “Larger angular update leads to higher training loss and lower test accuracy in equilibrium”, we doesn’t mean larger angular update lead to poor generalization performance. The assumption of our conjecture is that both optimization trajectories with larger angular update and smaller angular update will enter local basins with similar flatness (similar potential generalization performance), as Fig. 2 in appendix shows. Then the worse performance of larger angular update is only caused by under-fitting issues:  large angular update  in equilibrium will prevent the network from converging to the bottom of the local basin, so the trajectory has to oscillate around the local minimum, where the oscillation range is determined by scale of angular update. Those works which show "larger learning rate can lead to better generalization performance”, still have to shrink the learning rate at last to get the best performance. Otherwise, the training loss will be stuck at a high value with fixed large learning rate, as Fig. 3 (a) (d) in appendix show. There is no contradiction between between  the view in [Iyer et al., 2021](https://arxiv.org/pdf/2003.03977.pdf ) and our conjectures in Sec E.
>
> Besides, we need to remind the reviewer that the view “momentum can improve the convergence” is still controversial. Except for strong convex problem, the acceleration benefits of momentum can be theoretically proved in few situation. A popular view on role of momentum ([Zhang et al., 2019](https://proceedings.neurips.cc/paper/2019/file/e0eacd983971634327ae1819ea8b6214-Paper.pdf) ; [Smith et al., 2020](https://arxiv.org/pdf/2006.15081.pdf) ) is "SGDM is equivalent to SGD with larger learning rate”. [Smith et al., (2020)](https://arxiv.org/pdf/2006.15081.pdf) even verify this equivalence on cifar10 and imagenet experiments. Our experiment results provide further supportive evidence for this view (see Fig 3(d), (e) in appendix): momentum truly lead to worse performance (due to under-fitting issue).

---

> > ### Comment · Reviewer_VeBT · 2021-08-28
> > **Reply to authors response**
> >
> > Thank you for your response!
> >
> > Please, provide further comments on the following two points.
> >
> > >  Lobacheva et al., (2021) also admits that “jumping” phenomenon is uncommon with commonly used experiment settings.
> >
> > The experiments of Lobacheva et al. (2021) (see, e.g., Figure 6) demonstrate that jumps can be observed in reasonably practical settings as well if trained long enough. These jumps violate Assumption 3 since they indicate that $L_t$ &mdash; the expected value of squared unit gradient norm as figures in Lobacheva et al. (2021) suggest &mdash; might change rapidly. I assume that similar jumps could occur in your experiments in later epochs.
> >
> >
> > > As we have responded to weakness 1.1,1.2, $V_t / (L_t * l_t)$ can be always smaller than 1 as Fig 2(a) (d) shows.
> >
> > Unfortunately, I cannot agree here. Figure 2(a) does not demonstrate this; moreover, from epoch 20, the local gradient variance $V_t$ seems to be larger than 1 (as the range of the shaded area suggests) and continues growing, while $L_t * l_t$ is smaller than one by definition.
> >
> >
> > My comments on some other points.
> >
> >
> > > Of course we also can modify the theorem 1 and 2 to clarify the confusion.
> >
> > Yes, this would be appreciated to avoid such confusion. Or at least you could add a remark to clarify this point in the main text as you did in your response.
> >
> >
> > > We can add additional results to see if the behavior of unit gradient norm is universal.
> >
> > That would be very interesting, and I would be glad to see this in the supplementary materials.
> >
> >
> > > We will add longer experiment to enhance persuasion.
> >
> > Yes, please, add one.
> >
> >
> > > We will clarify the confusion in the main text, and add additional section to show the behavior of angular update in Adam in appendix.
> >
> > This would also be appreciated, thank you.
> >
> >
> > > Thanks for your reminder, but there is a misunderstanding in your comment.
> >
> > I would like to clarify that I fully understood your point. I just wanted to suggest relaxing your formulations since there is an alternative point of view on the impact of learning rate on model performance, depending on what part of the training process we consider. Of course, you need to shrink the learning rate *at the end* of the training to converge. However, *starting* from small learning rate values may harm final test accuracy.

---

> > > ### Author Response · Authors · 2021-08-29
> > > **Further response**
> > >
> > > Thanks for your reply.
> > >
> > > > The experiments of Lobacheva et al. (2021) (see, e.g., Figure 6) demonstrate that jumps can be observed in reasonably practical settings as well if trained long enough.
> > >
> > > In fact we **cannot** reproduce the experimental results in Lobacheva et al., (2021)  (we mainly focus on the resnet18 on cifar100 since it seems standard) in our  experiments, so we are waiting for them to release the code. We also run our experiments on CIFAR10 for additional 200 epochs (400 epochs totally), no periodic jumping phenomenon happens. But we manage to exhibit this periodic jumping phenomenon in a simple experiment: assume $A \in \mathbb{R}^{p \times p}$ is a positive definite symmetric matrix, then we want to find the optimum of $ \arg \min_{x \in \mathbb{R}^p} f(x) = x^T A x / ||x||_2^2$. We solve it using GD(gradient descent)and GD with WD(weight decay) respectively. Apparently, the optimum of this optimization problem is the eigenvector of $A$ corresponding to the least eigenvalue, GD can easily find it with fixed learning rate, but GD with WD experiment will meet periodic jumping phenomenon as the curve of $f(x_t) shows.
> > >
> > > The cause is: GD with WD can easily find the optimum as pure GD does at the beginning, the unit gradients is vanishing (close to zero) since trajectory is very close to the optimum. Then WD dominate the whole gradient descent part, weight norm constantly decreasing to 0. Note even the unit gradient is very close to $0$, but it is still not equal to $0$. Recall the inverse proportion relationship between gradient and weight norm (Eq.(6) in our paper), then after a critical iteration, **due to the numerical error of division operator**, the gradient will suddenly becomes extremely large, leading to extremely large angular update, therefore the trajectory will leave away from the optimum at a singer iteration, resulting in jumping phenomenon.
> > >
> > > The cause of periodic jumping phenomenon in our synthetic experiment is a little different from the interpretations in Lobacheva et al., (2021). But the essential cause is same: unit gradient norm is close to zero for a long time during the training. Combining with our experiments and the experimental results in Lobacheva et al., (2021), we derive three possible reasons why Lobacheva et al., (2021) can exhibit periodic phenomenon on cifar10/cifar100, while our experiments cannot.
> > >
> > > 1.  We take use of SGD instead of GD. When optimization trajectory is close to optimum, its full gradient will be close 0, but its mini-batch gradient will not necessarily be. We have stated in Sec D in appendix that with commonly settings, the training process will have “noise dominated regime”, where the variance of the stochastic gradients will dominate the norm of stochastic gradients. So the unit gradient norm we use in our theorem could be have large lower bound;  The discussion on data augmentation in Lobacheva et al., (2021) can also support this reason.
> > > 2. We take use of a suitable models for the datasets. We notice  Lobacheva et al., (2021) take use of a three-layer convolution network in CIFAR10 experiments, and freeze all unnormalized parts of Resnet18 in CIFAR100 experiment. So we suspect that  these manually adjusted networks all have sharp local optimum, their unit gradient norm doesn’t change smoothly. We think  a network with good generalization performance should have flat minimum, in which its unit gradient norm changes smoothly.
> > > 3. We use proper hyper-parameters in our experiments. We also notice in Lobacheva et al., (2021), when $\lambda \eta$ is too small at beginning, periodic jumping phenomena are more likely to occur. In our theorem  smaller $\lambda \eta$  implies smaller angular update in equilibrium, i.e the single step on the unit sphere is smaller, so we guess with too small $\lambda \eta$ at the very beginning, the optimum are more likely to enter a sharp local minimum, resulting in vanishing unit gradient; If  $\lambda \eta$ is suitable, the optimization trajectory can enter a flat basin on the loss landscape, then shrinking angular update by shrinking $\lambda \eta$, the trajectory is allowed to approach the bottom of basin and get better performance, but the jumping phenomenon won’t happen with smaller $\lambda \eta$ in this condition.
> > >
> > > In summary, we think the periodic jumping phenomenon implies that the optimization trajectory has been trapped in a sharp local minimum with poor performance. And the problem is mainly caused by inappropriate neural network structures or inappropriate training settings. In practice, periodic jumping phenomenon will not be observed with suitable experimental settings.
> > >
> > > > Figure 2(a) does not demonstrate this; moreover, from epoch 20, the local gradient variance $V_t$ seems to be larger than 1 (as the range of the shaded area suggests) and continues growing, while $L_t ∗ l_t$ is smaller than one by definition.
> > >
> > > We are sorry we do not get your point. As far as we are concerned, Figure 2(a) shows that, at epoch 20, according to shaded lines, we have $V_t < 2*2 = 4$,  according to solid line, we have $l_t > 5$,  so $L_t > 6$, then $V_t / (L_t * l_t ) < 4/30 \approx 0.13 < 1$. We do not understand why you say $L_t ∗ l_t < 1$.
> > >
> > > >  I just wanted to suggest relaxing your formulations since there is an alternative point of view on the impact of learning rate on model performance, depending on what part of the training process we consider.
> > >
> > > That is exactly what we want to show, thank you. We will change our demonstration to clarify this confusion.

---

> > > > ### Comment · Reviewer_VeBT · 2021-08-30
> > > > **Further response to authors**
> > > >
> > > > Thank you for your response!
> > > >
> > > > I really appreciate the work you have done to investigate further the periodic behavior of the training dynamics reported by Lobacheva et al. (2021) and the alternative interpretation of their results (especially point 3); however, the first two points are questionable. First, Lobacheva et al. (2021) conduct all of their experiments in the main text using SGD (they use GD only in Appendix). Second, they also provide results with common ResNet18 architecture (with all parameters trained) on CIFAR100, e.g., in Figure 1. Based on Figure 6 (right), I would also say that the networks exhibiting periodic behavior can achieve rather good generalization. All in all, we should indeed wait for the code release.
> > > >
> > > > After you explained your point about Figure 2 and Theorem 3, I realized my reasoning was flawed. Now I am convinced that you were right.
> > > >
> > > > I think that most of my concerns are addressed now and will increase my score. I hope that our discussion will be beneficial to your paper.

---

### Official Review · Reviewer_uKSg · 2021-07-14

**Rating:** 7
**Confidence:** 3

**Summary:**

This paper investigates the time-evolution of weights in normalized neural networks (e.g. one in which every layer has batch-norm applied). It proposes theorems regarding convergence to equilibrium, in which the norm of the weights approaches a fixed value. Compared to prior work, which established the fixed value, this work additionally gives the approach rate, and the scale of the variance due to stochastic gradients. The paper introduces the angular update parameter, and shows this approaches a fixed value. The paper emphasizes that equilibrium is not reached immediately, but is approached dynamically. It also asserts that equilibrium is not about convergence of stochastic oscillations to zero, but instead convergence to a fixed value, up to oscillations. The theoretical results are then investigated empirically, showing agreement with theoretical results for the weight norm and angular update, and the dynamical approach to equilibrium.

**Limitations And Societal Impact:**

The paper has no discussions of limitations (technical or otherwise), nor of societal impact. Since this is mostly theoretical, the societal impacts are not direct, so this seems ok. But I would like to see discussion of technical limitations - for example, what about networks in which not all weights are scale-invariant (just a subset), eg ones where only a few types of layer are normalzed.

**Main Review:**

**Originality**

As stated in the paper, the fixed value of the weight norm was already found in van Laarhoven, so the extra piece this work adds here is the rate of approach and the scale of the stochastic variance. It also has a corresponding result for momentum. For the angular update, the equilibrium fixed value had already been found Chiley and other papers (see also Kunin). The added bit here seems to also be the momentum term. I am not aware of whether the other papers conducted a similar experimental verification of their results. It would be good to clarify if Kunin is to be treated as concurrent or as prior work. The related work section was vague on how this relates to recent work, lines 124-126.

**Quality**

I have not checked the proofs of the statements in the appendix, but the experiments seem to be of high quality overall. My main comment here is that fig 4b regarding the drop in test accuracy could be strengthened by running more seeds etc as this might just be a random variation.

**Clarity**

The submission is clearly written and well organized. The weakest section is 4.2, where the motivation and explanation of the experiment is not that clear. Also, I’m not sure why a new term ‘spherical motion dynamics’ is necessary, and may also be misleading as the motion may sometimes not be restricted to a sphere (if I've understood correctly, when out of equilibrium, the motion is not restricted to the sphere).

**Significance**

If the main original contributions are the rate of approach and scale of variance, why are these useful? This is not explained or motivated. Similarly for the extension of existing results to the case of momentum. It would be good to explain further why this is useful to a practitioner.


-----

**Edit**:

I have updated my score from 6 to 7 based on the originality of this work being more clear to me, following author responses. I now see that the value to which the weight norm square tends to in equilibrium is not a constant value (as in prior/concurrent works), but is a non-constant value, due to the expected squared norm of the gradients being non-constant. This is also empirically demonstrated in Fig 2b,e.

The author reply also assured me how the clarity of the work will be improved in a possible camera-ready version.

**Time Spent Reviewing:**

5

---

> ### Author Response · Authors · 2021-08-10
> **Response to Reviewer uKSg**
>
> Thanks for your comments, we will response to your concerns one by one.
>
> **Originality**
>
> As we state in lines 53-69, all previous work (mentioned in lines 53-69) ahead our work did not find “equilibrium” phenomenon, they just propose a conjecture based on an actually wrong assumption (weight norm is fixed by optimization). That is the reason why they can neither theoretically prove their conjecture nor exhibit the phenomenon in experiments.
>
> As for more recent work ([Li et al., 2020](https://arxiv.org/pdf/2010.02916.pdf); [Tanaka and Kunin, 2021]( https://arxiv.org/pdf/2105.02716.pdf ); [Kunin et al., 2021](https://arxiv.org/pdf/2012.04728.pdf) ), actually these three works are all concurrent works. They all depict “equilibrium” concept by continuous model(SDE approximation, gradient flow). Due to the gap between continuous model and discrete formulation of SGD in practice, [Li et al., 2020]( https://arxiv.org/pdf/2010.02916.pdf ), [Tanaka and Kunin, 2021]( https://arxiv.org/pdf/2105.02716.pdf ), [Kunin et al., 2021](https://arxiv.org/pdf/2012.04728.pdf) can only provide indirect evidence of equilibrium. Our work is the only one which can give concrete and precise definition on equilibrium and prove its existence by both theorems and experiments.
>
> Another important novelty of our work is that we extend the concept "equilibrium" to "Spherical Motion Dynamics (SMD)", the approaching rate to equilibrium and theoretical value of angular update are both characteristics of SMD. Using SMD we can interpret something interesting phenomenon as Sec 4.2 shows. Due to the limitation of length, we cannot demonstrate the geometric meaning of angular update, and "dropping test accuracy" phenomenon in details in main text, so we place  the thorough demonstration in appendix, please refer to Sec E, F in appendix.
>
> **Quality**
>
> We will add more trials to verify the “dropping test accuracy” phenomenon is deterministic in standard SGD(M) training settings. But a more strong evidence is that the “dropping test accuracy” phenomenon can be seen in many irrelevant works, for example: Figure 2 in [Li and Arora, (2019)](https://arxiv.org/pdf/1910.07454.pdf), Figure 4 in [Zhang et al., (2019)](https://arxiv.org/pdf/1810.12281.pdf).  The “dropping test accuracy” phenomenon has been exhibited in previous works long before, but it was not taken seriously then.
>
> **Clarity**
>
> The contents in Sec 4.2 (“dropping test accuracy” phenomenon caused by SMD) is used to demonstrate how SMD affects learning dynamics of normalized neural networks in a way different from traditional machine learning theorem. Due to limitation of length, we have to place the thorough analysis about section 4.2 in appendix. Please refer to Sec E, F in appendix for more details.
>
> The term “spherical motion dynamics” (SMD) is inspired by the newtonian analysis of orbital motion in physics. It does not refer to the motion on a fixed sphere, it refer to the whole process caused by joint effect of BN, SGD and WD, for example, the process ahead of equilibrium, or the intermediate process between two different equilibrium states. “Equilibrium” is just a special state of SMD. As we demonstrate in Sec 4.1, equilibrium state are not necessarily reached if the total number of iterations is not enough, but the effect of SMD always exists.
>
> **Significance**
>
> Again, due to the limitation of length, we can only briefly demonstrate the significance of our work in main text lines 100-107, 237-242, 344-348. We leave more detailed demonstration to show our motivation and experiments results  in Sec D, E, F in appendix. We think the demonstrations in Sec E, F can be useful for practitioners to tune learning rate and weight decay factor, or treat “dropping test accuracy” phenomenon properly in practice.
>
> **Limitations**
>
> We will add more demonstrations on the limitations of works in next version.

---

> > ### Comment · Reviewer_uKSg · 2021-08-16
> > **reply to author response**
> >
> > Thanks for the reply, and for taking into account my review.
> >
> > **Originality**:
> > I recommend you are more clear about your work being concurrent with these three works, and that in the main text you clearly discuss your differences to these which use other approximations. As you seem to arrive at similar conclusions, what is the advantage of your analysis? This needs to be very concrete - I found the appendix D vague. Why does it matter if the scale of the gradients varies? Does this happen in practice? What does this predict wrongly, that your analysis captures? Your discussion could also be more balanced - you only discuss disadvantages of other works assumptions, without giving any detail about the disadvantages of your own assumptions. I would want to see more discussion of this in order for me to raise my score.
> >
> > **Quality**:
> > I will await more results from trials. It is not enough to point at other work here - I would like to see all experiments within this paper shown to have statistical significance, in this case, running more seeds, checking on other models etc.
> >
> > **Clarity**:
> > Section 4.2: What traditional machine learning theorem(s) are you comparing to here? I just wanted to have a clearer motivation for your experiment here. I found appendices E and F to be poorly written. It's not clear what is a conjecture and what is results in those appendices. I remain unconvinced about the clarity of this section, and the response has not helped.
> >
> > I also remain unconvinced about introducing a new term "spherical motion dynamics" for motion which is not spherical, and does not make clear that it is just SGDM with WD on a normalised neural network - why does this warrant a new term, the naming of which is misleading?
> >
> > **Significance**:
> > I wanted the reply here to be about what I consider the novel findings (rate of approach and scale of variance, and including momentum), and why they specifically are important and useful to the practitioner. Instead it just referred back to the paper, which only speaks about known parameters, such as angular update, but not it's rate of approach. I conclude that the authors are not aware of the usefulness of the novel aspects for the practitioner.

---

> > > ### Author Response · Authors · 2021-08-17
> > > **Further response to Reviewer uKSg (1/2)**
> > >
> > > Thanks for your further comments. We will response to your concerns one by one.
> > >
> > > **Originality**:
> > >
> > > >  As you seem to arrive at similar conclusions, what is the advantage of your analysis?
> > >
> > > We do not agree with your assessment that we *“seem to arrive similar conclusion”* as the other three works ([Li et al., 2020](https://arxiv.org/pdf/2010.02916.pdf), [Tanaka and Kunin, 2021]( https://arxiv.org/pdf/2105.02716.pdf ), [Kunin et al., 2021](https://arxiv.org/pdf/2012.04728.pdf)).  Our work and [Li et al. (2020)](https://arxiv.org/pdf/2010.02916.pdf), [Tanaka and Kunin (2021)]( https://arxiv.org/pdf/2105.02716.pdf ), [Kunin et al. (2021)](https://arxiv.org/pdf/2012.04728.pdf) do not just use different assumptions, but also obtain different conclusions. Our reasons are below:
> > >
> > > 1.  First, Kunin’s two works, [Tanaka and Kunin (2021)]( https://arxiv.org/pdf/2105.02716.pdf ), [Kunin et al. (2021)](https://arxiv.org/pdf/2012.04728.pdf), actually does not provide any new results on “equilibrium”, they still take use of “stationary” assumption (assuming weight norm has been fixed in advance). Their results on equilibrium can only be regarded as a continuous version of the very first literature ([van Laarhoven, 2017](https://arxiv.org/pdf/1706.05350.pdf)). While our work focus on interpreting how equilibrium can be reached, and affect the learning dynamics of normalized DNN on generalization performance. We need to remind the reviewer that the main contribution of  [Tanaka and Kunin (2021)]( https://arxiv.org/pdf/2105.02716.pdf ), [Kunin et al. (2021)](https://arxiv.org/pdf/2012.04728.pdf) do not concern equilibrium, they just use equilibrium concept as an example to demonstrate the effectiveness of their proposed theoretical model. It is unfair for us to conclude that our work has similar conclusion as [Tanaka and Kunin (2021)]( https://arxiv.org/pdf/2105.02716.pdf ), [Kunin et al. (2021)](https://arxiv.org/pdf/2012.04728.pdf).
> > >
> > > 2. As for [Li et al. (2020)](https://arxiv.org/pdf/2010.02916.pdf), it is the only work which systematically analyze the cause of equilibrium and its effect to learning dynamics of DNN besides our work. But their major conclusion/conjectures (see Sec 5.2 in [Li et al. (2020)](https://arxiv.org/pdf/2010.02916.pdf) ) drawn from their theory are actually refuted by our results:
> > >     1.  they claim the sign of equilibrium is convergence of (expected) weight norm. But we prove equilibrium can be reached even weight norm doesn’t converge to a fixed value across the whole training procedure (theorem 1, 2 in main text). We also exhibit experimental results to verify our theorem (see Fig 2 b, e in main text). The equilibrium state is reached within 20 epochs in our experiments, but
> > >     2.  They conjecture that performance(training loss/test accuracy) of DNN remains stable in equilibrium state because DNN converges to the equilibrium distribution. But we show equilibrium state of SMD is not equivalent to stationary distribution of DNN itself. Equilibrium state can be reached long before training/testing loss stop decreasing. (See figure 3 in appendix)
> > > Besides, our work give a thorough analysis on how SMD affect the performance of DNN via controlling angular update (we will further explain its significance below), while [Li et al. (2020)](https://arxiv.org/pdf/2010.02916.pdf) does not even notice it; [Tanaka and Kunin (2021)]( https://arxiv.org/pdf/2105.02716.pdf ), [Kunin et al. (2021)](https://arxiv.org/pdf/2012.04728.pdf) only derive theoretical value of “angular velocity" using stationary assumption, without any justification or further explanation on characteristics of “angular velocity" .
> > >
> > > All in all, the advantage of our analysis is clear: only our work can demonstrate the characteristics of equilibrium phenomenon precisely in both theoretical and experimental aspects.
> > >
> > > > Why does it matter if the scale of the gradients varies? Does this happen in practice?
> > >
> > > It is important that scale of the (expected) gradients can vary in equilibrium. [Li et al. (2020)](https://arxiv.org/pdf/2010.02916.pdf)  claims the sign of equilibrium is convergence of (expected) weight norm, and prove the convergence of weight norm based on the assumption that the scale of the (expected) gradients is stable. But in practice, the scale of the (expected) gradients may vary. We have exhibit an example in Fig 2 (a), (d) in main text: the unit gradient norm increases across the whole training process. Therefore, in Fig 2 (b), (e), the weight norm never converges within total 90 epochs, but the equilibrium has been reached in within first 20 epochs.
> > >
> > > > I found the appendix D vague.
> > >
> > > Though we carefully clarify our difference with most relevant concurrent work [Li et al. (2020)](https://arxiv.org/pdf/2010.02916.pdf), showing their results actually do not depict equilibrium in practice properly. But we still think their theorem is inspiring, and can be a powerful theoretical tool to investigate the equilibrium phenomenon deeper. So in appendix D, rather than just emphasizing the difference between our work and [Li et al. (2020)](https://arxiv.org/pdf/2010.02916.pdf), we focus on demonstrating the **connection** between our results and  [Li et al., 2020](https://arxiv.org/pdf/2010.02916.pdf). We hope by combining advantages of our work and  [Li et al. (2020)](https://arxiv.org/pdf/2010.02916.pdf), we can develop a firm theoretical foundation on how SMD affects the performance of normalized DNN. (So far no work has done that,  [Li et al. (2020)](https://arxiv.org/pdf/2010.02916.pdf) only gives some conjectures, while our work provides intuitive demonstrations and analysis.) We leave it as a future plan.
> > >
> > > >  What does this predict wrongly, that your analysis captures?
> > >
> > > Sorry, we do not catch your point. As far as we concern, all predictions we made in our work can perfectly match our observations in practice. Even the “mismatch” in Sec 4.1. Could you please explain what you want to ask further?
> > >
> > > > Your discussion could also be more balanced - you only discuss disadvantages of other works assumptions, without giving any detail about the disadvantages of your own assumptions
> > >
> > > We discuss much on the disadvantages of of other works assumptions, because they use unreasonable assumptions to derive the theoretical results, which are inconsistent with characteristics of equilibrium phenomenon in practice. We have discussed the weakness of our assumption in lines 197-201 in main text: we use a technical term “adjusted unit gradient norm” without any theoretical justification. We just empirically verify it is approximately equal to "unit gradient norm” in practice. Despite it, we think there is no other disadvantages in our assumptions: our assumption is general and consistent with real data experiments. That is the key reason why our theoretical results can agree very well with empirical observations.
> > >
> > > What we focus more is the limitation of our theorem: first the bounds in our theorem is too large (as we state in lines 227-229 in main text). We think it is mainly caused by our derivations in the proof. Our assumptions should be reasonable enough; Second, our theorem cannot demonstrate how SMD affects the performance of DNN by theoretical justification. Hence, even though we have pointed out the key is angular update (in main text), and demonstrated its connection to generalization performance of DNN by both intuition and experiments (in appendix E, F), this part is still unconvincing for those who are not familiar with this topic. We regard it as a future work.
> > >
> > > **Quality**
> > >
> > > We have varied our theoretical results on three different datasets (Cifar10, Imagenet, COCO), three different network structures (resent, mobilenet, shufflenet). Due to the policy, we cannot present more experimental results since we can neither upload new figures, nor can we update the manuscript. All our experiments settings are standard (cifar10, imagenet, COCO), and the codes are all open source. We will clean up our experiment codes to let readers reproduce our experimental results easily.

---

> > > > ### Comment · Reviewer_uKSg · 2021-08-23
> > > > **Further reply**
> > > >
> > > > **Originality**
> > > > You earlier claim that original aspect of your work is theoretical value of angular update, equation (14):
> > > > > [...] novelty of our work is [...] theoretical value of angular update are both characteristics of SMD
> > > >
> > > > > angular update is the key for SMD to affect the performance of DNN
> > > >
> > > > But basically the same expression for angular update is in Kunin et al. 2021, line 8, page 9. The only difference is theirs is a speed, yours a distance per timestep. To get your distance from theirs, you just need to multiply theirs by the learning rate (tells you how far to move in one step). For me, this difference is not original enough, unless you treat Kunin et al. 2021 as concurrent work, but it's not clear that you are treating it as such. I do note there is a change of emphasis between the papers, but you need to accurately attribute other papers results. I think the stationarity assumption in Kunin et al. 2021 is essentially the same as the limiting case you assume for your theorem 3 (for RHS of eqn 8 to vanish, the weight norm square approaches a constant value, ie the derivative vanishes - this is the stationarity assumption in Kunin).
> > > >
> > > > > All in all, the advantage of our analysis is clear: only our work can demonstrate the characteristics of equilibrium phenomenon precisely in both theoretical and experimental aspects.
> > > >
> > > > I think the above is too strong a claim.
> > > >
> > > > >> Why does it matter if the scale of the gradients varies? Does this happen in practice?
> > > >
> > > > > It is important that scale of the (expected) gradients can vary in equilibrium. Li et al. (2020) claims the sign of equilibrium is convergence of (expected) weight norm, and prove the convergence of weight norm based on the assumption that the scale of the (expected) gradients is stable. But in practice, the scale of the (expected) gradients may vary. We have exhibit an example in Fig 2 (a), (d) in main text: the unit gradient norm increases across the whole training process. Therefore, in Fig 2 (b), (e), the weight norm never converges within total 90 epochs, but the equilibrium has been reached in within first 20 epochs.
> > > >
> > > > I was actually referring to the scale of the gradient *variance*, not its expectation. See my original comment:
> > > > > so the extra piece this work adds here is the rate of approach and the scale of the stochastic variance.
> > > >
> > > > Do you have anything that may convince me of the significance of the scale of the stochastic variance of the gradients?
> > > >
> > > > > We have discussed the weakness of our assumption in lines 197-201
> > > >
> > > > You didn't discuss any disadvantages of your assumptions in these lines.
> > > >
> > > > > Sorry, we do not catch your point. As far as we concern, all predictions we made in our work can perfectly match our observations in practice. Even the “mismatch” in Sec 4.1. Could you please explain what you want to ask further?
> > > >
> > > > This is in relation to scale of stochastic gradient variance, see above.
> > > >
> > > > **Quality**:
> > > > Thank you for running these extra experiments - I trust they agree with your original results, and that you would be able to add these into a camera-ready version if the paper is accepted.

---

> > > > > ### Author Response · Authors · 2021-08-24
> > > > > **Further response**
> > > > >
> > > > > **Originality**
> > > > >
> > > > > First, we have acknowledged [Kunin et al. (2021)](https://arxiv.org/pdf/2012.04728.pdf) as a concurrent work in our main text (line 226) and our previous responses. But we cannot attribute the theoretical value of angular update in equilibrium as a contribution of [Kunin et al. (2021)](https://arxiv.org/pdf/2012.04728.pdf), because, as we state in our paper (line 226 in main text), a much earlier **previous** work [Chiley et al., 2019](https://arxiv.org/pdf/1905.05894.pdf), has derived the theoretical value of angular update in equilibrium (SGD) using stationary assumption (see equation (7) in  [Chiley et al., 2019](https://arxiv.org/pdf/1905.05894.pdf). Due to the existence of  [Chiley et al., 2019](https://arxiv.org/pdf/1905.05894.pdf), we cannot say [Kunin et al. (2021)](https://arxiv.org/pdf/2012.04728.pdf) can provides any orginal results on equilibrium.
> > > > >
> > > > > As for our work, we do not claim “theoretical value of angular update is our novel contribution” either in our paper or in our responses. Our contribution is: we derive it without stationary assumptions. You said our assumption in theorem 3 is essentially similar to stationary assumption in [Kunin et al. (2021)](https://arxiv.org/pdf/2012.04728.pdf), **but we do not assume weight norm square is a constant value in equilibrium**, our theoretical value of weight norm square in equilibrium is a function of squared norm of unit gradients, i.e, weight norm square is **not** a constant value in equilibrium. In this case, the stationary assumption (derivatives of weight norm equals 0) in [Kunin et al. (2021)](https://arxiv.org/pdf/2012.04728.pdf) does not hold true in practice. We also have conducted experiments (fig 2 b, e) to show weight norm square indeed keeps varying in practice, even it has been in equilibrium. We have emphasized this point repeatedly in our paper and our previous responses. Please, read our paper and our previous responses again to get our point.
> > > > >
> > > > > > I think the above is too strong a claim.
> > > > >
> > > > > We say this because all previous/cocurrent works use stationary assumption to anaylze equilibrium, except [Li et al. (2020)](https://arxiv.org/pdf/2010.02916.pdf), who regards it as a result and sign of equilibrium. Our theorems refute the stationary assumption/results. Experiment results also prove our results and predictions.
> > > > >
> > > > >
> > > > > About scale of gradients variance.
> > > > >
> > > > > Please notice the difference among these three terms: expected norm square of gradient, gradient’s expectation, and variance of gradients. We state their relation in Section D in appendix (see equation (182)): expected norm square of gradient = (gradient’s expectation)^2 + variance of gradients. When we refer to the variation of expected norm square of gradient, it means expected norm square of gradient will change across the whole training process, and the variation of expected norm square of gradient will make weight norm square vary in equilibrium, which refutes the stationary assumption from the other relevant works. In main text, we do not distinguish how the scale of gradient's variance or gradient’s expectation affect the variation of expected norm square of gradient since they are not relevant to our main results. But in Sec D in appendix, we analyze it in details because it is key to connect SDE models and practical observations.
> > > > >
> > > > > Again, the variation of expected norm square of gradient will make weight norm square vary in equilibrium, therefore refute the stationary assumptions/results used in [Chiley et al., 2019](https://arxiv.org/pdf/1905.05894.pdf), [Kunin et al. (2021)](https://arxiv.org/pdf/2012.04728.pdf),[Li et al. (2020)](https://arxiv.org/pdf/2010.02916.pdf), etc.  This is its significance.
> > > > >
> > > > > > You didn't discuss any disadvantages of your assumptions in these lines.
> > > > >
> > > > > Despite the fact that we have stated that the meaning of term “adjust unit gradients” we use in assumption (6) in theorem 2 is lack of justification and further interpretations, we used to think our assumptions has no any other disadvantages to discuss. But reviewer VeBT points out his/her concerns on uniformness in our assumption (weakness 1.1, 1.2), we think his/her concerns are very helpful, we will improve the formulation of our assumption and discuss their disadvantages as reviewer VeBT suggests. Please see the comments of reviewer VeBT and our response.

---

> > > > > > ### Comment · Reviewer_uKSg · 2021-08-25
> > > > > > **reply on originality**
> > > > > >
> > > > > > > but we do not assume weight norm square is a constant value in equilibrium
> > > > > >
> > > > > > I see - thank you for clarifying this for me, I missed this in my original reading, sorry for that. I think it would be helpful for readers if, before remark 1, there was a remark about L_t, hence w_t, being non-constant, as this is a crucial difference to previous work. At the moment this isn't discussed until line 168, which was a bit late considering other things were discussed first.
> > > > > >
> > > > > > > Our theorems refute the stationary assumption/results.
> > > > > >
> > > > > > I agree with this now, thank you.
> > > > > >
> > > > > > > When we refer to the variation of expected norm square of gradient, it means expected norm square of gradient will change across the whole training process, and the variation of expected norm square of gradient will make weight norm square vary in equilibrium, which refutes the stationary assumption from the other relevant works
> > > > > >
> > > > > > I now understand that this is as above, L_t, hence w_t, being non-constant, (because L_t is formed from expected norm square of gradient). Thanks for clarifying.
> > > > > >
> > > > > > >  we will improve the formulation of our assumption and discuss their disadvantages as reviewer VeBT suggests.
> > > > > >
> > > > > > I think this is a good idea.
> > > > > >
> > > > > > -------
> > > > > >
> > > > > > Overall I am now more satisified with the originality of this work, compared to prior/concurrent works. I will increase my score accordingly, as this was my main crux for original score.

---

> > > > > > > ### Author Response · Authors · 2021-08-26
> > > > > > > **Thanks**
> > > > > > >
> > > > > > > We are very glad your confusion has been solved. Thank you for increasing your score. More importantly, your suggestions are also very helpful to us, we will follow your suggestions to improve our paper.

---

> > > ### Author Response · Authors · 2021-08-17
> > > **Further response to Reviewer uKSg (2/2)**
> > >
> > > **Clarity**
> > >
> > > >  Section 4.2: What traditional machine learning theorem(s) are you comparing to here?
> > >
> > > In section 4.2, the traditional machine learning theorem(s) we refer to here is “Training too much epochs may cause overfitting issues, leading to decreasing generalization performance”([Tetko, et al., 1995](http://www.vcclab.org/articles/jcics-overtraining.pdf)). A more recent theoretical view is “too small leaning rate will make optimization trajectory trapped into sharp local minimum and lead to overfitting issues”([Keskar, et al., 2016](https://arxiv.org/abs/1609.04836)).  Both theorems cannot explain the  decreasing accuracy phenomenon we show in Sec 4.2. Our theorem provides the third interpretation: due to SMD, learning rate decay leads to an escaping (from local optimum) behavior, resulting in decreasing accuracy phenomenon (because of underfitting).
> > >
> > > > I just wanted to have a clearer motivation for your experiment here. I found appendices E and F to be poorly written. It's not clear what is a conjecture and what is results in those appendices.
> > >
> > > The motivation of contents in section 4.2, appendix E, F is to verify our conclusion: **angular update is the key for SMD to affect the performance of DNN**.
> > >
> > > In appendix E, since we cannot demonstrate the connection between angular update and performance of DNN by firm theorems, we have to take use of intuitional descriptions (lines 400-426, fig 2 in appendix) and conjectures (lines 427-429). Then we conduct experiments to verify our claim and conjectures (lines 430-445, fig 3 in appendix).
> > >
> > > In section 4.2 and appendix F, we try to enhance our conclusion by showing SMD can influence the performance of DNN in an abnormal way. Hence we explain the “escaping behavior” using the intuitional descriptions and conjectures in appendix E. We also conduct rescaling experiments to verify our explanations.
> > >
> > > We will refine the contents in section 4.2, appendix E, F in next version.
> > >
> > > > I also remain unconvinced about introducing a new term "spherical motion dynamics" for motion which is not spherical, and does not make clear that it is just SGDM with WD on a normalised neural network - why does this warrant a new term, the naming of which is misleading?
> > >
> > > First, please notice the intrinsic domain of a normalized neural network is actually a unit sphere. The optimization trajectory in weight space is equivalent to a trajectory projected on a unit sphere, where each update size on this sphere is exactly equal to angular update. So it is appropriate to call this process as "Spherical motion"; Besides, Spherical Motion can highlight its key characteristic: the weight norm is determined by the tug-of-war between “centrifugal force” and “centripetal force” . This tug-of-war effects lead to all phenomena of SMD.
> > >
> > > Second, we think it is worthy to give a new term and pay special attention on such abnormal behavior of SGDM with WD on normalized network. We need to remind the reviewer that "SGDM with WD on normalized network” accounts for most of deep learning training settings in practice. But few theorems can provide reasonable explanations on why such a naive optimization algorithm often gets best performance in practice. One of the core reasons, as [Li et al. (2020)](https://arxiv.org/pdf/2010.02916.pdf) has pointed out, is that the commonly used assumptions in many theoretical work are not consistent with practical observations. Now our work takes use of proper assumptions, finds a unique characteristic of "SGDM with WD on normalized network”, and confirms its abnormal effect to performance of network. We believe our work provides a new chance for theorists to investigate the intrinsic advantages of SGD(M) in more practical settings.
> > >
> > > **significance**
> > >
> > > > I wanted the reply here to be about what I consider the novel findings (rate of approach and scale of variance, and including momentum), and why they specifically are important and useful to the practitioner. Instead it just referred back to the paper, which only speaks about known parameters, such as angular update, but not it's rate of approach. I conclude that the authors are not aware of the usefulness of the novel aspects for the practitioner.
> > >
> > > We do not talk too much about the rate of approach or scale of variance, because we think our analysis on angular update are more significant and useful for practitioners than rate of approach or scale of variance. Here is our reasons:
> > >
> > > 1. As we have explained before, the cause of equilibrium is independent of performance of DNN. Equilibrium can be reached long before the performance of DNN remains stable. So rate of approach cannot provide any reliable clues on the convergence rate of DNN;
> > > 2. We have shown SMD can affect performance of DNN only by angular update, and angular update is only determined by hyper-parameters in equilibrium. That means we can manually set the value of angular update equal to its theoretical value in each step, then the learning dynamics of DNN will be in equilibrium at very beginning, neither rate of approach nor scale of variance matter at all.
> > >
> > > The rate of approach and scale of variance only contribute to confirming the existence of equilibrium phenomenon in practice. Angular update owns more potential significance since its behavior is relevant to the performance of DNN during training. We think it is unfair to regard angular update as a “known parameter”, since no previous work has ever discussed the property and significance of angular update seriously.
> > >
> > > As for momentum, we have discussed its effect to SMD and the performance of DNN in Section E (lines 437-445, Fig 3 in appendix). We acknowledge our current is brief, because the role of momentum remains mysterious. We regard it as another further work.
> > >
> > > Hopefully our response help can clarify your concerns, and you would re-evaluate your score. Please let me know if you have other questions.

---

> > > > ### Comment · Reviewer_uKSg · 2021-08-23
> > > > **Further Reply**
> > > >
> > > > **Clarity**
> > > > > We will refine the contents in section 4.2, appendix E, F in next version.
> > > >
> > > > Thank you - in order for me to increase my score for clarity, please supply text for the revision for this section. You should state clearly what is the motivation for this experiment, what is the experimental setup, what is the experimental hypothesis, and what is experimental result, and a discussion of whether the result confirms the hypothesis or not.
> > > >
> > > > > First, please notice the intrinsic domain of a normalized neural network is actually a unit sphere. The optimization trajectory in weight space is equivalent to a trajectory projected on a unit sphere
> > > >
> > > > I am confused now as to whether the motion is actually spherical or not, see your earlier reply:
> > > >
> > > > > It does not refer to the motion on a fixed sphere
> > > >
> > > > Please clarify. Either way, I still disagree with introducing a new term here for reasons in earlier reply.
> > > >
> > > > > "SGDM with WD on normalized network” accounts for most of deep learning training settings in practice.
> > > >
> > > > I disagree as the requirement to have every layer to have normalisation is not the case in practice.
> > > >
> > > > **Significance**
> > > > As mentioned in my comment on originality, I think the main originality is the rate of approach and scale of variance. The authors here agree that they don't go into details on these aspects, which I believe is necessary for this paper to be more significant. As such I won't update my score based on significance.

---

> > > > > ### Author Response · Authors · 2021-08-24
> > > > > **Further response**
> > > > >
> > > > > **Originality**
> > > > >
> > > > > We decide to rewrite section 4.2 as reviewer VeBT's suggestions (see weakness 4.2), we will also follow your instructive suggestions (*“ You should state clearly what is the motivation for this experiment, what is the experimental setup, what is the experimental hypothesis, and what is experimental result, and a discussion of whether the result confirms the hypothesis or not.”*), thanks for that. We will update this section after final decision (either in the camera-ready version if this paper is accepted or in the submission to the next venue).
> > > > >
> > > > > About the term “spherical motion dynamics (SMD)”
> > > > >
> > > > > We insist a new term is necessary, since the phenomenon denoted by SMD is very important (the other two reviewers both agree with the importance of our findings on SMD). But we agree with you that current name is probably misleading for the readers who are not familiar with this topic. We will clarify this confusion or come up with a new name.
> > > > >
> > > > > > "I disagree as the requirement to have every layer to have normalisation is not the case in practice.”
> > > > >
> > > > > We do **not** say every layers(weights) has normalization, our theorem only focus on the normalized layers(weights). Even in our experiments, we only shows the weights from convolutional layers, which have batch normalization. The other weights like fully connected layer do not have normalization. Our experiments shows these unnormalized part do not affect our prediction on normalized part. As long as our assumptions are satisfied.

---

> > > > > > ### Comment · Reviewer_uKSg · 2021-08-25
> > > > > > **reply to further response**
> > > > > >
> > > > > > > We will update this section
> > > > > >
> > > > > > Thank you.
> > > > > >
> > > > > > > current name is probably misleading
> > > > > >
> > > > > > Noted
> > > > > >
> > > > > > > We do not say every layers(weights) has normalization, our theorem only focus on the normalized layers(weights).
> > > > > >
> > > > > > Sorry for my confusion here, that is clear to me now, no change required here.

---

### Official Review · Reviewer_Zj3b · 2021-07-19

**Rating:** 8
**Confidence:** 4

**Summary:**

This paper studies the dynamics of Spherical Motion Dynamics (Stochastic Gradient Descent with momentum and Weight Decay), more specifically, the behavior of “effective learning rate" in the “equilibrium" state. The paper (1) gives the assumptions for the equilibrium state and prove the equilibrium can be reached in linear time; (2) describes an “angular update” as a substitute for effective learning rate to depict the state of SMD; (3) verifies the assumptions and theories on ImageNet and MSCOCO.

**Limitations And Societal Impact:**

Yes. The authors discuss the conflict between the theory and experiments in remark 3 though it is not so serious.

**Main Review:**

Pros:

+ This paper gives solid and rigorous theoretical results on the dynamics of Spherical Motion Dynamics. The study is timely and important. The proofs are given in detail. The results are well motivated. It may shed light in understanding deep learning (while this part is not discussed sufficiently; see below).

+ The theoretical results are well motivated. The authors give sufficient discussion to the idea of the theoretical study.

+ The paper gives clear proof sketch. The proof idea is clearly presented. Readers can easily follow the idea without checking the whole proof details in the appendices.

+ The paper gives comprehensive empirical results on very large-scale datasets, ImageNet and MSCOCO. The empirical results are in full agreement with the theory. This is much impressive for a theory paper.

Cons:

- The paper does not give a sufficiently comprehensive literature review. The presented contributions are not well positioned in the current state of research. Readers may fill confusing for some of the results. The advantages and difference of this work from the existing results.

- The paper does not give enough practical implications of the presented theory. The authors are encouraged to discussion how this work can contribute to future novel algorithm design.

- The paper has many typos. It seems written in a rush. I recommend the authors carefully proofread the paper in the next version.

**Time Spent Reviewing:**

one day

---

> ### Author Response · Authors · 2021-08-10
> **Response to Reviewer Zj3b**
>
> Very appreciate for your affirmation and helpful suggestions, we will respond to your concerns one by one:
>
> > “The paper does not give a sufficiently comprehensive literature review. The presented contributions are not well positioned in the current state of research. Readers may fill confusing for some of the results. The advantages and difference of this work from the existing results.”
>
>  As we state in lines 112-113, due to the limitation of length, we present the most related works (about “equilibrium” phenomenon of normalized DNN) in main text, and  have to place the thorough review in  Sec B in appendix. The research line on “equilibrium” of normalized neural network has a long history since [van Laarhoven (2017)](https://arxiv.org/pdf/1706.05350.pdf), but this research line has not been widely accepted by mainstream optimization/theoretical community (we say this because few works take the “equilibrium” phenomenon into account while analyzing dynamics of SGD(M) or designing new optimization algorithm, even most of they conduct experiments on normalized DNN). As we state in lines 53-69, the core reason is that all previous related works neither theoretical justify the existence of “equilibrium”, nor showed “equilibrium” phenomenon by experiments until recently ([Li et al., 2020](https://arxiv.org/pdf/2010.02916.pdf)).
>
> But [Li et al., (2020)](https://arxiv.org/pdf/2010.02916.pdf), and another two concurrent work ([Tanaka and Kunin, 2021]( https://arxiv.org/pdf/2105.02716.pdf ), [Kunin et al., 2021](https://arxiv.org/pdf/2012.04728.pdf) ) only provide indirect evidence by continuous model (We discuss the connection and difference between our work and these three most related work in Sec D in appendix). We regard our work as a final conclusion, showing “equilibrium” is a ubiquitous phenomenon in deep learning tasks rather than just a concept or a theorem. We also show how SMD affects the performance during training. We hope our work can inspire other works on analyzing dynamics of SGD(M) or designing new optimization algorithm outperforming SGD(M).
>
>  > “The paper does not give enough practical implications of the presented theory. The authors are encouraged to discussion how this work can contribute to future novel algorithm design.”
>
> We have provided two practical implications in this work: first we show how SMD can lead to pseudo dropping evaluation phenomenon in practice (Sec. 4.1, 4.2). Due to the limitation of length, we can only present a brief demonstration in main text (Sec. 4.2). The full demonstration is placed in appendix (Sec. F); second, we demonstrate geometric meaning of angular update, and the connection between angular update and performance of normalized neural network by experiments in appendix (Sec E).
>
> Besides, we have discussed the insight one can get from our theory when designing novel algorithm: angular update should be directly controlled rather than just tuning learning rate or weight decay factor (lines 237-242 in main text, Sec E in appendix). But we do not design new optimization algorithm based our theoretical result (like many other similar type of works), because we cannot conclude what is beneficial to control angular update in optimization so far: On one hand, the layer-wise fixed angular update manner caused by SMD seems to be one of the key reasons why SGD(M) is so simple but can usually outperforms other fancy algorithms (which are supposed to own better theoretical properties) in practice. There has been a type of algorithms, named [LARS]( https://arxiv.org/abs/1708.03888 ), which have taken the layer-wise fixed angular update manner into account and gained great success in large batch training tasks. (Note LARS has been proposed long time ago, but the connection between LARS and equilibrium of SGD was not recognized then. We has stated this in lines 96-102 in appendix); On the other hand,  Adam(W) doesn't own a clear SMD pattern as SGD, but Adam(W) can outperform SGD(M) in specific tasks like [vision transformer]( https://arxiv.org/pdf/2010.11929.pdf ).
>
> In summary, we show controlling angular update directly is the key to design more powerful optimization algorithm. We will highlight this discussion in next version. But how to control angular update better is still a problem, we leave this as a future work.
>
> > “The paper has many typos. It seems written in a rush. I recommend the authors carefully proofread the paper in the next version.”
>
> Thanks for your reminder, we will correct these typos in next version.

---

### Decision · Program_Chairs · 2021-09-27

**Decision:**

Accept (Spotlight)

**Comment:**

This is a clear accept.  Aside from the high scores of all the reviewers, I myself have seen versions of this paper in the past on arxiv (it was inexplicably rejected at a previous submission, but is even improved here) and have really benefitted from this work's thinking.  I think the contribution of this work is clear and important.   I think we'd be foolish not to accept this work here this year.